# Structures of a deAMPylation complex rationalise the switch between antagonistic catalytic activities of FICD

Luke A. Perera [1✉], Steffen Preissler [1], Nathan R. Zaccai [1], Sylvain Prévost [2], Juliette M. Devos [2], Michael Haertlein[2] & David Ron [1✉]

The endoplasmic reticulum (ER) Hsp70 chaperone BiP is regulated by AMPylation, a reversible inactivating post-translational modification. Both BiP AMPylation and deAMPylation are catalysed by a single ER-localised enzyme, FICD. Here we present crystallographic and solution structures of a deAMPylation Michaelis complex formed between mammalian AMPylated BiP and FICD. The latter, via its tetratricopeptide repeat domain, binds a surface that is specific to ATP-state Hsp70 chaperones, explaining the exquisite selectivity of FICD for BiP's ATP-bound conformation both when AMPylating and deAMPylating Thr518. The eukaryotic deAMPylation mechanism thus revealed, rationalises the role of the conserved Fic domain Glu234 as a gatekeeper residue that both inhibits AMPylation and facilitates hydrolytic deAMPylation catalysed by dimeric FICD. These findings point to a monomerisation-induced increase in Glu234 flexibility as the basis of an oligomeric state-dependent switch between FICD's antagonistic activities, despite a similar mode of engagement of its two substrates — unmodified and AMPylated BiP.

[1] Cambridge Institute for Medical Research, University of Cambridge, Cambridge, UK. [2] Institut Laue-Langevin, Grenoble, France. ✉email: lp397@cam.ac.uk; dr360@medschl.cam.ac.uk

The endoplasmic reticulum (ER) Hsp70, BiP, dominates the chaperoning capacity of the organelle[1]. BiP's abundance and activity are matched to the unfolded protein load of the ER at the transcriptional level, by the canonical UPR, and also post-translationally[2]. BiP AMPylation, the covalent attachment of an ATP-derived AMP moiety to the Thr518 hydroxyl group, is perhaps the best-defined BiP post-translational modification. AMPylation inactivates BiP by biasing it towards a domain-docked, linker-bound ATP-like Hsp70 state and away from the domain-undocked, linker-extended ADP-like state[3–5]. As such, AMPylated BiP (BiP-AMP) exhibits high rates of substrate dissociation and is refractory to ATPase stimulation by J-domain proteins[3–5].

BiP AMPylation inversely correlates with the ER protein folding load, increasing upon the inhibition of protein synthesis[6] and with a resolution of ER stress[3]. Conversely, as ER stress mounts, inactivated BiP-AMP is recruited into the chaperone cycle by deAMPylation[3,6,7].

A single bifunctional enzyme, FICD, is responsible for both AMPylation[3,8,9] and deAMPylation[10–12] of BiP. FICD is the metazoan exemplar of a family of bacterial Fic domain proteins[13] whose canonical AMPylation activity[14–16] is often autoinhibited by a glutamate-containing alpha helix ($\alpha_{inh}$)[17,18]. In FICD, the AMPylation-inhibiting Glu234 is also essential for deAMPylation[10]. Moreover, monomerisation is able to reciprocally regulate FICD's AMPylation/deAMPylation activity, converting the dimeric deAMPylase into a monomeric enzyme with primary BiP AMPylating functionality[19]. The recent discovery that the *Enterococcus faecalis* Fic protein (EfFic) possesses deAMPylation activity which is dependent on a glutamate homologous to FICD's Glu234[12], suggests conservation of the catalytic mechanism amongst Fic enzymes. However, the role of Glu234 in the oligomeric state-dependent regulation of FICD's mutually antagonistic activities remains incompletely understood.

Fic domain proteins are unrelated to the two known bacterial deAMPylating enzymes, SidD and the bifunctional GS-ATase. Both catalyse binuclear $Mg^{2+}$-facilitated deAMPylation reactions of a hydrolytic[20] and phosphorolytic[21] nature, utilising a metal-dependent protein phosphatase[20] and a nucleotidyl transferase[22,23] protein-fold, respectively. Fic proteins have a single divalent cation binding site and are evolutionarily and structurally divergent from these deAMPylases and, therefore, likely catalyse a distinct deAMPylation mechanism.

In addition to the aforementioned enzyme-based regulatory mechanism(s), there is evidence that AMPylation is also regulated by substrate availability (the concentration of free BiP:ATP). Cells with a constitutively monomeric FICD retain a measure of regulated BiP AMPylation[19]. FICD specifically binds and AMPylates the domain-docked ATP-state of BiP[3,19]. Client binding partitions Hsp70s away from their ATP-state, suggesting a simple mechanism for coupling BiP AMPylation to low protein folding loads. Furthermore, the finding that FICD selectively AMPylates and deAMPylates ATP-state-biased BiP suggests that FICD may recognise ATP-state specific features of its substrate in a conserved binding mode, that is independent of FICD's oligomeric-state or BiP modification status.

Here we present a structure-based approach to determine the nature of the FICD-BiP enzyme–substrate interaction, thereby elucidating the mechanism of eukaryotic deAMPylation and the basis for its regulation by an oligomerization-based switch in FICD's functionality.

## Results

**FICD engages AMPylated BiP via a bipartite interaction surface.** Mutation of the Fic motif catalytic histidine, which acts as an

essential general base in the AMPylation reaction[14,15,24], eradicates FICD's deAMPylation activity[10]. Upon mutation of this histidine (His363Ala), FICD and BiP-AMP formed a long-lived, trapped deAMPylation complex[19]. This feature was exploited to copurify FICD and AMPylated BiP by size-exclusion chromatography (SEC). A complex of otherwise wild-type dimeric FICD$^{H363A}$ and AMPylated BiP readily crystallised, but despite extensive efforts, these crystals did not yield useful diffraction data. However, the introduction of a monomerising Leu258Asp mutation and truncation of BiP's flexible α-helical lid facilitated purification of a heterodimeric FICD$^{L258D-H363A}$·BiP$^{T229A-V461F}$-AMP complex (Supplementary Fig. 1a; see the "Methods" section) that crystallised and produced two very similar sub-2 Å datasets (Table 1).

The crystal structures displayed (identical) extensive bipartite protein–protein interfaces totalling 1366 Å$^2$ (Fig. 1a and Supplementary Fig. 1b–d; state 1 crystal structure is shown). The deAMPylation substrate, AMPylated BiP, is in a domain-docked ATP-like state (despite lacking bound ATP), as reflected by the similarity with the isolated ATP-state BiP-AMP structure[4] (Fig. 1a; 1.02 Å root-mean-square deviation (RMSD) across all 521 Cα pairs). The FICD tetratricopeptide repeat domain motif 1 (TPR1) contacted a tripartite BiP surface (695 Å$^2$), comprised of its nucleotide-binding domain (NBD), interdomain linker and substrate-binding domain-β (SBDβ) (Fig. 1b(i)). The interacting residues located on FICD(TPR) are particularly well-conserved across metazoan FICD homologues (Supplementary Fig. 1c).

The second interface, by which FICD's catalytic Fic domain engaged BiP's SBDβ (671 Å$^2$), contained an intermolecular

---

**Table 1 Data collection and refinement statistics.**

|  | FICD·BiP-AMP DeAMPylation complex (State 1) | FICD·BiP-AMP DeAMPylation complex (State 2) |
|---|---|---|
| *Data collection* |  |  |
| Space group | $P2_12_12$ | $P2_12_12$ |
| *Cell dimensions* |  |  |
| a, b, c (Å) | 95.37, 104.08, 105.63 | 95.00, 103.89, 104.79 |
| α, β, γ (°) | 90.00, 90.00, 90.00 | 90.00, 90.00, 90.00 |
| Resolution (Å) | 105.63–1.70 | 52.40–1.87 |
|  | (1.73–1.70) | (1.92–1.87) |
| $R_{merge}$ | 0.085 (1.299) | 0.087 (1.793) |
| <$I/\sigma I$> | 10.3 (1.2) | 11.9 (1.0) |
| $CC_{1/2}$ | 0.992 (0.585) | 0.999 (0.536) |
| Completeness (%) | 99.8 (99.3) | 100.0 (100.0) |
| Redundancy | 6.6 (6.5) | 6.6 (6.9) |
| *Refinement* |  |  |
| Resolution (Å) | 74.25–1.70 | 52.40–1.87 |
| No. of reflections | 115,633 (5639) | 86,247 (6270) |
| $R_{work}/R_{free}$ | 0.195/0.221 | 0.201/0.228 |
| *No. of atoms* |  |  |
| Protein | 6667 | 6901 |
| Ligand/ion | 34/1 | 79/5 |
| Water | 960 | 590 |
| *<B-factors> (Å$^2$)* |  |  |
| Protein | 30.78 | 39.75 |
| Ligand/ion | 29.93/32.51 | 51.71/30.25 |
| Water | 40.09 | 42.72 |
| *R.M.S. deviations* |  |  |
| Bond lengths (Å) | 0.003 | 0.003 |
| Bond angles (°) | 1.171 | 1.199 |

Both of the deAMPylation complexes contain human FICD$^{L258D-H363A}$ (residues 104–445) bound to Chinese hamster BiP$^{T229A-V461F}$-AMP (residues 27–549). Note, over the residue range 27–549 hamster and human BiP have identical amino acid sequences. Values in parentheses correspond to the highest-resolution shell.

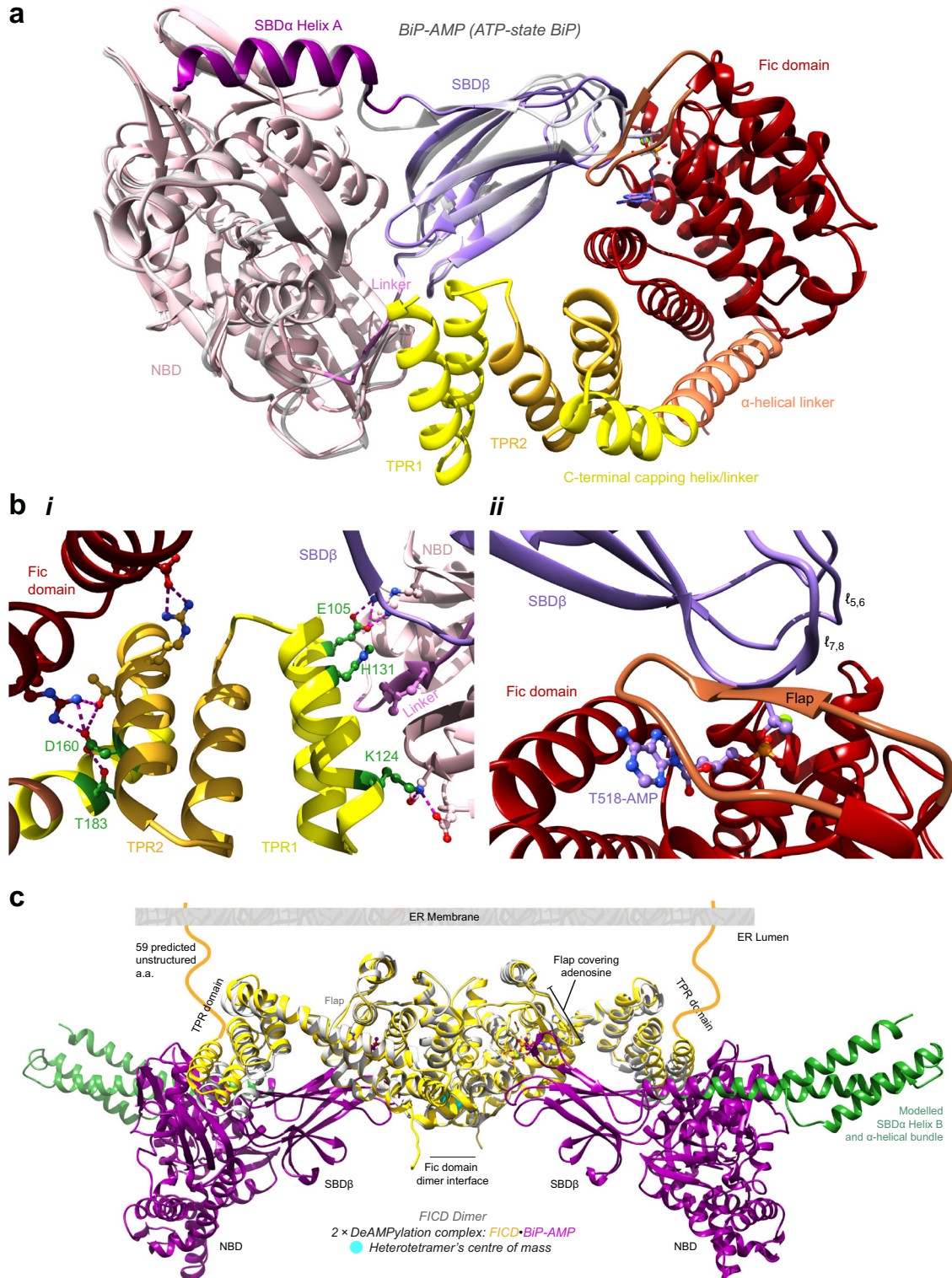

**Fig. 1 The deAMPylation complex crystal structure. a** The deAMPylation complex crystal structure is colour-coded to illustrate its (sub)domain organisation. The ATP-state structure of isolated BiP-AMP (PDB 5O4P, light grey)[4] is superimposed via alignment of its NBD. See Supplementary Fig. 1b–d. **b** A focus on the two intermolecular interaction surfaces. Selected interdomain contacting residues are shown. Polar interactions are depicted by pink dashed lines. Residues mutated in this study are shown in green. **i** The interaction of FICD(TPR1) with the tripartite BiP surface (NBD-linker-SBDβ) is highlighted alongside the intramolecular contacts between the FICD's TPR and Fic domain. **ii** The intermolecular β-sheet formed between the Fic domain flap (brown) and the Thr518 bearing BiP(SBDβ) loop ($\ell_{7,8}$) are highlighted. Note, this sequence-independent mode of substrate engagement is characteristic of Fic proteins[15,18] (see Supplementary Fig 2). **c** Superposition of two heterodimeric crystal structures (purple BiPs and yellow FICDs) with an FICD dimer structure (PDB 4U0U, grey)[24]. The α-helical BiP lid (SBDα, green, missing from the heterodimeric crystal structure) is modelled by alignment with the full-length BiP:ATP structure (PDB 5E84)[71]. The N-terminal unstructured region of FICD is shown in the context of an ER membrane[24] (see Supplementary Movie 1).

β-sheet between BiP's Thr518-bearing loop ($\ell_{7,8}$) and the Fic domain flap. The Fic domain flap has previously been implicated in a similar sequence-independent recognition of AMPylation-substrate or pseudo-substrate binding[15,18] (Supplementary Fig. 2). However, the previously observed, flap-mediated, hydrophobic clamping of the AMPylation target residue[15,18] is not observed here, suggesting that this is not a universal feature of Fic-substrate engagement (Supplementary Fig. 2c, d).

The AMP, covalently attached to BiP's Thr518, was inserted into the Fic domain active site, with the adenosine occupying the same position as in FICD:nucleotide complexes[19,24] (Fig. 1b(ii) and Supplementary Fig. 1b). Contacts between the AMP moiety and the FICD active site contributed an additional 306 Å² interaction surface to the deAMPylation complex.

Monomeric FICD retains deAMPylation activity[11,19], although reduced relative to that of the dimeric enzyme[19]. Superposition of two monomeric FICD-containing deAMPylation complexes (state 1) with a dimeric FICD structure (PDB 4U0U; 2.58 Å RMSD over 33 4Cα pairs across each FICD protomer)[24], demonstrates that the heterodimeric deAMPylation crystal structure is readily accommodated in a hypothetical deAMPylation complex of dimeric FICD engaging two full-length BiP-AMP molecules (Fig. 1c). The alignment also reveals a subtle intra-TPR domain movement (especially in the TPR1 motif region) within the substrate-bound FICD. This movement of the TPR1 motif away from the Fic domain catalytic core likely results from the direct interaction of FICD(TPR) with the tripartite BiP surface. Furthermore, the modelled heterotetrametric complex is compatible with FICD's presumed orientation within the ER[24,25], as it can be readily anchored to the membrane via the extended unstructured region linking FICD's transmembrane and TPR domains (Fig. 1c and Supplementary Fig. 1b).

**The deAMPylation complex crystal structure is representative of the internal arrangement of dimeric FICD engaged with AMPylated BiP in solution.** To assess the validity of the structural insights gained from the heterodimeric deAMPylation complex crystal (obtained with monomeric FICD$^{L258D-H336A}$ and a lid-truncated BiP-AMP), the properties of the intact protein complex was analysed by a solution-based structural method. Low-resolution structures of biomacromolecules can be resolved by small-angle X-ray and neutron scattering (SAXS/SANS)[26–29]. SAXS is sensitive to electron density, while SANS is sensitive to atomic nuclei. For mixed complexes with two components, contrast variation SANS is able to distinguish between proteins that are differentially isotopically labelled[27] and provide information pertaining to protein complex size, shape and internal arrangement with high precision[26]. To enable this analysis, complexes of partially deuterated and non-deuterated dimeric FICD$^{H363A}$ and full-length BiP-AMP were copurified by SEC into buffers with varying $D_2O$ content. Contrast variation solution scattering data were subsequently collected (Fig. 2a).

Analysis of the low-$q$ Guinier region (Fig. 2b and Supplementary Fig. 3a) provided information pertaining to the forward scattering, $I(0)$, and radius of gyration, $R_g$, in each solution. The former, along with the calculation of each complex's contrast match point (CMP; Fig. 2c), provided an estimate of the complex molecular weight (Supplementary Table 1)—which was in good agreement with a FICD·BiP-AMP 2:2 complex. The Stuhrmann plot (derived from the square of the $R_g$ data against the reciprocal of the contrast)[30] provided information on the internal arrangement of the heterotetramer (assigning FICD to the inside of the complex) and size ($R_g$) of the overall complex and its constituent components (Fig. 2d); all of which are consistent with those

calculated from the modelled heterotetramer structure (Supplementary Table 1, Fig. 1c and Supplementary Movie 1).

In addition, the relatively linear Stuhrmann plot shape derived from the deAMPylation complex containing partially deuterated FICD, suggests that this complex has a scattering length density (SLD) centre which is very close to the complex's centre of mass (COM). The converse is true for the partially deuterated BiP complex's Stuhrmann fit that reveals no overlap between the latter's SLD centre and COM. As partial-deuteration of a component increases its relative contribution to the SLD, these findings are consistent with a heterotetramer in which the complex centre of mass is closer to the centre of mass of the two FICD molecules than the centre of mass of the two BiP molecules. This arrangement fits well the structural model presented in Fig. 1c and Supplementary Movie 1.

Moreover, across the entire scattering range and at all $D_2O$ concentrations, the theoretical scattering profile of the heterotetramer (modelled in Fig. 1c) nicely correlated with the observed experimental scattering, producing an overall average $\chi^2$ of 3.3 ± 4 (mean ± standard deviation (SD)) or 2.4 ± 2 following anomalous dataset removal (Fig. 2a and Supplementary Fig. 3b). This was true even at $D_2O$ concentrations close to the CMP for each deAMPylation complex, where the scattering profile is very sensitive to both the shape and stoichiometry of the particles in solution. Furthermore, the best flex-fit structure (generated for each scattering dataset by allowing the input structure to undergo normal mode flexing of its domains) did not drastically improve model fitting (Fig. 1c, green dashed lines and Supplementary Fig. 3b). The SANS data thus indicate that the vast majority of particles in solution are engaged in a heterotetramer with neutron scattering properties predicted by a model based on the heterodimer crystal structure.

By analysing the data over the entire scattering $q$-range, through flex-fitting, it is also possible to capture some of the dynamics of the protein complex in solution (as demonstrated recently[29]). Although no individual flex-fit structure produced a significantly reduced average $\chi^2$ across all datasets, a number of flex-fit output structures did have significantly different and reduced $\chi^2$ variance (Supplementary Fig. 3c, underlined). The majority of flex-fit structures possessed $R_g$ parameters which were in good agreement with the Stuhrmann-derived $R_g$ values (Supplementary Fig. 3d) and the principal variation in the flex-fit structures was evident in BiP(NBD) and FICD(TPR) domain reorientation and in the BiP lid region (Supplementary Fig. 3e, f). Only around half of the flex-fit output structures maintained the C2 rotational symmetry present in the input heterotetramer structure (Supplementary Fig. 3c, d, bold), which stems from the C2 symmetry of the FICD dimer. As symmetry is expected for an average solution structure of a (symmetrical) dimeric FICD fully occupied at two independent BiP-binding sites, each flex-fitting strategy yielded one best-fit structure which was both symmetrical and had a significantly reduced $\chi^2$ variance (Fig. 2e, Supplementary Fig. 3g, c and d [bold and underlined] and Supplementary Movie 1).

Interestingly, the best-fit structure derived from leaving the high-affinity FICD dimer interface unconstrained (mean $\chi^2$ goodness-of-fit across the reduced data set 1.7 ± 0.4) maintains an intact dimer interface and is in fact more similar in overall conformation to the input structure than that obtained with a restrained dimer-interface (mean $\chi^2$ 2.4 ± 0.8), with an RMSD of 5.4 and 7.1 Å (across 1892 Cα pairs), respectively. Both output structures demonstrate good $R_g$ agreement with the Stuhrmann analysis. Importantly, the complexes' FICD $R_g$s are increased, and in better agreement with the experimentally derived values, relative to the input structure (Supplementary Fig. 3d and

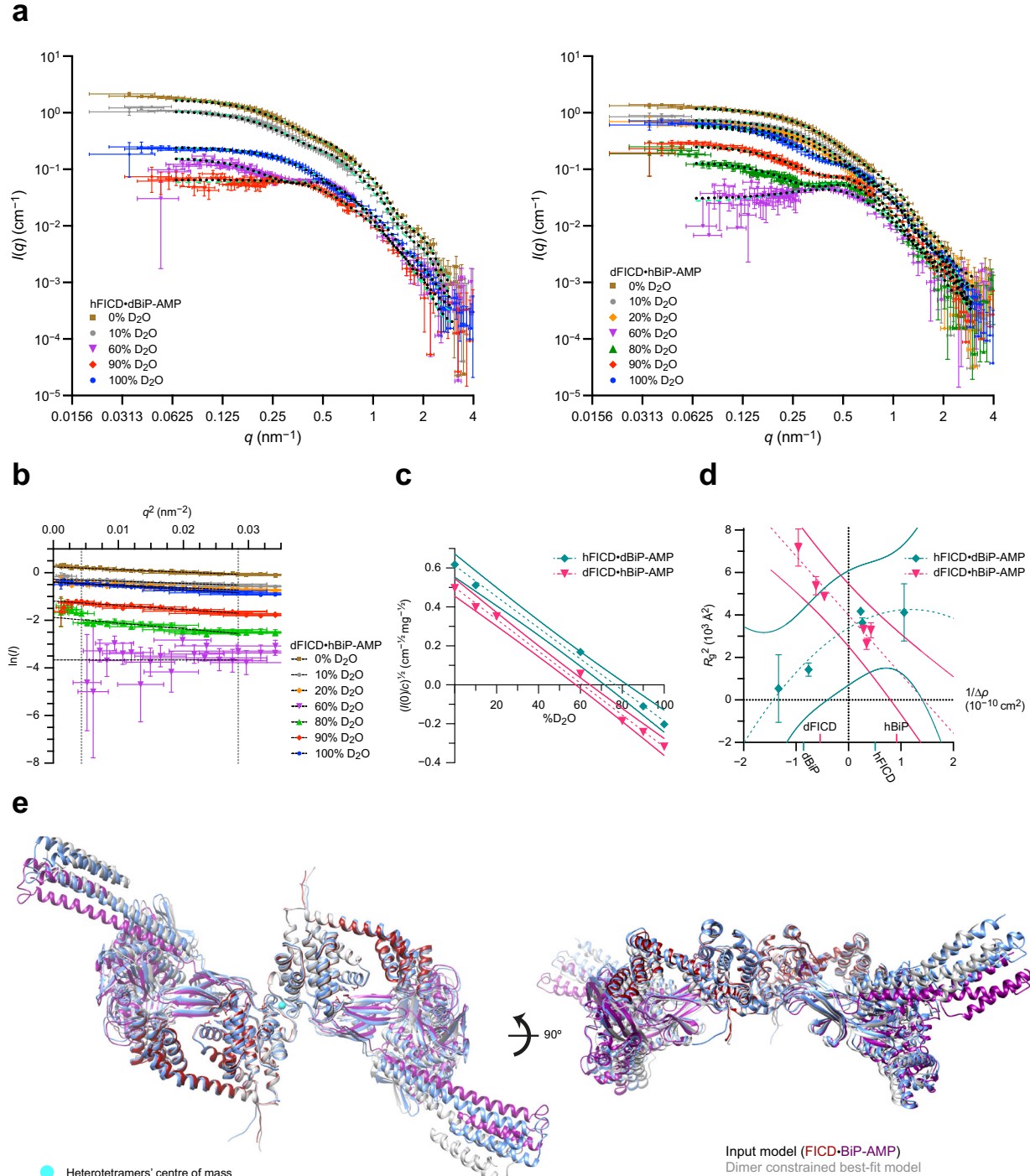

**Fig. 2 The DeAMPylation complex probed by small-angle neutron scattering (SANS). a** Contrast-variation SANS curves of copurified dimeric FICD and full-length AMPylated BiP. Overlaid dotted black lines are theoretical scattering curves based on the modelled heterotetramer shown in Fig. 1c, dashed green lines are the theoretical scattering curves from flex-fitting of the input heterotetramer model to each scattering curve individually (with a constrained FICD dimer interface) (see Supplementary Fig. 3b–e). In each experiment 'd' and 'h' refers to the partially deuterated and non-deuterated components, respectively. Mean values are shown alongside error bars representing the standard error of the mean (SEM) calculated with respect to the number of pixels used in the radial data averaging. **b** Guinier plot of partially deuterated FICD and non-deuterated AMPylated BiP (derived from the scattering data shown in **a**, with mean ± SEM values plotted). Vertical, grey dotted lines represent the $q$-range for fitting the linear best-fit curves (black dashed lines) (see Supplementary Fig. 3a). **c** Scattering amplitude plots. Linear best-fits are shown with dashed lines and 95% confidence interval bands are shown with colour-matched solid lines. **d** Stuhrmann plot with best-fit dashed curves. 95% confidence prediction bands are shown with solid lines. The determined match points of the individual complex components are indicated on the x-axis. In **c** and **d** best-fit values are shown alongside the standard errors of the Guinier fit parameters. **e** Optimal flex-fit structures with respect to overall agreement of theoretical scattering to all experimental contrast-variation SANS datasets. Output structures are aligned to the input heterotetramer model, itself derived by imposing the C2 symmetry of the FICD dimer (PDB 4U0U)[24] onto the heterodimeric deAMPylation complex crystal structure as in Fig. 1c. See Supplementary Movie 1. Source data are provided as a Source Data file.

Supplementary Table 1). In addition, the model derived from flex-fitting without constraints on the FICD dimer interface (as well as possessing the overall lowest average $\chi^2$ value and $\chi^2$ variance) also possesses a BiP $R_g$ which is fully consistent with the Stuhrmann analysis from both oppositely labelled deAMPylation complexes (Supplementary Table 1). Therefore, the observed model deviation is indicative of additional deAMPylation complex flexibility in solution, in particular in the composite FICD(TPR)–BiP(NBD) interface and in the disposition of the BiP lid (Supplementary Movie 1). This flexibility is inaccessible to the crystallographic analysis of BiP (complexes) but is consistent with previous observations of Hsp70 conformational dynamics in the Hsp70 ATP-state[5,31].

**Engagement of the FICD TPR domain with BiP-AMP is essential for complex assembly and deAMPylation.** To test the importance of contacts between FICD's TPR domain and BiP in complex formation, catalytically inactive (His363Ala) but structurally intact FICD variants (Supplementary Fig. 4a–c) were analysed for their ability to interact with immobilised BiP by BioLayer Interferometry (BLI). As FICD selectively binds to the ATP-state of BiP[19], BiP was pre-incubated with MgATP (Supplementary Fig. 4d). Consistent with previous findings[19], BiP bound more tightly to monomeric FICD$^{L258D-H363A}$ than to dimeric FICD. The converse was true for AMPylated BiP. As noted previously[19], complex dissociation was accelerated by the presence of ATP in the dissociation buffer (Fig. 3a); likely a reflection of an allosteric effect on FICD (when engaging unmodified BiP:ATP)[19], or competition for FICD's active site (when engaging BiP-AMP). Upon removal of the TPR1 motif, dimeric FICD lost all appreciable binding to either BiP ligand, whereas, the isolated TPR domain measurably interacted with both BiP ligands irrespective of their modification status (Fig. 3a).

Mutation of residues at the FICD(TPR1)–BiP interface significantly affected the association and dissociation of both monomeric and dimeric FICD variants (Fig. 3b and Supplementary Fig. 4e). This agrees with the idea (supported by small-angle scattering data) that monomeric and dimeric FICD similarly engage AMPylated BiP. In keeping with the crystallographically observed multivalent nature of the deAMPylation complex, the biphasic kinetics of FICD$^{L258D-H363A}$.BiP-AMP interaction becomes more monophasic upon disruption of FICD(TPR1)–BiP contacts (Fig. 3b(i)).

As previously observed[19], FICD's TPR domain can fully disengage from the linker helix, exhibiting a 'TPR-out' conformation (PDB 6I7K and 6I7L)[19]. To analyze the effect of TPR flexibility on FICD function, we artificially stabilised the BiP binding-competent 'TPR-in' conformation by mutating Asp160 and Thr183 (Fig. 1b(i)) to cysteines and oxidising the protein to form an intramolecular disulfide bond (TPRox, Supplementary Fig. 4c). TPR oxidation within a monomeric FICD$^{L258D-H363A}$ background resulted in more biphasic binding kinetics and a significant decrease in dissociation rate from BiP-AMP (Fig. 3b (i)). This suggested that the covalent fixation of the 'TPR-in' conformation outweighed any destabilising effects of perturbing the precise alignment of the intramolecular Fic-TPR domain contacts by mutagenesis. Indeed, oxidation of this protein resulted in marked stabilisation relative to the cysteine reduced form (TPRred), which also does not reversibly associate with BiP-AMP (Supplementary Fig. 4b and f). Notably, the effect of TPRox on dimeric FICD binding to BiP-AMP was less pronounced (Fig. 3b(ii)).

The differential effects of TPRox on monomeric and dimeric FICD is consistent with the 'TPR-out' conformation having been observed only in monomeric FICD structures[19] and suggests that

dimeric FICD has an intrinsically less flexible TPR domain. Nevertheless, 'TPR in' fixation (by oxidation) does alter dimeric FICD-binding kinetics: dissociation of dimeric FICD from BiP-AMP is accelerated by TPR domain oxidation. The difference between FICD$^{H363A}$(TPRox) and FICD$^{H363A}$ dissociation rates is further accentuated by the presence of ATP (Fig. 3b(ii) and Supplementary Fig. 4e). This difference is presumably a manifestation of the disruption of FICD's interdomain TPR-Fic contact— that results from the mutation of Asp160 and cannot be rescued by intramolecular disulfide bond formation (Fig. 3b(i)). This notion is further supported by the observation that TPRred behaves similarly to TPRox in context of the dimeric FICD (Supplementary Fig. 4f).

Consistent with the essential role played by the TPR domain in deAMPylation complex assembly, mutation or removal of the TPR1 motif significantly reduced the catalytic efficiency ($k_{cat}/K_M$) of BiP-AMP deAMPylation, by both the monomeric and dimeric enzymes in vitro (Fig. 3c, and Supplementary Fig. 5a–c). Interestingly, although fixing the 'TPR in' conformation by oxidation (TPRox) did not significantly diminish the affinity of FICD for AMPylated BiP (Fig. 3b and Supplementary Fig. 4f), it did compromise the deAMPylation activity of both monomeric and dimeric FICD (Fig. 3c, bottom). This effect on catalytic efficiency plausibly reflects a contribution of TPR domain flexibility and intra-FICD interdomain communication to deAMPylation turnover number ($k_{cat}$).

**FICD's TPR domain is responsible for the recognition of unmodified ATP-state BiP.** The importance of contacts between FICD's TPR domain and BiP to deAMPylation, demonstrated above, explains previous observations that the isolated AMPylated BiP SBD is refractory to FICD-mediated deAMPylation[10]. It is noteworthy that FICD also specifically binds[19] and AMPylates ATP-state BiP with a preference for more domain-docked BiP mutants and fails to AMPylate the isolated BiP SBD[3]. Furthermore, the observation that FICD's interaction with unmodified BiP:ATP was abrogated by TPR1 deletion (Fig. 3a) suggests that FICD recognises the ATP-state of unmodified BiP (for AMPylation) in a similar fashion to ATP-state biased BiP-AMP (for deAMPylation).

Structures of unmodified BiP indicate that a domain-undocked ADP-state BiP loses the tripartite NBD-linker-SBDβ surface that is recognised by FICD's TPR1 motif in the context of deAMPylation (Fig. 4a and Supplementary Movie 2). This engagement is reminiscent of the ATP state-specific interaction of J-domain proteins with Hsp70s (Supplementary Fig. 6a). Furthermore, even if FICD were able to bind the NBD or the $\ell_{7,8}$ SBDβ region (which also becomes less accessible in BiP's ADP-state) of a nucleotide-free (apo) or ADP-bound BiP, the Hsp70's heavy bias towards the domain-undocked conformation[5,32] would render engagement of the other FICD–BiP interaction surface unlikely (Fig. 4a and Supplementary Movie 2).

Consistent with this structural analysis (Fig. 4a), the isolated TPR domain of FICD was able to specifically bind the domain-docked ATP-state of unmodified BiP with a $K_D$ of 1.1 μM (Fig. 4b, c). Moreover, the lack of detectable interaction of the domain-undocked BiP:Apo suggests that the $K_D$ of FICD's TPR domain for BiP:Apo must be >160 μM. The fast-on and fast-off binding kinetics of the isolated TPR domain with ATP-state BiP are in keeping with its role as a substrate recognition domain of an enzyme evolved for efficient catalysis.

To further test the role of conserved TPR–BiP contacts in formation of a pre-AMPylation complex (consisting of FICD and BiP:ATP) we returned to the BLI setup of Fig. 3b, but with ATP-bound unmodified BiP immobilised as a ligand. In this context

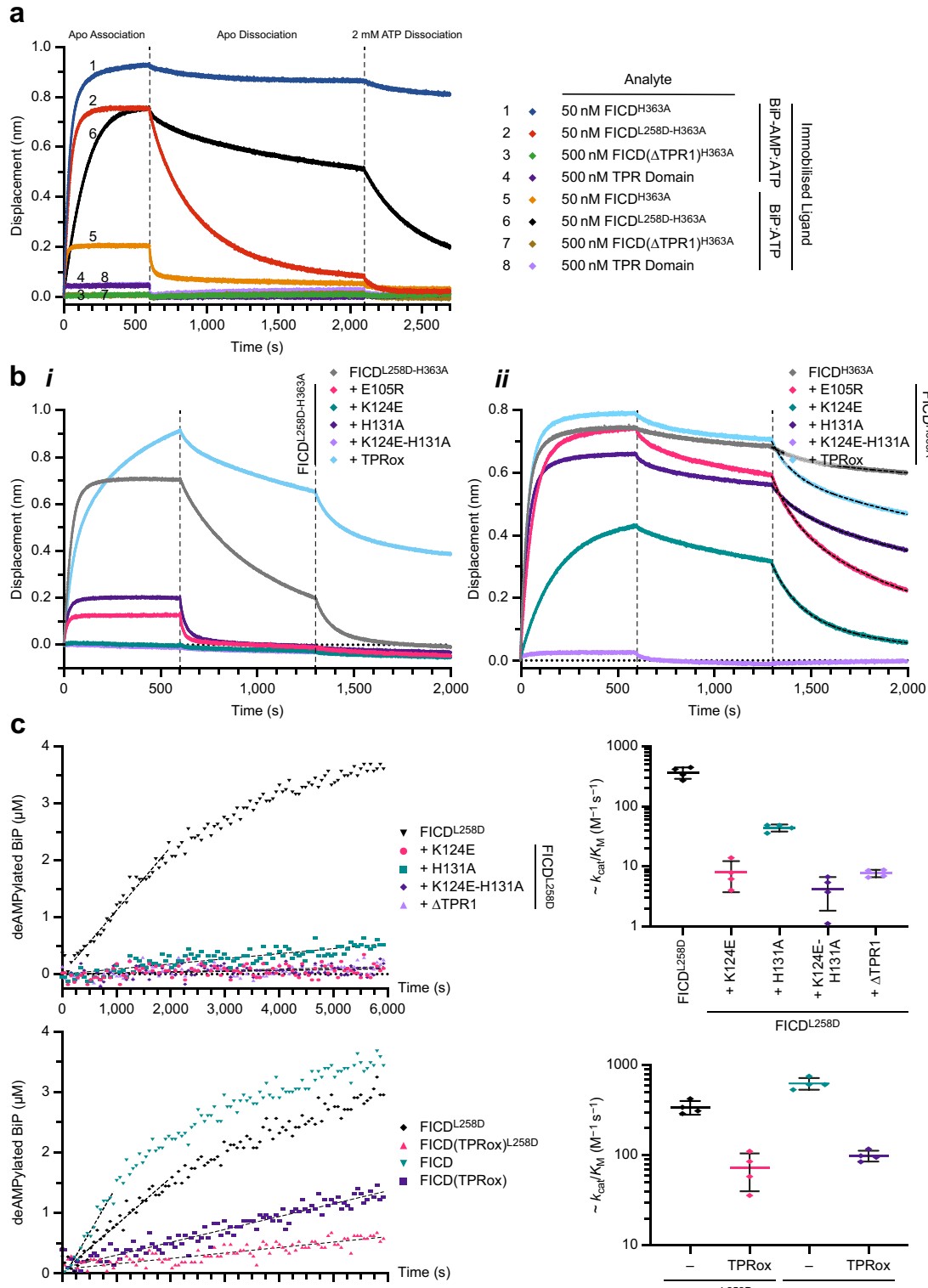

**Fig. 3 FICD's TPR domain is essential for BiP-AMP binding and deAMPylation. a** Representative BLI association–dissociation curves of FICD analytes from immobilised BiP (either AMPylated or unmodified) bound to ATP ($n = 3$ independent experiments). See Supplementary Fig. 4d. **b** Representative BLI analysis of TPR domain mutants of monomeric (**i**) and dimeric (**ii**) FICD binding to immobilised AMPylated BiP ($n = 3$ independent experiments). In both **a** and **b** the buffer used in the second dissociation step (see vertical dashed lines) was supplemented with 2 mM ATP. In **b**(**ii**) the second dissociation step traces (where applicable) are overlaid with the best-fit curves derived from a two-phase exponential decay model. See Supplementary Fig. 4e, f. **c** Analysis of the ability of different FICD variants to deAMPylate BiP. Left, Fluorescent polarisation-derived time courses of BiP-AMP(FAM) deAMPylation. Fits of the initial linear reaction phase are overlaid. Right, quantification of the approximate catalytic efficiencies of the different FICD variants. Mean values of approximate $k_{cat}/K_M$ values for each FICD variant ± standard deviation (SD), from $n = 4$ independent experiments, are shown. See Supplementary Fig. 5. For reasons of experimental expedience, the Glu105Arg FICD mutant was not incorporated into an enzymatically competent FICD background and was, therefore, neither tested in this in vitro deAMPylation assay nor in the vitro AMPylation assays shown later (Fig. 5a). Source data are provided as a Source Data file.

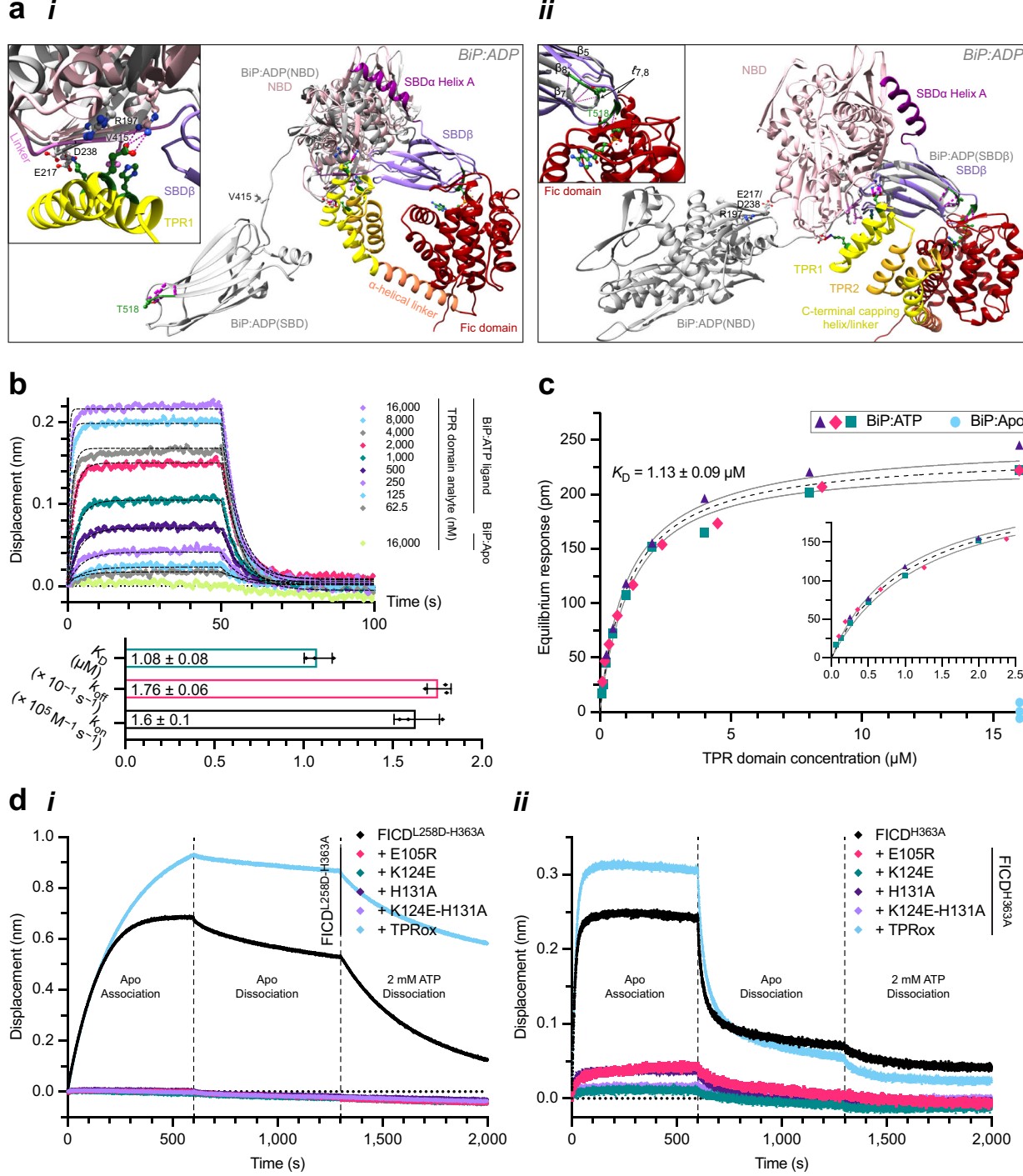

**Fig. 4 FICD's TPR domain is essential for the recognition of ATP-bound unmodified BiP. a** The deAMPylation complex (coloured as in Fig. 1a with selected BiP interaction partners labelled) is aligned via its NBD (**i**) or SBDβ (**ii**) with an ADP-state BiP (PDB 7A4U; grey)[72]. Inset (**i**), a closeup view of FICD(TPR1)-BiP(NBD) contacts. (**ii**) the intermolecular β-sheet region of $\ell_{7,8}$ (green) is shortened in BiP:ADP. Inset, disposition of Thr518 is highlighted (pink lines; hydrogen bonds). See Supplementary Movie 2 and Supplementary Fig. 6a. **b** FICD's isolated TPR domain specifically binds the ATP-state of BiP. A representative BLI experiment demonstrating the ability of FICD(TPR) to engage immobilised BiP:ATP but not BiP:Apo. A global fit analysis of a one-phase association–dissociation model (black dashed lines) is overlaid. Below are the resulting kinetic binding parameters (mean ± SD) of the interaction of FICD(TPR) and BiP:ATP, from $n = 3$ independent experiments. **c** Steady-state equilibrium binding response analysis of the representative TPR domain binding experiment shown in **b**. Results from the analyte dilution series from the three independent BiP:ATP binding experiments are represented by different symbols. The fit from a one site binding model is shown with 95% confidence bands (dashed black line and solid grey lines, respectively). The calculated $K_D$ is also annotated (mean ± SD). The inset panel highlights the same data and fitting over the lower analyte concentration range. Note, the lack of detectable steady-state binding of the TPR domain to BiP:Apo. **d** Representative BLI analysis of TPR domain mutants of monomeric (**i**) and dimeric (**ii**) FICD binding to immobilised ATP-bound BiP, from $n = 3$ independent experiments. See Supplementary Fig. 6b. Source data are provided as a Source Data file.

the effects of TPR1 motif mutations on FICD binding were magnified relative to their effect on the deAMPylation complex (Fig. 4d and Supplementary Fig. 6b). This is consistent with a greater contribution of FICD's TPR domain to the assembly of the pre-AMPylation complex compared to the deAMPylation complex (as only the latter benefits from the interaction interface between the covalently BiP-linked AMP moiety and FICD's active site). Loss of TPR–BiP contacts by surface mutations in TPR1 also impaired BiP AMPylation by monomeric FICD in vitro (Fig. 5a and Supplementary Fig. 6c), paralleling the effect of these mutations on deAMPylation (Fig. 3). Fixation of the TPRin conformation by TPR oxidation, although stabilising the pre-AMPylation complex of monomeric FICD and BiP:ATP (Fig. 4d and Supplementary Fig. 6b), nonetheless decreases the in vitro AMPylation rate (Fig. 5a). Taken together, the effects of TPRox suggest that TPR domain flexibility contributes to the $k_{cat}$ of both AMPylation and deAMPylation.

To examine the effect of the TPR surface mutations on BiP AMPylation in cells, we compared the ability of otherwise wild-type, hyperactive, monomeric FICD lacking the gatekeeper glutamate (FICD$^{E234G-L258D}$) and TPR mutant versions thereof to promote a pool of AMPylated BiP in cells lacking endogenous FICD. Levels of AMPylated BiP, detected by its mobility on native-PAGE, were significantly lower in cells targeted with the FICD$^{K124E-E234G-L258D}$ and FICD$^{K124E-H131A-E234G-L258D}$ TPR1 mutations (Fig. 5b). The higher levels of expression of the TPR1 mutant FICDs, compared to the parental FICD$^{E234G-L258D}$, is also consistent with previous observations of an inverse relationship between FICD variant expression level and AMPylation activity[19].

BiP inactivation, by deregulated AMPylation, increases ER stress[19]. This feature was exploited to quantify the functional effect of the TPR1 mutations in an orthogonal assay, based on the ER stress-responsive reporter XBP1::Turquoise, utilising flow cytometry (Fig. 5c and Supplementary Fig. 6d). In cells expressing the various TPR1 mutant FICD derivatives, reporter activity (analysed by its bimodal distribution) correlated well with the levels of AMPylated BiP detected by native-PAGE and with the hierarchy of the mutations' effects on BiP binding (Fig. 3b). The totality of these observations leads us to conclude that TPR surface mutations in residues that contact BiP in the deAMPylation complex also contribute to enzyme–substrate interaction during FICD-mediated AMPylation. Moreover, BiP's Thr518 can be readily modelled into the active site of a AMPylating monomeric FICD alongside its MgATP co-substrate, by alignment with the deAMPylation complex's Fic domain (Supplementary Fig. 7). This provides further support for a similar mode of FICD substrate engagement facilitating both of its mutually antagonistic enzymatic activities.

**The mechanism of eukaryotic deAMPylation**. The requirement for Glu234, His363 and the identification of AMP and unmodified BiP as products of the deAMPylation reaction[10,19] led us to previously propose a speculative model for the deAMPylation reaction mechanism[10]. This subject has now been re-visited in light of the functionally validated architecture of the high-resolution eukaryotic deAMPylation complex.

The state 1 deAMPylation complex crystal structure contains well-resolved electron density for BiP's AMPylated Thr518 residue within FICD's active site (Fig. 6a). The phosphate of Thr518-AMP is coordinated by a Mg$^{2+}$ held in position by Asp367 of FICD's Fic motif. A similarly positioned Mg$^{2+}$ coordinates the α and β phosphates of ATP in the AMPylation-competent enzyme[19] and in FICD:MgADP (Supplementary Fig. 2c). Glu234 (located atop the α$_{inh}$ helix) tightly engages a

water molecule located within FICD's LR-type anion-binding nest[33] (Fig. 6a and Supplementary Fig. 8). The latter (Fic motif) feature contributes towards the stabilisation of ATP's α and β phosphates in the AMPylating enzyme[18,24].

The aforementioned Glu234-coordinated water molecule sits almost directly in-line with the Pα-Oγ(Thr518) phosphodiester bond (Fig. 6b, Supplementary Fig. 8 and Supplementary Movie 3) and likely constitutes the hydrolytic water molecule. When also modelled with a catalytic histidine (from PDB 6I7K; 0.45 Å RMSD over 214 Cα pairs aligned over the Fic domain residues 213–426)[19] the structure is highly suggestive of an acido-basic hydrolytic mechanism: Glu234 aligns and activates a water molecule for an S$_N$2-type nucleophilic attack into the α-phosphate, with His363 positioned to facilitate the concerted protonation of the Thr518 alkoxide leaving group (generating unmodified BiP and AMP as products[10]).

**Increased Glu234 flexibility enfeebles monomeric FICD deAMPylation activity.** The deAMPylation complex active site presented in Fig. 6 explains the essential role of gatekeeper Glu234 in Fic domain-catalysed deAMPylation[10,12]. However, a second sub-2 Å deAMPylation complex-crystal structure (referred to as state 2), which is almost identical to that previously presented (Table 1, Supplementary Fig. 9a and Supplementary Movie 3), hints at an important detail. As in the state 1 structure (Fig. 6), the FICD active site contains clear electron densities for BiP's Thr518-AMP, Fic domain catalytic residues and a coordinated Mg$^{2+}$ cation (Supplementary Fig. 9b). However, alignment with the state 1 structure reveals a different orientation of Glu234 (Fig. 7a, Supplementary Fig. 9c and Supplementary Movie 3). In the state 2 structure Glu234's sidechain points away from the position of the catalytic water molecule, clearly visible in state 1, towards the Mg$^{2+}$.

The variability in Glu234 conformation, noted above, fits previous observations that FICD monomerisation increases Glu234 flexibility, disfavouring autoinhibition of AMPylation activity[19]. The reorientation of Glu234 noted in state 2 also perturbs the proposed deAMPylation mechanism by inducing a slight shift in the Mg$^{2+}$ octahedral coordination complex (Fig. 7a and Supplementary Movie 3). Although there is some remaining electron density in the region of the catalytic water molecule noted in state 1, this density is merged with that of a Mg$^{2+}$-coordinating water molecule. The elongated density is incompatible with the presence of two water molecules (as accommodating the Mg$^{2+}$-coordination geometry would necessitate an infeasible inter-water distance of 1.89 Å) and suggests that there may be a dynamic shuttling of a water to and from the primary Mg$^{2+}$-coordination sphere into a position conducive to catalysis. It is therefore clear that the Glu234 position observed in the state 2 crystal structure does not permit the stable positioning of a catalytic water molecule in-line for nucleophilic attack.

A corollary of the two tenets—that Glu234 is necessary for coordinating a catalytic water molecule for deAMPylation and that Glu234 flexibility increases upon monomerisation—is the prediction that FICD-mediated deAMPylation activity should decrease upon monomerisation. This has already been demonstrated in terms of a 46% decrease in catalytic efficiency (Fig. 3c and Supplementary Table 2)—the calculated $k_{cat}/K_M$ of FICD is 1.9-fold (±0.2-fold (SEM)) greater than that of FICD$^{L258D}$. Moreover, dimeric FICD's $k_{cat}/K_M$ is in good agreement with that derived from a previous Michaelis–Menten analysis of a GST-tagged FICD[10] (Supplementary Table 2).

However, an increase in Glu234 flexibility is also expected to intrinsically affect deAMPylation catalysis and thus lower the $k_{cat}$. In order to directly measure the turnover number for monomeric

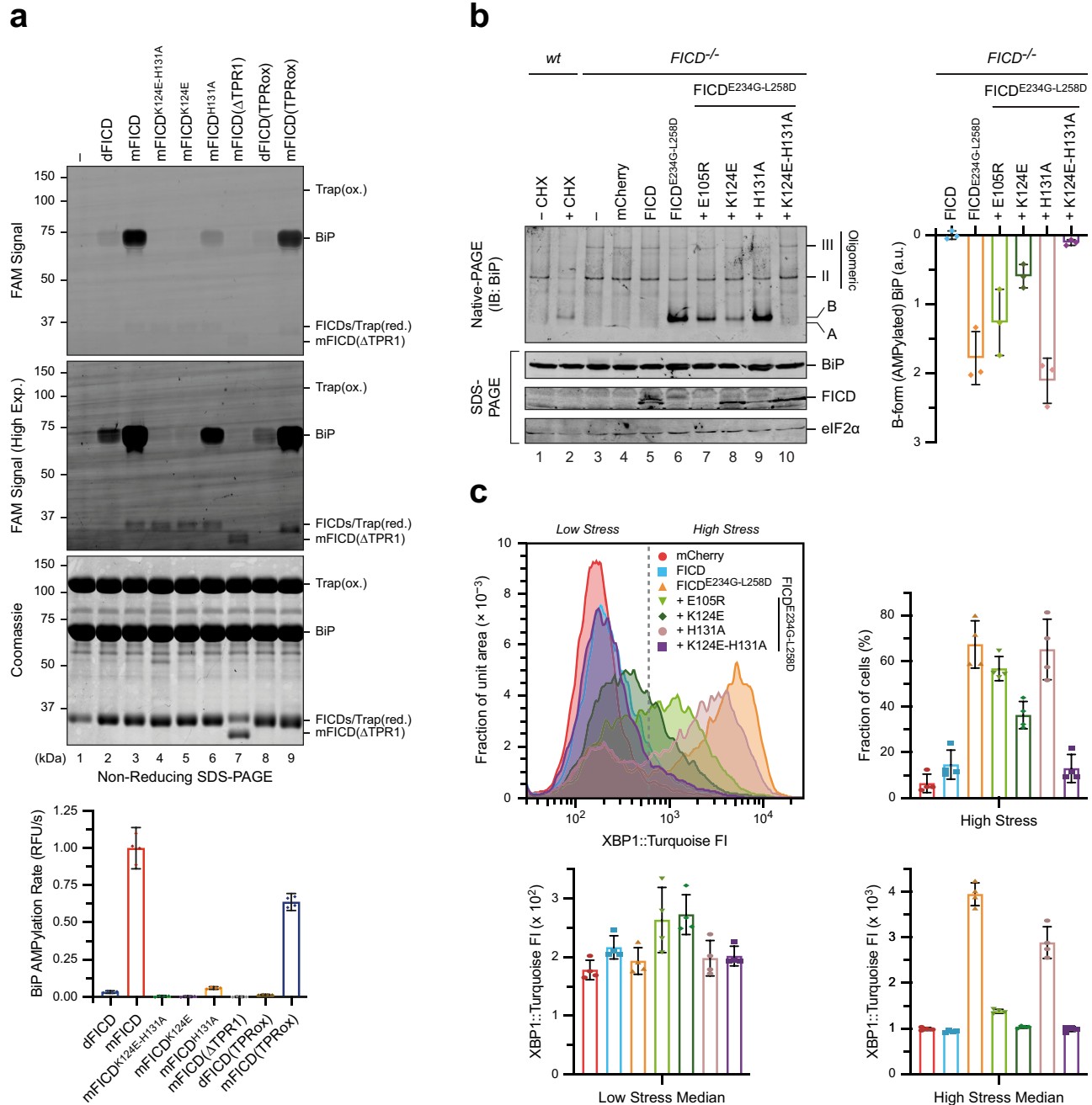

**Fig. 5 FICD's TPR domain is essential for AMPylation of ATP-bound BiP. a** Fluorescence and Coomassie gel-images of an in vitro AMPylation assay, utilising ATP(FAM) as the AMPylation co-substrate, in the presence of excess product trap (Trap(ox), to discourage BiP-AMP(FAM) deAMPylation[10]). dFICD, dimeric FICD; mFICD, monomeric FICD[L258D]. Gels from a representative experiment are shown with the initial rates (mean ± 95% confidence interval (CI)) of BiP-AMPylation (in relative fluorescent units/s), normalised to the rate of mFICD-mediated BiP-AMPylation, from $n = 4$ independent experiments. Note, the lack of correlation between FICD (*cis*)auto-AMPylation and BiP substrate AMPylation. See Supplementary Fig. 6c. **b** Native-PAGE immunoblot analysis of the accumulation of AMPylated (B-form) BiP in CHO cells lacking endogenous FICD transfected with FICD variants, as indicated. Major, non-AMPylated BiP species (A, II and III) are noted. Right, quantification of AMPylated B-form BiP from $n = 3$ independent experiments (mean ± SD). **c** Histograms of the FACS signal of an XBP1::Turquoise UPR reporter in *FICD*$^{-/-}$ CHO cells expressing the indicated FICD derivatives. Note the bimodal distribution of the fluorescent signal in FICD-transfected cells. Quantification of the fraction of cells that are stressed, as well as the median FACS signal of the low and high stressed cell populations are shown from $n = 4$ independent experiments (mean values ± SD). Bars and datapoints are (colour-) coded according to the histogram legend. Source data are provided as a Source Data file.

and dimeric FICD both enzymes must be saturated with deAMPylation substrate. It was found that the initial rates of deAMPylation were indistinguishable at substrate concentrations of 100 and 150 μM BiP-AMP (Fig. 7b and Supplementary Fig. 9d, e), implying that FICD and FICD[L258D] are saturated by BiP-AMP

under the given experimental conditions. Therefore, at these substrate concentrations the initial deAMPylation rates represent maximal enzyme velocities, from which a $k_{cat}$ parameter can be extracted (Fig. 7c). As expected for the less-flexible Glu234-bearing dimeric FICD, its deAMPylation $k_{cat}$ was significantly

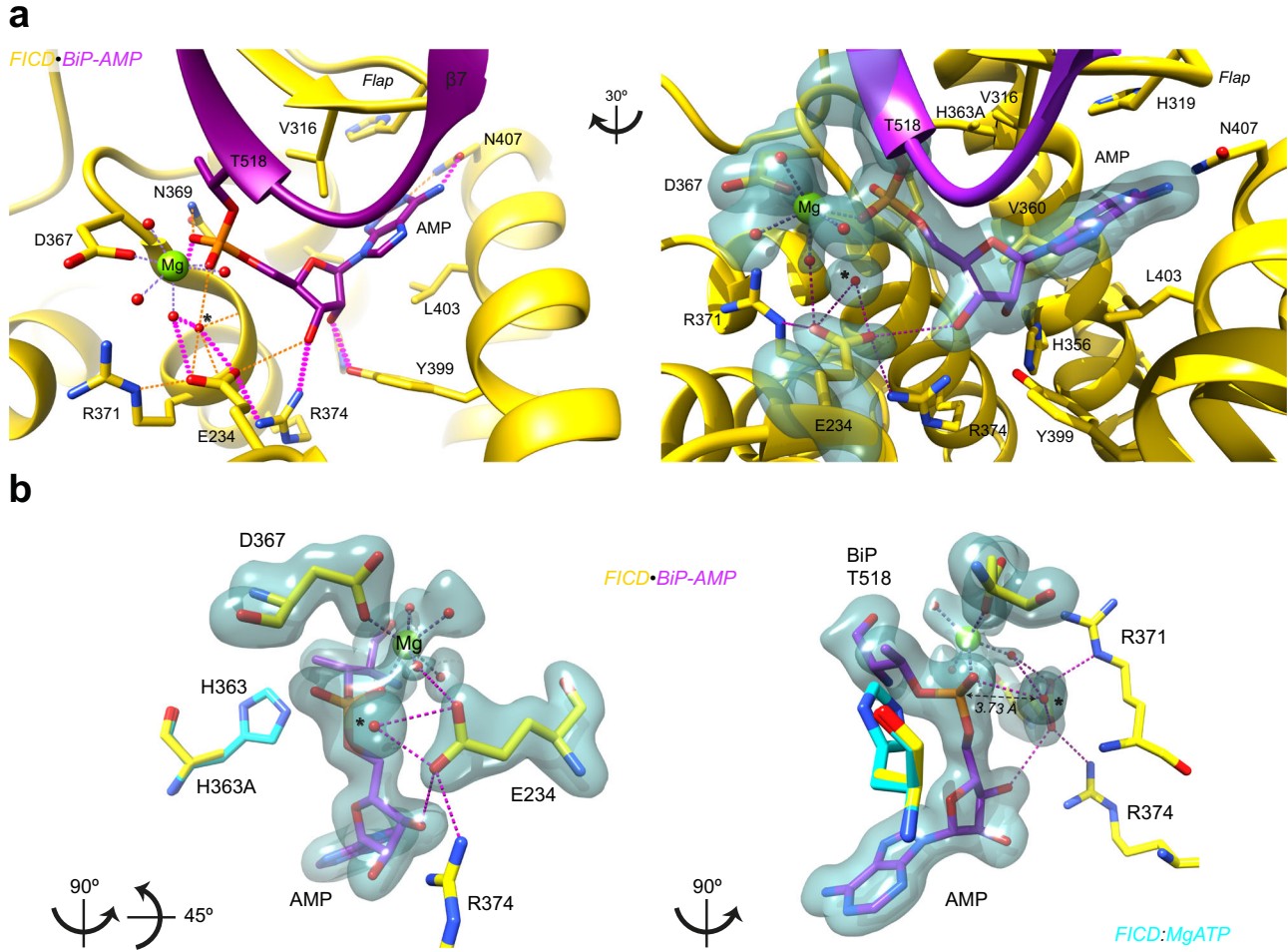

**Fig. 6 The enzymatic mechanism of eukaryotic deAMPylation. a** BiP's Thr518-AMP (purple) bound to FICD (yellow). Left, the arrangement of BiP's AMPylated Thr518 and $Mg^{2+}$ cation within the Fic domain active site. Residues interacting with $Mg^{2+}$ and the AMP moiety are shown as sticks and annotated. Hydrogen-bonds involving the AMP moiety and FICD's Glu234 sidechain are shown (with high confidence hydrogen bonds depicted with thick, pink dashed lines and those only meeting relaxed hydrogen bond constraints depicted with orange dashed lines). Note, the putative catalytic water molecule* forms hydrogen bonds to Glu234 (located at the top of $\alpha_{inh}$) and potential hydrogen bonds to the backbone NH groups of the Fic domain anion-binding nest ($G^{368}NG^{370}$) and Arg371 (see Supplementary Fig. 8). Right, a slightly rotated view of the FICD-active site, shown on the left, overlaid with an unbiased polder OMIT electron density map, contoured at 4σ. For clarity only hydrogen bonds formed by Glu234 are shown (pink dashed lines). **b** As in the right-hand side of **a** but reduced to highlight Glu234's coordination of the catalytic water molecule* and its position in-line for nucleophilic attack into the α-phosphate. Additionally, the putative general acid, His363, is modelled based on an alignment of FICD:MgATP (PDB 6I7K, turquoise)[19]. See Supplementary Movie 3.

greater (by a factor of $1.8 \pm 0.2$) than that of monomeric $FICD^{L258D}$ (Fig. 7c and Supplementary Table 2).

Together, the comparison of dimeric and monomeric FICD deAMPylation catalytic efficiencies and turnover numbers suggests that both oligomeric states of FICD possess very similar $K_M$ values for BiP-AMP (16–17 μM, see Supplementary Table 2). Thus, the increased deAMPylation $k_{cat}$ of dimeric FICD is compensated for by the increased affinity of dimeric FICD for BiP-AMP (see Fig. 3a). Note, the $k_{cat}$ and $K_M$ values derived for dimeric FICD are in good agreement with those previously obtained from Michaelis–Menten analysis of GST-FICD (Supplementary Table 2) adding credibility to the method of $k_{cat}/K_M$ and $k_{cat}$ determinations presented here.

## Discussion
Here, we have leveraged insights from crystal structures of a deAMPylation complex of FICD and BiP-AMP to gain a detailed understanding of eukaryotic deAMPylation and a broad understanding of the enzyme–substrate interactions of FICD that underpin its mutually antagonistic activities of BiP AMPylation and deAMPylation. Biochemical and cellular studies of structure-guided mutations in FICD have shed light on both substrate- and enzyme-level regulation of BiP's AMPylation cycle as it matches BiP activity to ER stress in a post-translational strand of the UPR (Fig. 8 and Supplementary Fig. 10).

The specific recognition of ATP-state BiP is mediated by an interaction of FICD's TPR1 domain with a tripartite ATP state-specific surface composed of BiP's NBD, linker and SBDβ. Moreover, the TPR domain of FICD is only able to direct BiP's $\ell_{7,8}$ SBDβ region into the Fic domain active site when BiP's NBD and SBD are closely opposed, as in the domain-docked ATP-state. These features explain the finding that the client protein-bound ADP-state BiP is not a substrate for AMPylation[3] and suggests a facile mechanism for substrate-level regulation of BiP AMPylation—in which substate availability is inversely proportional to the unfolded protein load in the ER.

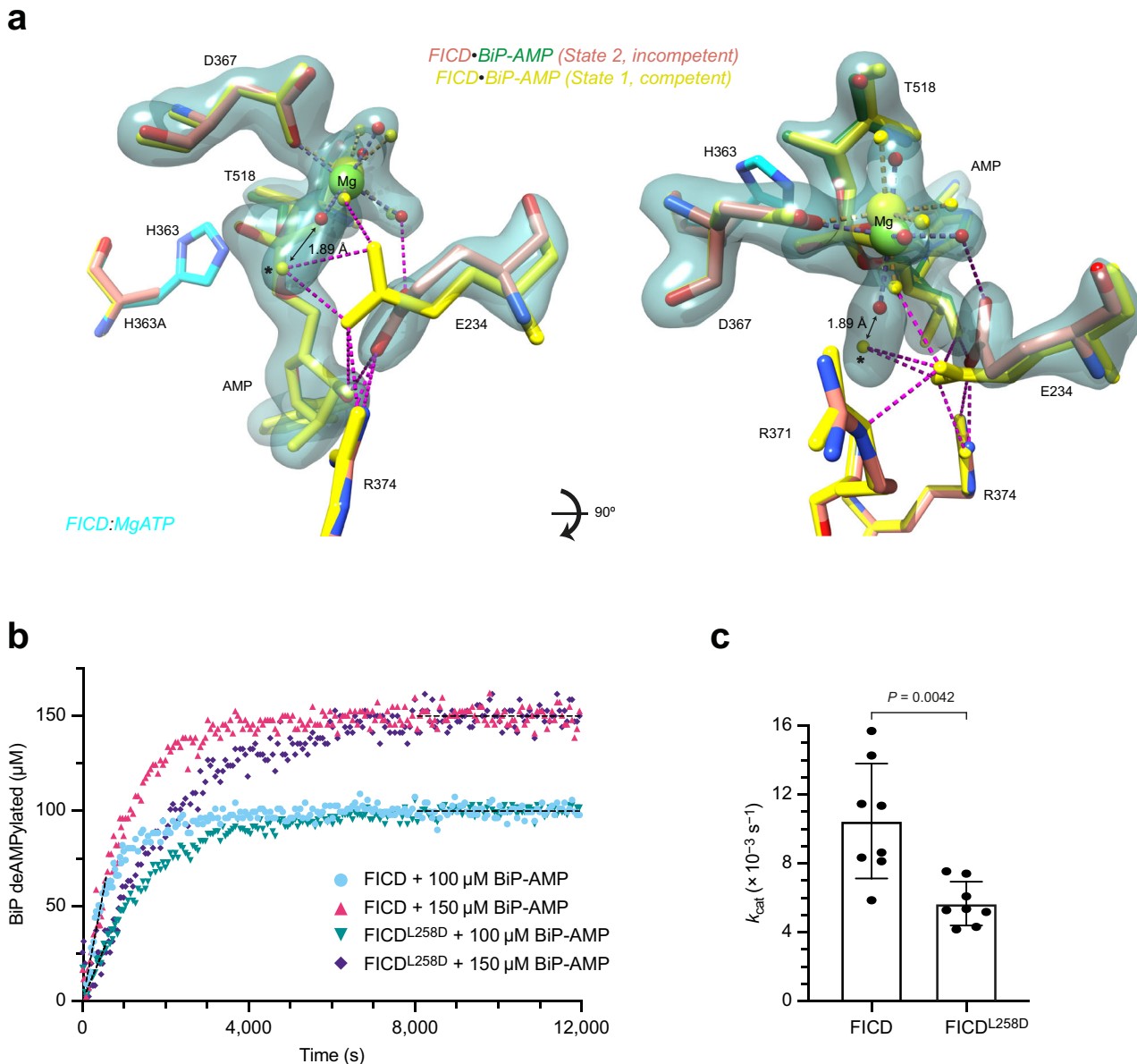

**Fig. 7 Monomerisation increases the likelihood of a non-deAMPylation competent Glu234 conformation. a** An unbiased polder OMIT electron density map from a second deAMPylation complex structure (state 2), contoured at 6σ, covering selected Fic domain catalytic residues (orange), the Mg$^{2+}$-coordination complex and BiP's Thr518-AMP (green). The reduced (state 2) active site is aligned with the active site of the (deAMPylation competent) state 1 complex (yellow). His363 is modelled from an alignment of catalytically competent FICD (PDB 6I7K, as in Fig. 6b)[19]. Residues interacting with the AMP moiety are shown as sticks and the catalytic water (from state 1) is annotated with *. The distance between the Mg$^{2+}$ first-coordination sphere water (red, state 2) and the (state 1) catalytic water* is annotated. Hydrogen bonds formed by Glu234 are shown as pink dashed lines. See Supplementary Movie 3 and Supplementary Fig. 9a–c. **b** A representative BiP-deAMPylation time course with 10 μM FICD or FICD$^{L258D}$, demonstrating that 100 and 150 μM BiP-AMP both represent saturating concentrations of deAMPylation substrate. See Supplementary Fig. 9d, e. **c** The derived $k_{cat}$ parameters, from $n = 4$ independent experiments each with two saturating concentrations of BiP-AMP (as in **b**). The mean ± SD is shown with the $P$-value from a two-tailed Welch's $t$-test annotated. Source data are provided as a Source Data file.

A reciprocal mechanism for substrate-level regulation of deAMPylation is unlikely, as AMPylated BiP is intrinsically biased towards the ATP-like domain-docked state[4]. Biochemical and cell-based experiments, pointing to similar modes of BiP engagement in FICD-mediated AMPylation and deAMPylation, thus suggest that regulatory changes in FICD's active site must contribute to the enzyme's ability to respond to changes in the burden of ER unfolded proteins. Previous studies uncovered a role for a monomerisation-induced increase in Glu234 flexibility, which permits AMPylation competent binding of MgATP within the FICD-active site[19]. However, the basis for the relationship

between oligomeric state and deAMPylation activity remained obscure, awaiting clarification of the enzymatic mechanism and the essential role played by Glu234 in FICD-mediated deAMPylation.

Whilst this manuscript was in revision we became aware of a crystal structure of BiP covalently bound to FICD (PDB 6ZMD)[34], likely that of a transient intermediate post-AMPylation state. This covalent complex possess a near-identical FICD (TPR)–BiP interface to the deAMPylation complexes presented here, which accounts for the consistent outcome of the muta-genesis carried out in both studies. However, due to mutations

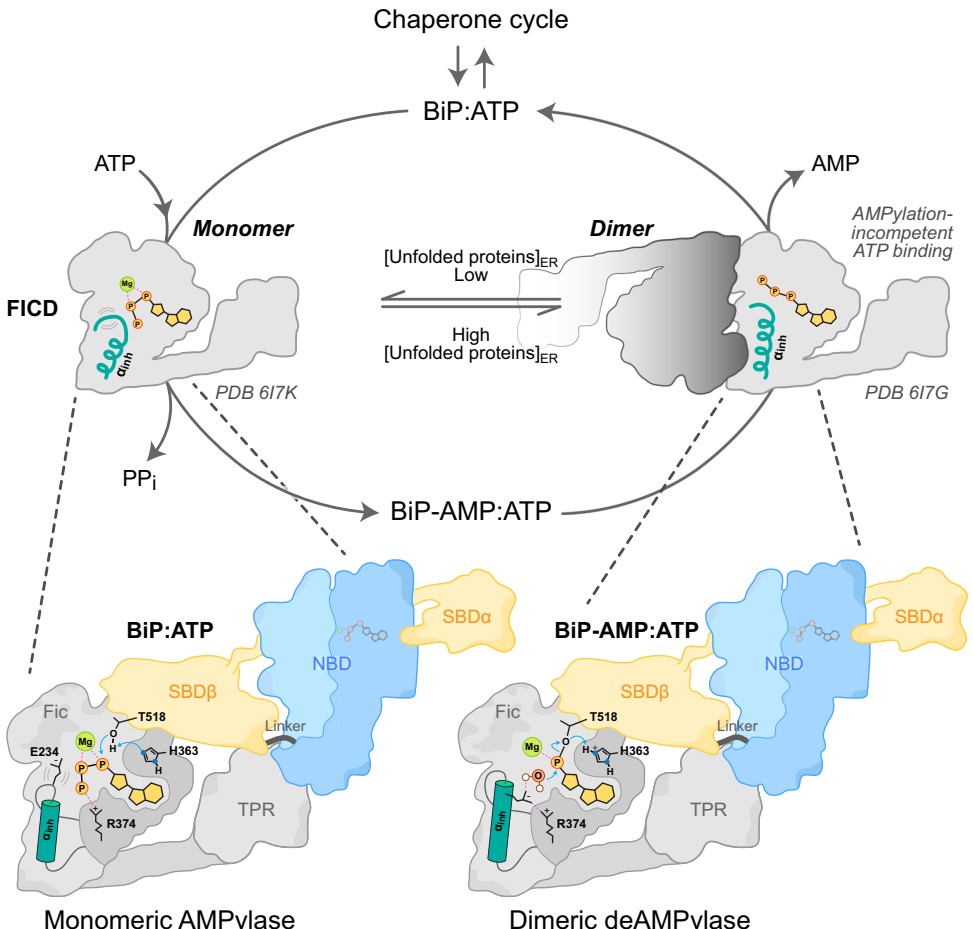

**Fig. 8 Model of FICD AMPylation and deAMPylation of BiP.** FICD's TPR domain and catalytic Fic domain recognise, respectively, the linker-docked NBD and the $\ell_{7,8}$ region of the SBDβ of either AMPylated or unmodified BiP. Simultaneous engagement of both interfaces is only possible when BiP is in a domain-docked ATP-like state. Dimeric FICD has a relatively rigid gatekeeper Glu234 that facilitates efficient alignment of an attacking water for BiP deAMPylation and inhibits AMPylation competent binding of ATP. Enhanced flexibility of monomeric FICD's Glu234 decreases deAMPylation efficiency whilst permitting AMPylation competent binding of MgATP by monomeric FICD. We speculate that the FICD monomer-dimer equilibrium is adjusted in response to changing levels of unfolded proteins within the ER by processes which may include a direct response to changes in the ER energy status (ATP/ ADP ratio)[19]. Further details of the FICD-catalysed deAMPylation reaction are presented in Supplementary Fig. 10.

introduced into the FICD in question (FICD$^{L258D-E234G-E404C}$) and the nature of the covalent capture strategy employed (which corrupts the interface of the Fic domain and BiP(SBDβ)), the covalently linked FICD–BiP complex provides no details pertaining to FICD's enzymatic mechanism (Supplementary Fig. 11).

The crystal structures presented in this work provide strong support for a mechanism of eukaryotic deAMPylation that is acido-basic in nature and in which Glu234 aligns a catalytic water molecule in-line for an $S_N2$-type nucleophilic attack into α-phosphate of Thr518-AMP (Supplementary Fig. 10). Glu234, may act as a catalytic (but not a general) base through a mechanism involving late proton transfer analogous to the role played by the catalytic aspartates of some protein kinases[35,36]. This proposed deAMPylation mechanism (which also rationalises the essential role for a divalent cation and His363) is far removed from the binuclear metal-catalysed reactions catalysed by the other two known (bacterial) deAMPylases[20,23]. Moreover, other mechanisms of phosphodiester bond cleavage, including anchimeric assistance or an E1cB-type elimination reaction, which are capable of generating the products of FICD-mediated deAMPylation (AMP and unmodified BiP), are rendered very unlikely by the

structure of the deAMPylation complex (Supplementary Fig. 8 and Fig. 6a).

As a bacterial Fic protein (EfFic) has also been observed to possess gatekeeper glutamate-dependent deAMPylation activity[12], it is likely that the mechanism of deAMPylation outlined above is conserved across this class of proteins. This conclusion, pertaining to the immediate role of Glu234 in enabling BiP-AMP hydrolysis, permits various inferences to be made about the role of monomerisation and increased Glu234 flexibility in the regulation of deAMPylation activity. These, are supported by the direct observation of a monomeric FICD–deAMPylation complex with an alternative Glu234 conformation, resulting in a (state 2) deAMPylation non-competent active site that lacks a stably coordinated catalytic water molecule. Thus, increased Glu234 flexibility, induced by FICD monomerisation, not only considerably increases AMPylation activity[19] (Fig. 5a) but also decreases the deAMPylation $k_{cat}$ (Fig. 7c).

Oligomeric-state changes in the disposition of the gatekeeper Glu234 may not be the only mechanism for enzyme-based regulation of the BiP AMPylation–deAMPylation cycle. Observations that monomeric FICD binds more tightly to unmodified BiP

than BiP-AMP and the converse being true for dimeric FICD[19] (Fig. 3a), remain unexplained by the structure of the FICD deAMPylation complex. Modelling of the AMPylation complex active site does implicate oligomeric state-linked changes in Glu234 flexibility in directly contributing to the observed differential substrate-binding affinities (Supplementary Fig. 7b). However, it is entirely possible that other as yet unidentified differences in the interactions between FICD and BiP are mediated by changes in FICD oligomeric state or BiP modification status or by FICD protein dynamics. Indeed, a role for the latter is hinted at by the crystallographic and SANS-based evidence for TPR domain flexibility and by the effects of TPR fixation on enzyme–substrate complex formation and catalysis.

These caveats notwithstanding, this study advances our mechanistic understanding of the reciprocal-regulation of enzymatic activity afforded by FICD's oligomerisation-state-dependent switch (Fig. 8). This leaves unanswered the question of if and how the FICD monomer–dimer equilibrium responds to changing conditions in the ER. There is some evidence that FICD may respond to the energy-status of the ER, as a proxy for ER stress[19]. Given that Hsp70 proteins can directly modulate the oligomeric status (and thus activity) of their own regulators within the ER[37] and cytosol/nucleus[38] (BiP/Ire1α and Hsc70/Hsf1, respectively), the possibility of an additional layer of BiP-driven FICD-regulation is therefore an intriguing one to consider.

## Methods

**Plasmid construction**. The plasmids used in this study have been described previously or were generated by quick change polymerase chain reactions [annotated as 'QC' primers in Supplementary Table 4], Q5 site-directed mutagenesis (NEB) using Q5 Hot Start High-Fidelity DNA Polymerase (NEB) according to the manufacturer's protocol, or restriction digestion and ligation into existing plasmids. Full lists of the utilised plasmids and primers are provided in Supplementary Tables 3 and 4, respectively.

**Protein purification**. All proteins were purified using the method for FICD protein expression detailed in ref. [19], with only minor modifications. In brief, proteins were expressed as N-terminal His$_6$-Smt3 fusion constructs from either pET28-b vectors (expressed in T7 Express lysY/I$^q$ (NEB) *Escherichia coli* (*E. coli*) cells), or pQE30 vectors (expressed in M15 *E. coli* cells (Qiagen)). T7 Express cells were grown in LB medium containing 50 μg/ml kanamycin. M15 cells were grown in the same medium supplemented with an additional 100 μg/ml ampicillin. All cells were grown at 37 °C to an optical density (OD$_{600nm}$) of 0.6 and then shifted to 18 °C for 20 min, followed by induction of protein expression with 0.5 mM isopropylthio β-D-1-galactopyranoside (IPTG). Cells were harvested by centrifugation after a further 16 h at 18 °C.

Only the predicted structured regions of human FICD were expressed (residues 104–445). For 'full-length' BiP constructs, that is to say constructs containing the complete structured region of the SBDα lid subdomain, residues 27–635 of Chinese hamster BiP were expressed. This excludes an unstructured acidic N-terminal region and the C-terminal unstructured region bearing the KDEL. Note, in the recombinantly expressed residue range hamster and human BiP are identical in terms of amino acid identity. For use as an immobilised BLI ligand full-length BiP was expressed with an avi-tag inserted C-terminal to Smt3 and N-terminal to a GS linker and hamster BiP residues 27–635.

All BiP constructs used in this study were made ATPase[39] and substrate-binding[40] deficient via introduction of Thr229Ala and Val461Phe mutations, respectively. Thr229Ala allows BiP to bind and domain-dock in response to MgATP, even when immobilised via an N-terminal biotinylated Avi-tag[19]. The lack of ATP hydrolysis enables BiP to remain bound to ATP in its domain-docked state for prolonged periods of time, a feature which favours binding to[19] and AMPylation by FICD[3]. Both Thr229Ala (in the presence of ATP) and Val461Phe (independent of nucleotide) disfavour the binding of proteins within BiP's SBD (which principally occurs in the apo or ADP-state).

Following harvesting and lysis of the bacterial pellets, proteins were purified through the use of Ni-NTA agarose (Thermo Fisher), on-bead Ulp1 cleavage, anion exchange and gel filtration chromatography (based on the protein purification method within[19]). All purification was conducted at 4 °C. Unless otherwise specified (below) anion exchanges were conducted using a RESOURCE Q 6 ml column (GE Healthcare) with a linear gradient ranging from 95% AEX-A (25 mM Tris–HCl pH 8.0) and 5% AEX-B (25 mM Tris–HCl, 1 M NaCl) to 50% AEX-A and 50% AEX-B (see Supplementary Table 3). Gel filtration was conducted, depending on protein size and amount, on either a HiLoad 16/60 Superdex 75 or 200 prep grade column or a S200 or S75 Increase 10/300 GL column (see

Supplementary Table 3). All proteins were purified to homogeneity and >95% purity, as assessed by Coomassie-stained SDS–PAGE. Unless the protein was deliberately oxidised they were supplemented after gel filtration with 1 mM tris(2-carboxyethyl)phosphine (TCEP). Proteins were concentrated to >150 μM using centrifugal filters (Amicon Ultra; Merck Millipore), aliquoted and snap-frozen and stored at −80 °C. All protein concentrations were calculated using $A_{280}$, measured on a NanoDrop One (Thermo Fisher), and the protein's predicted extinction coefficient at 280 nm ($\varepsilon_{280}$).

*Preparative BiP AMPylation*. In the case of preparative scale AMPylation of BiP, this was achieved post-Ulp1 cleavage by addition of 10 mM MgCl$_2$, 5 mM ATP and 1/50 (w/w) GST-TEV-FICD$^{E234G}$ (UK1479[19]). The AMPylation reaction was incubated for 16 h at 25 °C. GST-TEV-FICD was then depleted by a 1 h incubation with GSH-Sepharose 4B matrix (GE Healthcare). AMPylation was confirmed as being stoichiometric by intact-protein mass spectrometry (LC-ESI-MS)[4].

*Forming disulfide-linked FICD dimers*. Disulfide-linked FICD dimers ($_{s–s}$FICD$^{A252C-H363A-C421S}$; UK2269)[19], used as a BiP-AMP trap for in vitro AMPylation assays, were purified as above with modifications. In brief, after the affinity chromatography step and the on-column Upl1-StrepII cleavage, the retained cleavage products were washed off the beads with TN-Iz10 (25 mM Tris–HCl pH 8.0, 150 mM NaCl, 10 mM imidazole) in the absence of reducing agent. The pooled eluate was concentrated and diluted 1:4 with TN-Iz10 (to further reduce the TCEP concentration). To allow for efficient disulfide bond formation the samples were supplemented with 20 mM oxidised glutathione and for 16 h at 4 °C. Afterwards, the protein solutions were diluted 1:2 with 25 mM Tris–HCl pH 8.0 and further purified by anion exchange and size-exclusion chromatography, as above. The final $_{s–s}$FICD$^{A252C-H363A-C421S}$ preparations were analysed by non-reducing SDS–PAGE to confirm quantitative formation of covalently linked dimers (>95%).

*In vitro biotinylation of BiP*. In vitro biotinylation of N-terminally avi-tagged BiPs was conducted on the expression tag-cleaved forms of unmodified or AMPylated BiP$^{T229A-V461F}$ residues 27–635 (UK2359). Biotinylation was conducted with 100 μM target protein, 200 μM biotin (Sigma) and 2 μM GST-BirA (UK1801) in a buffer of 2 mM ATP, 5 mM MgCl$_2$, 25 mM Tris–HCl pH 8.0, 150 mM NaCl and 1 mM TCEP. The reaction mixture was incubated for 16 h at 4 °C. The protein was made nucleotide-free by the addition of 2 U calf intestinal alkaline phosphatase (NEB) per mg of BiP, plus extensive dialysis into TN buffer supplemented with 1 mM DTT and 2 mM EDTA. The protein was then incubated with 0.5 ml GSH-Sepharose 4B matrix, for 1 h at 4 °C, to deplete the GST-BirA. The biotinylated BiP-containing supernatant was diluted 1:1 with AEX-A and loaded onto a MonoQ 5/50 GL column (GE Healthcare), equilibrated in 92.5% AEX-A and 7.5% AEX-B. BiP protein was eluted using a linear gradient of 7.5–50% AEX-B, over 20 CV at a flow rate of 1 ml/min. The Mono Q eluted protein fractions were supplemented with TCEP, diluted with glycerol and stored at −20 °C in a final buffer of TNTG (12.5 mM Tris–HCl pH 8.0, ~150 mM NaCl, 0.5 mM TCEP and 50% (v/v) glycerol) at a concentration >1 μM. Protein samples were validated as being nucleotide-free (apo) by their $A_{260/280}$ ratio and reference to IP-RP-HPLC analysis[10]. Proteins were confirmed as being >95% biotinylated via a streptavidin gel-shift assay.

*FICD TPR domain oxidation*. Purification of TPR domain oxidised (TPRox) FICD$^{D160C-T183C-C421S}$-derivative proteins was achieved as above (for other FICDs), with the addition of an oxidation and clean-up AEX step. Note, the cysteine free FICD$^{C421S}$ mutation was previously observed to have no effect on FICD-mediated deAMPylation or BiP-AMP binding and a slight stimulatory effect on FICD-mediated AMPylation[19].

In order to form the disulfide bond, the FICD protein (post-Ulp1 cleavage and Ni-NTA column elution) was diluted down to a concentration of 5 μM in a final buffer of 25 mM Tris–HCl pH 8.0 and 100 mM NaCl, supplemented with 0.5 mM CuSO$_4$ and 1.75 mM 1,10-phenanthroline (Sigma), and incubated for 16 h at 4 °C. The oxidation reaction was then quenched by the addition of 2 mM EDTA. The protein solution, diluted with 25 mM Tris pH 8.0 to a final NaCl concentration of 50 mM, was then purified on a HiTrap 5 ml Capto Q column (equilibrated in 95% AEX-A and 5% AEX-B buffer) using a linear gradient of 5–50% AEX-B over 10 column volumes. Proteinaceous fractions were further purified as detailed above (beginning with RESOURCE Q column purification), culminating in the purification of dimeric or monomeric FICD (as appropriate) by gel filtration.

Stoichiometric disulfide bond formation was confirmed by the use of an electrophoretic mobility assay (see Supplementary Fig. 4c), in which the putatively oxidised protein was heated for 10 min at 70 °C in SDS–Laemmli buffer ± DTT; all available thiols were then reacted with a large excess of PEG 2000 maleimide (30 min at 25 °C). All unreacted maleimides were then quenched by the addition of a molar excess of DTT before samples were analysed by SDS–PAGE. Significant PEG modification of FICD(TPRox) proteins was only observed in samples first denatured in reducing conditions (+DTT), suggesting that the two TPR domain-cysteines were not accessible for alkylation in the absence of DTT (on account of being oxidised to form an intramolecular disulphide bond).

**Protein crystallisation and structure determination**. Monomeric FICD$^{L258D-H363A}$ (residues 104–445) [UK2093] and monomeric lid-truncated BiP$^{T229A-V461F}$-AMP (residues 27–549) [UK2090] were purified as above and gel filtered into a final buffer of T(10)NT (10 mM Tris–HCl pH 8.0, 150 mM NaCl and 1 mM TCEP). As outlined in the text, FICD's His363Ala mutation facilitates a stable trapping of its deAMPylation substrate. As mentioned above, BiP$^{T229A-V461F}$ favours its monomeric ATP-state, in which it is less likely to bind substrates in its SBD and to form BiP oligomers. The removal of all but helix A of the SBDα (BiP residues 27–549) was also implemented to reduce the affinity of BiP substrate binding and oligomerisation and to increase the likelihood of crystallisation and high-resolution diffraction by removal of the flexible SBDα helix B, which in other Hsp70s has been documented to only transiently interact with the NBD in the ATP-state[31]. Heterodimer copurification was achieved by mixing FICD$^{L258D-H363A}$ and BiP$^{T229A-V461F}$-AMP in a 1.5:1 molar ratio, supplemented with an additional 250 μM ATP, 50 mM KCl and 2 mM MgCl$_2$. The mixture was incubated for 10 min at 4 °C and purified by gel filtration on an S200 Increase 10/300 GL column equilibrated in TNKMT buffer (10 mM Tris–HCl pH 8.0, 100 mM NaCl, 50 mM KCl, 2 mM MgCl$_2$ and 1 mM TCEP) with ≤5 mg of protein injected per SEC run. Heterodimeric protein fractions were pooled (as indicated in Supplementary Fig. 1a) and concentrated to 10.3 mg/ml using a 50 kDa MWCO centrifugal filter.

Crystallisation solutions, consisting of 100 nl protein solution and 100 nl crystallisation reservoir solution, were dispensed using a mosquito crystal (SPT Labtech) and the complex was crystallised via sitting drop vapour diffusion at 25 °C. State 1 crystals were obtained from reservoir conditions of 0.1 M MES pH 6.5, 10% PEG 4000 and 0.2 M NaCl; state 2 crystals were obtained from conditions of 0.1 M Tris pH 8.0 and 25% PEG 400. Crystals were cryoprotected in a solution consisting of 25% glycerol and 75% of the respective reservoir solution (v/v).

Diffraction data were collected from the Diamond Light Source at 100 K (beamline I04-1) utilising the Generic Data Acquisition (GDA) software (v9.2, Diamond Light Source). The crystallography datasets were indexed, integrated and scaled using xia2 software[41] through either the DIALS[42] (state 1 crystal) or XDS[43] (state 2 crystal) processing pipelines. The resulting unmerged data was then further processed by Pointless (for space group determination) and Aimless (for scaling and merging), both part of the CCP4 module Aimless (CCP4i2 [v1.0.2])[44,45]. Structures were solved by molecular replacement using the CCP4 module Phaser[44,46]. AMPylated BiP (PDB 5O4P)[4] and monomeric FICD (PDB 6I7L)[19] structures from the Protein Data Bank were used as initial search models. Manual model building was carried out in COOT[47] and refined using refmac5[48] with TLS added. Metal binding sites were validated using the CheckMyMetal server[49]. Zero Ramachandran outliers were present in either crystal structure with 98.37% and 98.49% of residues falling within Ramachandran favoured regions (for the state 1 and state 2 complex, respectively). The respective MolProbity scores were 0.81 (100$^{th}$ percentile score) and 1.04 (100$^{th}$ percentile score).

Polder (OMIT) maps were generated using the Polder Map module of Phenix[50,51]. Structural figures were prepared using UCSF Chimera[52] and PDB structures with evolutionary conservation score depictions were taken from the ConSurf Database[53] (https://consurfdb.tau.ac.il/), estimates of interaction surface areas were derived from PISA[54], interaction maps (Supplementary Figs. 1d and 8) were based on an initial output from LigPlot+[55] and the chemical reaction pathway (Supplementary Fig. 10) was created in ChemDraw (PerkinElmer Informatics).

**Contrast variation small angle neutron scattering**. Non-deuterated BiP$^{T229A-V461F}$-AMP (residues 27–635) and FICD$^{H363A}$ (residues 104–445) [hBiP-AMP and hFICD] were purified as detailed above but were gel filtered into a final buffer of TNKMT(0.2) [TNKMT buffer with TCEP reduced to 0.2 mM]. The matchout deuterium-labelled protein equivalents were produced in the ILL's deuteration laboratory (Grenoble, France). Proteins were expressed from E. coli BL21 Star (DE3) cells (Invitrogen) that were adapted to 85% deuterated Enfors minimal media containing unlabelled glycerol as carbon source[56,57], in the presence of kanamycin at a final concentration of 35 μg/ml. The temperatures at which the cells produced the highest amount of soluble matchout-deuterated BiP or FICD were chosen for cell growth using a high cell density fermentation process in a bioreactor (Labfors, Infors HT). For BiP expression, cells were grown using a fed-batch fermentation strategy at 30 °C to an OD$_{600}$ of 20. The temperature was then decreased to 18 °C and protein expression was induced by addition of 1 mM IPTG. After a further 22 h of protein expression at 18 °C, bacteria were harvested by centrifugation. FICD expression was conducted likewise, but with induction at OD$_{600}$ 19 and at a temperature of 22 °C. FICD-expressing cells were incubated for a further 21.5 h at 22 °C before harvesting. Matchout-deuterated proteins (dBiP$^{T229A-V461F}$-AMP and dFICD$^{H363A}$) were isolated and purified from deuterated cell pastes using H$_2$O-based buffer systems, as mentioned above, and gel filtered into TNKMT(0.2).

Heterotetrameric complexes were copurified by gel filtration of a mixture of either dBiP-AMP and hFICD or hBiP-AMP and dFICD (in a 1.25:1 molar ratio of BiP-AMP:FICD), with ≤5 mg of protein injected per SEC run, supplemented with 250 μM ATP. The gel filtration was conducted on an S200 Increase 10/300 GL column equilibrated with TNKMT(0.2) buffer. Heterotetrameric complex fractions were collected and concentrated to >7 mg/ml. Some of this purified complex was further exchanged by the same SEC process into TNKMT(0.2) in which the solvent used was D$_2$O. That is to say, the complex was exchanged into 100% D$_2$O buffer.

Protein fractions in 100% D$_2$O buffer were subsequently concentrated to >6 mg/ml. The elution profile appeared largely identical in both deuterated and non-deuterated buffers. Complexes at different %D$_2$O were obtained by either dilution with the appropriate matched buffer (± D$_2$O) or by the mixing of one complex purified in 0% D$_2$O buffer with the same complex in 100% D$_2$O buffer.

SANS data were collected from a total of 17 samples at various D$_2$O buffer compositions at 12 °C at the ILL beamline D11. Protein complexes (ranging from 4.3 to 5.5 mg/ml) were analysed in a 2 mm path-length quartz cell with a 5.5 Å wavelength neutron beam at distances of 1.4, 8 and 20.5 m. Data from relevant buffer-only controls were also collected with similar data collection times and subtracted from the radially averaged sample scattering intensities to produce the I(q) against q scattering curves presented in Fig. 2a. Scattering data were initially processed with the GRASP (Graphical Reduction and Analysis SANS Programme for Matlab; developed by Charles Dewhurst, ILL) and with the Igor Pro software (WaveMetrics) using SANS macros[58]. Data analysis was conducted using Prism (v8.4, GraphPad) and PEPSI-SANS (for fitting of theoretical scattering curves; software based on PEPSI-SAXS[28]). Flexible fitting model generation was also implemented through PEPSI-SANS software (https://team.inria.fr/nano-d/software/pepsi-sans/). In order to generate the best flex-fit model for each scattering curve a nonlinear rigid block (NOLB) normal mode analysis (NMA) method, utilising an all-atom anisotropic network model (ANM)[59], was employed. In brief from the starting input model (Fig. 1c) 100 models were sampled from along the 10 lowest frequency NMA trajectories. The derived models, after energy minimisation, were assessed for improved fit to the experimentally obtained scattering data. The best fitting model was then selected for a further round of NOLB NMA, as above. Iterative re-computation of the normal modes was carried out in this fashion for a total of 10 cycles. Rigid blocks (for NOLB NMA-based flex-fitting) were defined as BiP's NBD, SBDβ and SBDα and FICD's Fic domain, α-helical linker and TPR domain (see Fig. 1a and Supplementary Fig. 1b). In addition, a rigid block was defined as encompassing both Fic domains of the (dimeric) input structure in order to facilitate the flex-fitting analysis with a constrained FICD dimer interface (Fig. 2 and Supplementary Fig. 3, as indicated). Note, the utility of NOLB NMA has recently been demonstrated in both its ability to capture biologically relevant collective as well as localised protein transitions present in solution structures (and not captured in crystallo) without perturbing local protein geometry[60].

Comparison of the ln(Transmission) of the 0% and 100% D$_2$O buffers alone with the ln(Transmission) of each sample confirmed that the %D$_2$O of each sample was within the margin of error of the theoretical D$_2$O content[61].

Parameters from the Guinier plots were derived from fitting of the Guinier approximation[62]:

$$\ln(I(q)) = \ln(I(0)) - \frac{R_g^2}{3}q^2 \qquad (1)$$

The upper and lower q limits for fitting are shown (grey, vertical dashed lines in Fig. 2b and Supplementary Fig. 3a—except for the fitting of hFICD·dBiP-AMP in 60% D$_2$O buffer where the lower q limit ($q_{min}$) is denoted by a purple, vertical dashed line). These fitting ranges resulted in $0.15 < q_{min}R_g < 0.57$ and $0.39 < q_{max}R_g < 1.3$ (with the exception of the fitting of dFICD·hBiP-AMP in 80% D$_2$O buffer data where $q_{max}R_g = 1.4$).

The contrast match point analysis (CMP) in Fig. 2c indicated complex match points of 76.7% D$_2$O (95% confidence interval (CI): 71.5–82.4% D$_2$O) and 61.4% D$_2$O (95% CI: 57.4–65.5% D$_2$O) for hFICD·dBiP-AMP and dFICD·hBiP-AMP, respectively. Comparison of the experimental CMPs with theoretical values calculated by MULCh[63] (which takes into account buffer composition effects (at 20 °C) and protein sequence, whilst assuming a 1:1 complex and 95% labile H/D-exchange) suggested that there was 66.5% deuteration of dBiP-AMP and a 63.8% deuteration of dFICD. Note, these calculated values of non-exchangeable hydrogen deuteration are in-line with those expected from the 85% deuterated media and non-deuterated carbon source E. coli growth conditions, see above. For instance, under the same growth conditions, maltose-binding protein was found to be 64% deuterated at non-exchangeable hydrogens by intact protein mass spectrometry[57].

These values of dBiP-AMP and dFICD (non-labile) protein (partial) deuteration were used to calculate theoretical I(0)/c values in SASSIE[64], using the same assumptions as above. Comparison of the theoretical I(0)/c values with those determined from the experimental Guinier analysis facilitated experimental protein-complex MW estimation[27] (Supplementary Table 1). The contrast at each %D$_2$O (the difference in scattering length density (SLD), Δρ, between the ρ$_{protein}$ and ρ$_{buffer}$) was also derived from MULCh.

Stuhrmann analysis was carried out by the fitting of the relationship[30]:

$$R_g^2 = R_m^2 + \frac{\alpha}{\Delta\rho} - \frac{\beta}{\Delta\rho^2} \qquad (2)$$

In which $R_m^2$ represent the protein complex $R_g$ if it were to have a homogenous SLD. The value of α reflects the radial distribution of SLD, with values >0 suggesting that higher contrast components are located towards the outside of the complex and vice versa. The value of β reflects the distance of the centre of the complex's SLD from the complex's centre of mass. In the case of the Stuhrmann plot of dFICD·hBiP-AMP a linear best-fit line (suggesting β ≈ 0) was a considerably better fit to the data (shown in Fig. 2d; $R^2 = 0.93$) than the fitting of a quadratic curve ($R^2 = 0.66$). Theoretical $R_g$ values, derived from structural models, were calculated using CRYSON[65]. The symmetry of structural models was assessed through the use of AnAnaS software[66].

**Differential scanning fluorimetry (DSF).** DSF experiments were performed on a CFX96 Touch Real-Time PCR Detection System (Bio-Rad) in 96-well plates (Hard-Shell, Bio-Rad) sealed with optically clear Microseal 'B' Adhesive Sealer (Bio-Rad). Each sample was measured in technical duplicate and in a final volume of 20 μl. Protein was used at a final concentration of 2 μM, ATP or ADP (if applicable) at 5 and 2 mM, respectively, and SYPRO Orange dye (Thermo Fisher) at a 10× concentration in a buffer of HKM (25 mM HEPES–KOH pH 7.4, 150 mM KCl, 10 mM MgCl$_2$). Solutions were briefly mixed and the plate spun at 200 × g for 10 s before DSF measurement. Fluorescence of the SYPRO Orange dye was monitored on the FRET channel over a temperature range of 25–90 °C with 0.5 °C intervals (each lasting 5 s). Background fluorescence changes were calculated and subtracted from the protein sample fluorescence data using no-protein control (NPC) wells. NPC fluorescence was unchanged by the addition of ATP or ADP. Data was then analysed in Prism (v8.4, GraphPad), with melting temperatures calculated from the global minimums of the negative first derivatives of the relative fluorescent unit (RFU) melt curves (with respect to temperature).

**Bio-layer interferometry (BLI).** AMPylated or non-AMPylated biotinylated-AviTag-haBiP$^{T229A-V461F}$ (UK2359), was AMPylated if applicable, in vitro biotinylated, made apo and purified as detailed above in the section "Protein purification". Both proteins were confirmed as being >95% biotinylated by streptavidin gel-shift. All BLI experiments were conducted on the FortéBio Octet RED96 System (Pall FortéBio) using a buffer basis of HKM supplemented with 0.05% Triton X-100 (HKMTx). Streptavidin (SA)-coated biosensors (Pall FortéBio) were hydrated in HKMTx for at least 30 min at 25 °C prior to use. Experiments were conducted at 30 °C. BLI reactions were prepared in 200 μl volumes in 96-well microplates (greiner bio-one).

In Figs. 3 and 4 ligand loading was performed with biotinylated BiP-AMP:Apo at 7.5 nM and with biotinylated BiP:Apo at 5.8 nM, such that the rate of ligand loading was roughly equivalent and all tips reached a threshold of 1 nm binding signal (displacement) within 300–600 s. All ligands loaded with a range of 1.0–1.2 nm. After loading of the immobilised ligand, BiP was activated in 2 mM ATP for 200 s. This was followed by a 50 s baseline in HKMTx alone, before association with apo FICD variants (all bearing a catalytically inactivating His363Ala mutation and at 50 nM unless otherwise specified) in HKMTx (see schematic in Supplementary Fig. 4d). A no analyte reference biosensor was used to control for baseline drift. Note, immobilised (unmodified) BiP was previously observed to domain-dock, and remain domain-docked for extended periods of time in ATP-replete buffer, following this protocol of ATP activation[19]. The first dissociation step was initiated by the dipping of all tips into wells lacking FICD analyte (only HKMTx). The second dissociation step was induced by the dipping of the biosensor tips into HKMTx supplemented with 2 mM ATP. In Fig. 3b(ii) the second dissociation step was analysed by fitting a two-phase exponential decay, assuming a final plateau value of 0 nm. Experiments were conducted at a 1000 rpm shake speed and with a 5 Hz acquisition rate.

Analysis of the binding of the isolated TPR domain (UK2051) to unmodified BiP (Fig. 4b, c) was conducted as outlined above but with some modification. The ligand, biotinylated BiP:Apo (at 11.6 nM), was loaded onto the biosensor to ≥1.8 nm. Following a 50 s baseline in HKMTx the biosensor (bearing BiP:Apo) was dipped into a well solution containing 16 μM TPR domain for 50 s, followed by a 50 s dissociation step (also in HKMTx buffer). The same biosensor then underwent a 200 s activation step (by dipping into an HKMTx buffer supplemented with 2 mM ATP). This was proceeded by a 10 s baseline step in HKMTx and then sequential 50 s association and 50 s dissociation steps of the BiP:ATP bound biosensor into increasing concentrations of TPR domain (from 62.5 nM to 16 μM). Between each 100 s association–dissociation cycle a 10 s re-activation (in HKMTx plus 2 mM ATP) and subsequent 10 s baseline (in HKMTx) was carried out. Experiments were conducted at a 400 rpm shake speed and with a 10 Hz acquisition rate and a parallel no ligand reference biosensor was used to control for any non-specific analyte binding signal. All BLI data were processed in Prism (v8.4, GraphPad). In Fig. 4b the BiP:ATP-binding traces were analysed by global fitting of a one-phase association–dissociation-binding model (assuming a shared value of $k_{on}$, $k_{off}$ and $B_{max}$). In Fig. 4c the steady-state equilibrium binding response data was analysed by fitting a one site specific binding model. Equilibrium binding response was calculated by averaging the binding signal between 45 and 49.9 s in each association step (and adjusted relative to the initial level of BiP ligand loading on the biosensor after the first 50 s baseline step—between 1.8–2.0 nm). The lower estimate of the $K_D$ of FICD(TPR) for BiP:Apo was reached assuming that the same number of immobilised BiP:Apo and BiP:ATP molecules would possess equivalent maximum FICD(TPR)-binding potentials ($B_{max}$ values), whilst noting that binding of FICD(TPR) to BiP:ATP was detectable at an analyte concentration of <0.1 × $K_D$ and that no binding of 16 μM FICD(TPR) to immobilised BiP:Apo was observed.

**In vitro deAMPylation (fluorescence polarisation) assay.** The probe BiP$^{T229A-V461F}$ (UK2521) modified with FAM-labelled AMP:BiP$^{T229A-V461F}$-AMP(FAM)) was generated by pre-incubating 100 μM apo BiP$^{T229A-V461F}$ with 5 μM GST-FICD$^{E234G}$ (UK1479) and 110 μM ATP in HKM buffer for 5 min at 20 °C, followed by addition of 100 μM ATP-FAM [N$^6$-(6-Amino)hexyl-ATP-6-FAM; Jena Bioscience] and further incubation for 19 h at 25 °C. To ensure complete BiP AMPylation 2 mM ATP was then added to the reaction which was incubated for a

further 1.25 h at 25 °C. The reaction mixture was then incubated with GSH-Sepharose 4B matrix for 45 min at 4 °C in order to deplete the GST-FICD$^{E234G}$. The BiP containing supernatant was buffered exchanged into HKM using a Zeba Spin desalting column (7 K MWCO, 0.5 ml; Thermo Fisher) in order to remove the majority of free (FAM labelled) nucleotide. 2 mM ATP was added to the eluted protein and incubated for 15 min at 4 °C (to facilitate displacement of any residual FAM-labelled nucleotide derivates bound by the NBD of BiP). Pure BiP-AMP (FAM) with BiP-AMP was then obtained by gel filtration using an S75 Increase 10/300 GL column equilibrated in HKM at 4 °C. 1 mM TCEP was added to the protein fractions, which were concentrated using a 50 K MWCO centrifugal filter and snap frozen. A labelling efficiency of 1.8% was estimated based on the extinction coefficient for BiP-AMP:ATP ($\varepsilon_{280}$ 33.5 mM$^{-1}$ cm$^{-1}$), FAM ($\varepsilon_{492}$ 83.0 mM$^{-1}$ cm$^{-1}$) and a 280/492 nm correction factor of 0.3 (Jenna Biosciences).

DeAMPylation reactions were performed in HKMTx(0.1) buffer [HKM supplemented with 0.1% (v/v) Triton X-100] in 384-well polystene microplates (black, flat bottom, μCLEAR; greiner bio-one) at 30 °C in a final volume of 30 μl containing trace amounts of fluorescent BiP$^{T229A-V461F}$-AMP(FAM) probe (10 nM), supplemented with BiP$^{T229A-V461F}$-AMP (5 μM) and FICD proteins (0.5 μM). A well lacking FICD protein was used for baseline FP background subtraction. 10 nM ATP-FAM alone was also included as a low FP control. Under these conditions $[E]_0$ was assumed to be << $[S]_0 + K_M$ (with $[E]_0 = 0.5$ μM, $[S]_0 = 5$ μM and the presumed $K_M$ (Michaelis constant) ≥ GST-FICD $K_M$ of 16 μM[10]) such that quasi-steady-state reaction kinetics should apply with respect to the initial reaction rate. Furthermore, $[S]_0$ was considered to be sufficiently small relative to the FICD variant presumed $K_M$ values such that, by derivation from the Michaelis–Menten equation[67], the following relationship holds true:

$$v_0 \approx \frac{k_{cat}}{K_M}[E]_0[S]_0 \qquad (3)$$

where $v_0$ is the measured initial reaction velocity. On account of the close correspondence between the values calculated here and previously (from a Michaelis–Menten analysis of GST-FICD[10]) these assumptions are clearly valid for wild-type FICD. More accurately all presented ~ $k_{cat}/K_M$ values are in fact equivalent to $k_{cat}/(K_M + [S]_0)$.

Fluorescence polarisation of FAM ($\lambda_{ex} = 485$ nm, $\lambda_{em} = 535$ nm) was measured with an Infinite F500 plate reader (Tecan). The mFP $y_0$ difference between the FICD$^{L258D}$ time course and the same reaction composition pre-incubated for 5 h at 25 °C before the beginning of data collection, was interpreted as the ΔmFP equivalent to complete (5 μM) BiP-AMP deAMPylation (see Supplementary Fig. 5a). Fitting of the initial linear reaction phase was achieved using Prism (v8.4, GraphPad).

For direct calculation of $k_{cat}$ values deAMPylation assays were conducted as above but with 10 μM FICD or FICD$^{L258D}$ and 100 or 150 μM BiP-AMP substrate. Following subtraction of a no enzyme background from all datasets, the mFP difference for each sample (between $t = 0$ and the mFP plateau) was interpreted as the ΔmFP equivalent to complete BiP-AMP deAMPylation ($[S]_0$).

**In vitro AMPylation.** In vitro AMPylation reactions were performed in HKM buffer in a 7 μl volume. Reactions contained 10 μM ATP-FAM, 5 μM ATP-hydrolysis and substrate-binding-deficient BiP$^{T229A-V461F}$ (UK2521), 7.5 μM oxidised $_{s-s}$FICD$^{A252C-H363A-C421S}$ (UK2269, trap) to sequester any modified BiP [BiP-AMP(FAM)] and, unless otherwise stated, 0.5 μM FICD. Reactions were started by addition of nucleotide. Apart from the presented time courses (Supplementary Fig. 6c) after a 60 min incubation at 25 °C the reactions were stopped by addition of 3 μl 3.3 × LDS sample buffer (Sigma) containing NEM (40 mM final concentration) for non-reducing SDS–PAGE or DTT (50 mM final concentration) for reducing SDS–PAGE and heated for 10 min at 70 °C. Samples were applied to an SDS–PAGE gel and the FAM-label was imaged with a Chemidoc MP (Bio-Rad) using the Alexa Flour 488 dye setting. Gels were subsequently stained with Quick Coomassie (Neo Biotech).

**Mammalian cell culture and lysis.** The CHO-K1, CHO-K1 *FICD*$^{-/-}$, CHO-K1 S21 *FICD*$^{-/-}$ cell lines used in this study were described previously[3]. Cells were cultured in Nutrient mixture F-12 Ham (Sigma) supplemented with 10% (v/v) serum (FetalClone II; HyClone), 1 × Penicillin–Streptomycin (Sigma), and 2 mM L-glutamine (Sigma) at 37 °C and 5% CO$_2$. Cells were grown on tissue culture dishes or multi-well plates (Corning) and experiments were performed at cell densities of 60–90% confluence. Cell lines were confirmed as being free of *Mycoplasma* contamination by random testing using the MycoAlert *Mycoplasma* Detection Kit (Lonza). Where indicated, cells were treated for 3 h with cycloheximide (Sigma) by exchanging the culture medium with pre-warmed (37 °C) medium supplemented with cycloheximide at 100 μg/ml. Cell lysates were obtained and analysed as in[19] but with a HG lysis buffer consisting of 20 mM HEPES–KOH pH 7.4, 150 mM NaCl, 2 mM MgCl$_2$, 33 mM D-glucose, 10% (v/v) glycerol, 1% (v/v) Triton X-100 and protease inhibitors (2 mM phenylmethylsulphonyl fluoride (PMSF), 4 μg/ml pepstatin, 4 μg/ml leupeptin, 8 μg/ml aprotinin) with 100 U/ml hexokinase (from *Saccharomyces cerevisiae* Type F-300; Sigma).

**Immunoblot (IB) analysis**. After separation by SDS–PAGE or native-PAGE[19] proteins were transferred onto PVDF membranes. The membranes were blocked with 5% (w/v) dried skimmed milk in TBS (25 mM Tris–HCl pH 7.5, 150 mM NaCl) and incubated with primary antibodies diluted in 3% (w/v) BSA in TBS supplemented with 0.1% Tween-20 (TBST). Primary antibodies and antisera against hamster BiP [chicken anti-BiP[68]], eIF2α [mouse anti-eIF2α[69]] and FICD [chicken anti-FICD[3]] were used at a dilutions of 1/1000, 1/5000 and 1/1000 (v/v), respectively. Following the primary antibody incubation, the PVDF membrane was washed with TBST and then incubated with IRDye fluorescently labelled secondary antibodies (LI-COR) at a dilution of 1/2000 (v/v) in a solution of 3% (w/v) dried skimmed milk in TBS. The membranes were scanned with an Odyssey near-infra-red imager (LI-COR). Where applicable, IB band quantification was carried out with Image Studio Lite software (LI-COR).

**Flow cytometry**. FICD over-expression-dependent induction of unfolded protein response signalling was analysed by transient transfection of CHO-K1 S21 *FICD*−/− UPR reporter cell lines with plasmid DNA encoding the complete FICD coding sequence (with mutations as indicated) and mCherry as a transfection marker, using Lipofectamine LTX (Thermo Fisher). 0.5 μg DNA was used to transfect cells growing in 12-well plates. 40 h after transfection the cells were washed with PBS and collected in PBS containing 4 mM EDTA, and single live-cell fluorescent signals (20,000 collected per sample) were analysed by dual-channel flow cytometry with an LSRFortessa cell analyser utilising FACSDiva acquisition software (BD Biosciences). Turquoise and mCherry fluorescence was detected using a 405 nm excitation laser with a 450/50 nm emission filter and a 561 nm excitation laser with a 610/20 nm emission filter, respectively. Data were processed using FlowJo X (BD Bioscience) and the extracted population parameters were plotted in Prism (v8.4, GraphPad).

**Reporting summary**. Further information on research design is available in the Nature Research Reporting Summary linked to this article.

## Data availability

The deAMPylation complex crystal structures of monomeric FICD and AMPylated BiP have been deposited in the Protein Data Bank (PDB) with the following accession codes: 7B7Z (state 1) and 7B80 (state 2). Crystal structure data from previous studies are also available in the PDB, deposited with the following accession codes: 5O4P, 4U0U, 5E84, 7A4U, 6I7K, 6I7L and 6ZMD. Raw SANS data is available from https://doi.org/10.5291/ILL-DATA.8-03-963[70]. Source data are provided with this paper.

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

## Acknowledgements
We thank the Huntington lab for access to the Octet machine and the CIMR flow cytometry core facility team (Reiner Schulte, Chiara Cossetti and Gabriela Grondys-Kotarba). This work was supported by Wellcome Trust Principal Research Fellowship to D.R. (Wellcome 200848/Z/16/Z). We are grateful to the Diamond Light Source for X-ray beamtime (proposal MX-21426) and the staff of beamline I04-1 for assistance with data collection; and to the ILL for neutron beamtime as part of proposal 8-03-963 with particular thanks to Anne Martel for her assistance. For advice pertaining to the use of PEPSI-SANS software we thank Sergei Grudinin. We are indebted to Yahui Yan for the gift of FICD's TPR domain-expressing plasmid and to Cláudia Rato da Silva for advice and guidance on the in vivo experiments. We also thank Yahui Yan, Alisa F. Zyryanova and Lisa Neidhardt for comments on the manuscript.

## Author contributions
L.A.P. led and conceived the project, designed and conducted the experiments, analysed and interpreted all the data, purified and crystallised proteins, collected, analysed and interpreted the X-ray diffraction and neutron scattering data, and wrote the manuscript. S. Preissler conducted the FP assays and purified proteins. N.R.Z. and S. Prévost helped to collect and process scattering data. J.M.D. and M.H. expressed the deuterated proteins. All authors contributed to revising the article. D.R. conceived and oversaw the project, interpreted the data and cowrote the manuscript.

## Competing interests
The authors declare no competing interests.
