## [Peer Review File · Nature Communications]

REVIEWER COMMENTS

Reviewer #1 (Remarks to the Author):

The manuscript by Perera et al., submitted to Nature communications, represents a state-of-the-art structural and functional study of the heat shock protein BiP, a key player regulating the chaperoning activity in endoplasmatic reticula, and the enzyme FICD, which is responsible for a post-translational modification ((de)AMPylation) of BiP.

The authors determined the crystallographic structure of a slightly modified FICD-BiP complex with respect to the wild type (single amino acid mutation and truncation of some residues). In addition to the X-ray crystallographic data solution neutron scattering (SANS) data of the intact, wild-type complex are presented. Complementary to the structural data at high and low resolution, biophysical measurements (Differential Scanning Fluorimetry, BioLayer Interferometry) and a number of biochemical assays (in vitro deAMPylation on several mutants) were carried out to further characterize the interaction between both protein partners. Based on the high-quality ensemble of structural, biophysical and biochemical data, the authors propose a detailed mechanism of eukaryotic deAMPylation, both at low-resolution (conformational flexibility, role of oligomeric state) as well as at atomic resolution (e.g. role of specific amino acid residues).

Overall, the manuscript by Perera et al. is of outstanding quality. The biological topic is introduced both at a general level which makes it accessible to the broad readership of Nature Communications but also in sufficient depth for a more specialized community working in this field. All data are of excellent statistical quality. The thorough and innovative combination of X-ray crystallography, SANS, biophysical techniques and mutational analysis support most of the authors' interpretations and conclusions (however, see my remarks below on flexibility inferred from solution scattering).

The design, presentation, analysis and interpretation of the SANS data is particularly innovative but some points require clarification. My following remarks and inquiries concern mainly this part of the manuscript.

Major points:

- 1) The innovative "flex-fit model" approach is at the core of the SANS data analysis and interpretation. However, some clarifications and additional information need to be provided by the authors. First, it is not clear what models the method actually generates. The authors state (p. 8, top) that normal modes deform the input structure by "flexing". Usually, a normal mode defines a

trajectory for each atom and not a single, unique structure. Do the fitted structures (e.g. values shown in Suppl. Fig. 2b-d) correspond to some extreme conformation within each normal mode or are the χ^2 values averaged along each normal mode trajectory?

2) Related to the same issue: the authors need to better explain and demonstrate how their “flex-fit” approach supports the presence of flexibility in the present system, in particular regarding the presentation of the results on p. 8 and Suppl. Fig. 2. As they rightly state (p. 8, top), their SANS data indicate that the (static) input model presented in Fig. 1c is already in very good agreement with all scattering curves (dotted black lines in Fig. 2a). By visual inspection, it is not evident that the fits with the “flexible-fit approach” (dashed green lines) are actually better than the fits with the static input model (Fig. 2a and Suppl. Fig. 2g). In each case, actual χ^2 values need to be indicated, and ideally also a comparative plot of the fit residuals. The authors also need to indicate if the dashed green lines correspond to an average curve, calculated from all 12(?) flex-fit structures (Suppl. Fig. 2c).

3) Supplementary Figure 2c also requires some explanations: are the horizontal broken lines referring to the χ^2 value obtained with the fit of the input model shown in Fig. 1c? Do the χ^2 values, and the errors reported, represent an average χ^2 from all SANS curves shown in Fig. 2a? And are the SANS data series from both oppositely labeled complexes combined here?

4) The central issue, however, and unless I misunderstood or missed something, is the following: assuming that there was some degree of flexibility, and it were possible to represent it by normal modes. The resulting SANS curve would then correspond to a weighted average of all possible modes. If the SANS curve resulting from this average cannot be distinguished (in terms of χ^2) from the SANS curve of a single input structure, it is impossible to infer the presence of flexibility, even though some individual modes might have better or worse fits with the SANS curves. It is true that Kratky plots (as shown by the authors in Suppl. Fig. 2g) can indicate the presence of flexibility, e.g. via a plateau or increasing intensities at high q -values. Unfortunately, the q -range accessible in the SANS experiment does not seem to be sufficient to conclude on this point, and the intensities rather display a tendency to decrease at higher q -values which would indicate compact, non-flexible complexes.

5) Guinier fits (Fig. 2b and Suppl. Fig. 2a): according to the material and methods (p. 23), the vertical dashed lines delimit the ranges of the fits. However, the fits themselves (broken black lines?) extend beyond the lower limits down to smallest angles. It is also not clear why there are several grey dashed vertical limits, but also a single purple dashed limit (Suppl. Fig. 2a). More importantly, it is unlikely that the upper limit of the fit range (defined by $qR_g < 1.3$ for most samples) is the same for all conditions (as suggested by the identical ranges of the black broken lines in all cases). In general, R_g varies as a function of contrast and macromolecular labeling, and this seems to be the case in the present study (as supported by the values reported in the Stuhrmann plot, Fig. 2d). All these points need some clarification. I would suggest that the authors limit and adopt the black broken lines to the actual Guinier fit ranges in each case. Finally, the authors seem to denote the zero y -axis by dotted black symbols. This choice is confusing since it recalls the symbols of the flex-fits in Fig. 2a. Finally, it must read “Supplementary Fig. “2a” (and not “2b”) at the bottom of p. 23.

Minor points:

a) Did the authors determine the final deuteration degree of the “match-out deuterated” BiP and FICD proteins experimentally, e.g. by mass spectrometry? Or can they at least cite the theoretically expected values of the deuteration protocols (the information is probably available in references 49 or 50 but it would be preferable if it could be explicitly indicated in the supplement). In this context, it would also be useful if the authors could quickly comment on the distribution of deuteration throughout the protein structures: is it uniform/random for all amino acids or targeted at certain types of amino acids? Finally, was the determined (or expected) deuteration degree in agreement within the error margins of the experimentally determined match-points (Fig. 2c) and the values predicted by MULCh?

b) Supplementary Fig. 2b reveals a certain tendency of the quality of fits as a function if either the one or the other protein is deuterated: in the case of the hFICD-dBiP complex, the best fits are obtained at high D2O levels of the solvent, whereas for the oppositely labeled complex, the best fits are obtained at low D2O levels. Can the authors comment on this tendency?

c) The statement/observation that leaving the dimer interface unrestrained yields better fits to the SANS curves than a restrained interface (p.8, bottom) seems obvious due to the greater degree of freedom, and needs some explanation.

d) It is a little bit confusing to present an average chi2 value of 3.4 with a standard deviation of +/- 4 (p. 7, bottom). Chi2 are always positive. Can the authors comment?

e) A couple of uncommon abbreviations (e.g. “Cl”, “SEM”) should be introduced throughout the manuscript upon their first occurrence.

f) Please provide some general references for SAXS and in particular for SANS when they are first mentioned (p. 7, top).

g) The black dots in Fig. 2a are hardly visible. Please either use a different symbol or increase dot size.

Reviewer #2 (Remarks to the Author):

The manuscript by Perera et al reported a crystal structure of a FICD-BiP complex stabilized by a number of mutations on both FICD and BiP. This structure was validated by SAXS/SANS and biochemical analysis on mutations. As the master regulator of protein homeostasis in ER, BiP is regulated through AMPylation by FICD. Understanding the molecular mechanism of the AMPylation of BiP by FICD is important for dissecting the regulation of BiP's function in stress response. This manuscript is from the laboratory of Dr. David Ron, who is one of the leaders in the chaperone field and has published extensively in the area of BiP regulation by AMPylation.

However, there are a number of major concerns to prevent it from publishing.

1. Although the structure determination, solution and biochemical analysis are solid, there is no novel mechanism proposed. The model proposed in Fig. 6 is essentially identical to the model published in their 2019 EMBO paper (Fig. 7) except that BiP was positioned. Even the claimed “unanticipated finding” on Glu234 flexibility was published in the EMBO paper. It seems that nothing mechanistically new was gained through this manuscript.

2. To support their FICD-BiP crystal structure, SAXS/SANS was carried out. There are a number of serious issues with this work.

1) The authors kept on describing this work as “solution structure”. SAXS/SANS can not generate any solution structure. They are simply models/predictions. These models/predictions are of very low resolution (less than 20 Å). These solution approaches generate only 2D data and try to fit 3D structures. Their application is quite limited.

2) When fitting their SAXS/SANS data on full-length BiP, the authors used their truncated BiP structure with modeled SBD alpha. Structures of full-length BiP in complex with ATP are available. The authors should use these structures. Otherwise, the deduced model is less reliable since it's based on modeled structures.

3) For these studies, the authors first need to show conformational homogeneity of the protein sample. Otherwise, it's almost impossible to deduce any meaningful structural information.

3. For the reported crystal structures, I/σ at the highest resolution shell is 1, i.e. $I=\sigma$. There is no signal at all. The 100% completeness is meaningless.

4. The authors kept on saying that the ATP-bound conformation of BiP is essential for FICD binding based on their complex structure. This seems one of the key points of the manuscript. The authors should test this claim using BiP mutations.

Moreover, Hsp40 co-chaperones also specifically recognize and bind the ATP-bound conformation of Hsp70s. Do the observed BiP-FICD interfaces overlap with the Hsp70-Hsp40 interfaces?

5. Hsp70s are highly conserved, especially the ATP-bound conformation as shown by a number of structural analyses. Are the BiP residues involved in interacting with FICD conserved in Hsp70s? If they are, does FICD modify cytosolic Hsp70 and Hsc70s since FICD has been found in the cytosol?

In addition, the authors should do a detailed structural comparison of the reported FICD-BiP complex with published Fic-enzymes in complex with their substrates (such as the IbpA-Cdc42 complex). It seems that the TPR-like motifs of IbpA also interacts with Cdc42. Are there any similarities?

6. The authors should test the interface between the FICD catalytic domain and the SBD beta of BiP where T518 is located. The key question is how BiP-T518 is recognized by FICD as a substrate. Can their structure provide any insights?

7. Throughout the manuscript, the authors have been comparing the new FICD-BiP complex structure with only structures solved by their own group, ignored the structural work on BiP and other Hsp70s published by other groups. To understand mechanism, their BiP structures may not be the best for comparison since these structures carry more truncations and mutations. For Fig. 1, besides using their published SBD alpha truncation structure of BiP for comparison, they should compare their new FICD-BiP complex structure with the full-length BiP structures.

Overall, the authors referenced their published work extensively. The authors should also reference works from other labs.

Page 9: the third line, should reference Hsp70-ATP and Hsp70-ADP crystal structures, not only NMR work.

8. TRPox: should test the reduced mutant protein to exclude the possibility that the observed effect is due to the cysteine mutations themselves. Does the reduced form behave like WT? If not, the observed effect is caused by the Cys mutations, not the disulfide bond.

9. Fig. 3bii, there is no significant difference between the WT and the three mutations (E105R, H131A, and TRPox), especially H131A. But, the authors described in the manuscript that all the mutants have significant difference. This is misleading.

10. For the BioLayer Interferometry, the authors should do the BiP apo and BiP-ADP controls to exclude the possibility that FICD binds BiP as a substrate. Moreover, should check ATP hydrolysis by BiP at 30 degree during the course of the experiments. Normally Hsp70s hydrolyze ATP completely within 30 min at 30 degree.

11. Fig. 3C: why was E105R not tested?

Reviewer #3 (Remarks to the Author):

The eukaryotic FICD enzyme targets the Hsp70 chaperone BiP from the ER. FICD is a bifunctional enzyme with antagonistic activities: AMPylation of BiP inactivates the chaperone, whereas deAMPylation relieves the block. The deAMPylation activity of FICD was discovered by the Ron group a few years ago (Preissler et al., 2017), followed by an investigation into the structural determinants (oligomeric state) of FICD regulation (Perera et al., 2019).

Now, Perera et al report the crystal structure of the Michaelis-Menten complex of FICD with AMPylated BiP. Turnover of the BiP-AMP substrate was thereby prevented by using a FICD mutant that had the histidine of the active site replaced by alanine. The structure represents an important advancement in our understanding of (1) FICD/target recognition, which is dependent on the conformation of the BiP target, and (2) the mechanism of FICD catalyzed deAMPylation. Latter result may well be of relevance also for the understanding of bacterial Fic enzymes (see the quoted Veyron et al., 2019 paper).

More specifically, the central role of FICD Glu234 in the deAMPylation reaction (in addition to its well-known inhibitory role in AMPylation) is rationalized in that it directly aligns the hydrolytic water. The statement in the abstract, however, that monomerization-induced increase in flexibility of this glutamate is an “unanticipated finding” of this study seems not to be correct, since this was the major conclusion of the Pererea et al., 2019 paper on the differential regulation of FICD.

Since the crystal structure was determined from a monomerization mutant, solution studies were carried out on the dimeric wild-type enzyme confirming that the same FICD/BiP interactions occur as in the crystal. Furthermore mutant data are presented validating the observed FICD/BiP interactions and their dependence on the conformational state (domain-docked or not) of BiP.

The presentation of the paper should be stream-lined and improved considerably. Wouldn't it be better to (1) present the overall complex structure, (2) show the solution and mutant studies that are relevant to the overall complex structure, and (3) zooming into the details of the active site and discuss deAMPylation mechanism?

Below I'm adding specific comments/suggestions.

Figures:

0. A figure showing a schematics of the employed constructs (as in Preissler et al. eLife 2017, Fig. 2a) would be very helpful.

1. Amazingly all the main findings are crammed into Fig. 1. I'd suggest to

1.1. Move panel a to Supplement.

1.2. Show the overview (1.bi) without superposition. Superpositions merely demonstrating no difference are better placed in the Suppl. (if needed at all).

1.3. Show the two contact areas (1.bii) in roughly the same orientation as in the overview.

1.4.. For the Fic - SDBbeta contact: are there other interactions apart from those with T518-AMP?

1.5. Show flap with same color in panel bi as in panel bii.

1.6. Panel c is way too complex. I understand that it is merely there to demonstrate that BiP can bind also to dimeric FICD. This is shown later in Fig. 2 anyway. So, omit this panel.

2.1. 1.d and 1.e deserve their own figure. Show 1.d without density, for clarity. Don't forget the H-bonds between water * and the main-chain amides of the N-cap of the underlying helix. Enlarge the figure, such that the FICD flap is seen (with H-bonds of augmented β -sheet).

2.2. I'd strongly suggest to add a figure of the competent FICD/AMPPNP complex (6i7l) in same orientation (side-by-side comparison). Should be helpful for discussing FICD's bifunctionality.

3.1. Fig. 3a: What is meant by 2 mM ATP dissociation? Addition of ATP to start AMPylation reaction? Explain in the legend.

3.2. Comparing Figs. 3 and 4 I'm rather confused. Fig. 3 deals with deAMPylation of BiP-AMP only, as suggested by the title? Fig. 4 deals with AMPylation? Would it not be better to organize into separate figures corresponding to the methods employed?

4.1. Fig. 6: show Ndelta proton on active His. Show protons of catalytic water. The double arrow between low and high would be better placed in the middle of the catalytic cycle.

5. Using depth-cueing would help for the complex figures.

Figure legends:

Always refer to related Suppl. Figures in the legends. E.g. Fig.1 > Fig. S1.

1061 and in other places: ...attack into the α -phosphate >> ... onto ...

1058: ...is essential for AMPylated BiP binding and deAMPylation >> ...is essential for BiP-AMP:ATP binding and deAMPylation 1070: phrase analogously as in 1058 (unless figures get re-organized)

1092: The title is misleading. Glu234 flexibility (compared to what?) is not investigated in this chapter. So, don't confuse results and interpretation.

1104: Be more clear. The authors mean that the initial velocities are the same for the two concentrations and conclude that the enzyme is operating at saturating.

Tables:

5.1 Would be very helpful to have, for comparison, the results (kcat, Km) of all kinetic measurements in a table. This would also help to make the text more readable.

Text:

40 ff: expand, introduce the BiP domains and define the domain-docked, linker-bound conformation

67: Unclear. You mean substrate concentration and whereby amount of dimeric species?

Paragraph starting 67: Review the current understanding how AMPylation renders BiP inactive. After all, the modification affects a very peripheral site (T518).

89: Define more clearly the constructs (best with a scheme).

97: Sounds strange: "...despite having hydrolysed its bound MgATP".

paragraph starting line 94: use present tense throughout.

101: You may want to rephrase to s.th. like "Beta-strand augmentation of the Fic flap by the modifyable BiP segment "

104: (implicated in a bacterial Fic protein AMPylation-substrate binding 16,19) >> (as has been observed for bacterial Fic proteins 16,19)

116: "...linker between ... and the ER membrane" Be more clear. How is BiP anchored to the membrane?

127: "oxyanion hole" Term used for proteases. Here the situation is rather an "anionic nest" at the N-term of a helix (see Goepfert et al., 2013 and reference therein)

132: "and likely participates in catalysis " >> "most likely constitutes the hydrolytic water molecule "

244: "TPR oxidation " comes out of the blue. In the wild-type, is there a disulfide? This has not been introduced.

267 (and elsewhere): Adopt a consistent and simple notation. E.g. BiP:ATP, BiP-AMP:ATP, etc. So, use BiP-AMP:ATP, instead of "unmodified ATP-state BiP"

322: "Increased flexibility": this is not shown here (increased w/r to what ? No comparative measurements shown). It is rather an interpretation. So the title should be rather something like : " monomerization mutant shows decreased deAMPylation activity"

408: It appears improbable that Glu234 can act as a base for deprotonating the hydrolytic water. It forms ion interactions with two arginines, so the energy cost of protonation would be very high (pK shift towards low value)! Please, revise. However, the highly positive environment of the catalytic water (bonded to main-chain amides (oxyanion hole. By the way: better call it anionic nest, see Goepfert reference) and the Mg⁺⁺ in the vicinity may shift the water pK (Lewis base concept).

411: What, in fact, is the role of the Mg⁺⁺?

435: ... binds more tightly to unmodified BiP than BiP-AMP > ... binds more tightly to unmodified BiP than to BiP-AMP By the way: how can this be explained by the structure?

447: The last sentence of the discussion is cryptic. Can't be understood without checking the cited papers probably.

Tilman Schirmer

POINT-BY-POINT RESPONSE TO REVIEWERS COMMENTS ON NCOMMS-21-06435

We thank the reviewers for their time and effort in offering a detailed critique of our work. In an effort to respond to the many detailed comments, the revised version of the manuscript has been substantively rearranged (in terms of the order in which various figures/figure panels and results sections are presented). Where applicable in the following point by point we will attempt to direct the reviewers to the updated figure panels. Note, where quoted from the manuscript text will appear in blue and where new text is presented it will also be highlighted.

Reviewer #1 (Remarks to the Author):

The manuscript by Perera et al., submitted to Nature communications, represents a state-of-the-art structural and functional study of the heat shock protein BiP, a key player regulating the chaperoning activity in endoplasmatic reticula, and the enzyme FICD, which is responsible for a post-translational modification ((de)AMPylation) of BiP.

The authors determined the crystallographic structure of a slightly modified FICD-BiP complex with respect to the wild type (single amino acid mutation and truncation of some residues). In addition to the X-ray crystallographic data solution neutron scattering (SANS) data of the intact, wild-type complex are presented. Complementary to the structural data at high and low resolution, biophysical measurements (Differential Scanning Fluorimetry, BioLayer Interferometry) and a number of biochemical assays (in vitro deAMPylation on several mutants) were carried out to further characterize the interaction between both protein partners. Based on the high-quality ensemble of structural, biophysical and biochemical data, the authors propose a detailed mechanism of eukaryotic deAMPylation, both at low-resolution (conformational flexibility, role of oligomeric state) as well as at atomic resolution (e.g. role of specific amino acid residues).

Overall, the manuscript by Perera et al. is of outstanding quality. The biological topic is introduced both at a general level which makes it accessible to the broad readership of Nature Communications but also in sufficient depth for a more specialized community working in this field. All data are of excellent statistical quality. The thorough and innovative combination of X-ray crystallography, SANS, biophysical techniques and mutational analysis support most of the authors' interpretations and conclusions (however, see my remarks below on flexibility inferred from solution scattering).

The design, presentation, analysis and interpretation of the SANS data is particularly innovative but some points require clarification. My following remarks and inquiries concern mainly this part of the manuscript.

Major points:

1) The innovative "flex-fit model" approach is at the core of the SANS data analysis and interpretation. However, some clarifications and additional information need to be provided by the authors. First, it is not clear what models the method actually generates. The authors state (p. 8, top) that normal modes deform the input structure by "flexing". Usually, a normal mode defines a trajectory for each atom and not a single, unique structure. Do the fitted structures (e.g. values shown in Suppl. Fig. 2b-d) correspond to some extreme conformation within each normal mode or are the chi2 values averaged along each normal mode trajectory?

We thank the reviewer for allowing us to clarify this point we have now expanded on the nature of the flexible fitting strategy employed within the *Contrast Variation Small Angle Neutron Scattering* section of the **Materials and Methods**, adding the following paragraph:

Flexible fitting model generation was also implemented through PEPSI-SANS software (<https://team.inria.fr/nano-d/software/pepsi-sans/>). In order to generate the best flex-fit model for each scattering curve a nonlinear rigid block (NOLB) normal mode analysis (NMA) method, utilising an all-atom anisotropic network model (ANM)⁵⁸, was employed. In brief from the starting input model (Fig. 1c) 100 models were sampled from along the 10 lowest frequency NMA trajectories. The derived models, after energy minimisation, were assessed for improved fit to the experimentally obtained scattering data. The best fitting model was then selected for a further round of NOLB NMA, as above. Iterative re-computation of the normal modes was carried out in this fashion for a total of 10 cycles. Rigid blocks (for NOLB NMA-based flex-fitting) were defined as BiP's NBD, SBD β and SBD α and FICD's Fic domain, α -helical linker and TPR domain (see Fig. 1a and Supplementary Fig. 1b). In addition, a rigid block was defined as encompassing both Fic domains of the (dimeric) input structure in order to facilitate the flex-fitting analysis with a constrained FICD dimer interface (Fig. 2 and Supplementary Fig. 3, as indicated). Note, the utility of NOLB NMA has recently been demonstrated in both its ability to capture biologically relevant collective as well as localised protein transitions present in solution structures (and not captured in crystallo) without perturbing local protein geometry⁵⁹.

2) Related to the same issue: the authors need to better explain and demonstrate how their “flex-fit” approach supports the presence of flexibility in the present system, in particular regarding the presentation of the results on p. 8 and Suppl. Fig. 2. As they rightly state (p. 8, top), their SANS data indicate that the (static) input model presented in Fig. 1c is already in very good agreement with all scattering curves (dotted black lines in Fig. 2a). By visual inspection, it is not evident that the fits with the “flexible-fit approach” (dashed green lines) are actually better than the fits with the static input model (Fig. 2a and Suppl. Fig. 2g).

We agree with the reviewer that by visual inspection the flex-fitting of the input model to each scattering curve, presented in Fig. 2a and generating 12 output structures, is very similar to the fitting achieved with the input model (from Fig. 1c) at all %D₂O buffer conditions. As highlighted in the text, we believe this speaks to the very good agreement between our input structure and the structure and homogeneity of the complex present in solution. Moreover, as pointed out in the text, the good agreement of our input with the scattering data at D₂O buffer conditions approaching the contrast match points of both oppositely isotopically labelled complexes, suggest that the SANS protein samples contain a highly homogenous solution with shape and stoichiometry very close to that predicted from the input structure.

Despite the similarity by visual inspection, subtle changes between the fitting of the input structure and flex-fit structures are visible in Fig. 2a (for example in the q region around 1 nm⁻¹ in the scattering curves of hFICD•dBiP-AMP in 0 and 10% D₂O). Furthermore, we have now added an extra reference Supplementary Fig. 3b (was Supplementary Fig. 2b).

In each case, actual chi² values need to be indicated, and ideally also a comparative plot of the fit residuals.

Supplementary Fig. 3b provides a visual representation of all the 12 dimer-constrained flex-fit models plus the input model to each of the 12 scattering datasets. Comparison of the NW to SE cross-diagonal to the input model column demonstrates the difference between the fitting of the 12 flex-fit structures (from the derived scattering curves, i.e. green dashed lines in Fig. 2a) and the fits of the input model structure (black dotted line in Fig. 2a). In Supplementary Fig. 3c (was Supplementary Fig. 2c) the mean \pm SEM of the χ^2 values derived from fitting each of the 12 output models (plus the input model) to all 12 scattering curves is displayed (i.e. the symbols in the left panel of Supplementary Fig. 3c represent the averages of the columns of χ^2 data presented in Supplementary Fig. 3b). We have

now added extra reference to this panel in the results section (upon the introduction of the flex-fit strategy) and also within the figure legend of **Fig. 2a**.

The authors also need to indicate if the dashed green lines correspond to an average curve, calculated from all 12(?) flex-fit structures (Suppl. Fig. 2c) [**Fig. 2a**].

We have clarified this point in the presentation of **Fig. 2a** and have amended the figure legend appropriately:

dashed green lines are the theoretical scattering curves from flex-fitting of the input heterotetramer model to each scattering curve individually (with a constrained FICD dimer interface). See **Supplementary Fig. 3b–e**.

3) Supplementary Figure 2c also requires some explanations: are the horizontal broken lines referring to the chi2 value obtained with the fit of the input model shown in Fig. 1c? Do the chi2 values, and the errors reported, represent an average chi2 from all SANS curves shown in Fig. 2a? And are the SANS data series from both oppositely labeled complexes combined here?

Likewise, we have now added further details to the corresponding legend:

c, Comparison of the mean χ^2 for each model derived from analysis of the goodness of fit to all scattering datasets (generated from both oppositely labelled complexes). The 'reduced data' average χ^2 (green) is derived from fitting to all data excluding the anomalous scattering observed for dFICD•hBiP-AMP in 100% D₂O (*b). Error bars represent standard errors of the mean (SEM) of each average χ^2 value and the horizontal dotted lines illustrate the χ^2 values of the input model.

4) The central issue, however, and unless I misunderstood or missed something, is the following: assuming that there was some degree of flexibility, and it were possible to represent it by normal modes. The resulting SANS curve would then correspond to a weighted average of all possible modes. If the SANS curve resulting from this average cannot be distinguished (in terms of chi2) from the SANS curve of a single input structure, it is impossible to infer the presence of flexibility, even though some individual modes might have better or worse fits with the SANS curves.

As stated in the text and highlighted by the reviewer, it is true that the average χ^2 value of the best flex-fit structures (from leaving the FICD dimer interface constrained and unconstrained) are not statistically significantly reduced relative to the average χ^2 of the input structure (against the entire reduced scattering dataset). However, there is much less variation in the goodness of fit of the two output structures with the scattering datasets. This decrease in average χ^2 standard deviation is represented graphically in **Supplementary Fig. 3c** and the difference in variances are statistically significant by F test [between the input structure and the best-fit dimer constrained output structure (#5) $P = 0.0067$ ($F = 6.489$, $DFn = 10$, $DFd = 10$) and between the input structure and the best-fit dimer non-constrained output structure (#1) $P < 0.0001$ ($F = 28.33$, $DFn = 10$, $DFd = 10$). This result alone suggests that the two ensembles of χ^2 values generated from the two flex-fitting methods are significantly different to the ensemble of χ^2 values generated from the input structure.

Moreover, we have now modified the result section to emphasize the fact that the best flex-fit structures are also in better agreement with the experimentally (Stuhrmann analysis) derived R_g values:

Both output structures demonstrate good R_g agreement with the Stuhrmann analysis. Importantly, the complexes' FICD R_g s are increased, and in better agreement with the experimentally derived values, relative to the input structure (**Supplementary Fig. 3d** and **Supplementary Table 1**). In addition, the model derived from flex-fitting without constraints on the FICD dimer interface (as well as possessing the overall lowest average χ^2 value and χ^2 variance) also possesses a BiP R_g which is fully consistent with the Stuhrmann analysis from both oppositely labelled deAMPylation complexes (**Supplementary Table 1**).

It is true that Kratky plots (as shown by the authors in Suppl. Fig. 2g) can indicate the presence of flexibility, e.g. via a plateau or increasing intensities at high q -values. Unfortunately, the q -range accessible in the SANS experiment does not seem to be sufficient to conclude on this point, and the intensities rather display a tendency to decrease at higher q -values which would indicate compact, non-flexible complexes.

The reviewer makes an excellent point. We have amended the legend to **Supplementary Fig. 3g** to more accurately reflect the inferences that can in fact be made from the shape of the Kratky plots. Namely, that:

the scattering intensity profiles are consistent with FICD•BiP-AMP being a **folded** protein complex.

5) Guinier fits (Fig. 2b and Suppl. Fig. 2a): according to the material and methods (p. 23), the vertical dashed lines delimit the ranges of the fits. However, the fits themselves (broken black lines?) extend beyond the lower limits down to smallest angles. It is also not clear why there are several grey dashed vertical limits, but also a single purple dashed limit (Suppl. Fig. 2a). More importantly, it is unlikely that the upper limit of the fit range (defined by $qR_g < 1.3$ for most samples) is the same for all conditions (as suggested by the identical ranges of the black broken lines in all cases). In general, R_g varies as a function of contrast and macromolecular labeling, and this seems to be the case in the present study (as supported by the values reported in the Stuhrmann plot, Fig. 2d). All these points need some clarification. I would suggest that the authors limit and adopt the black broken lines to the actual Guinier fit ranges in each case.

We have, as suggested, now added clarifying remarks in both the relevant figure legends and in the corresponding section of the **Materials and Methods** section. Namely, in the legend of **Fig. 2b**:

Vertical, grey dotted lines represent the q -range for fitting of the linear best-fit curves (black dashed lines). See **Supplementary Fig. 3a**.

In the legend of **Supplementary Fig. 3a**:

The q -range for fitting is denoted as in **Fig. 2a** with the exception that the low- q limit for the scattering of hFICD•dBiP-AMP in 60% D₂O is represented by the vertical, purple dotted line.

And further details pertaining to the qR_g range is now in the **Contrast Variation Small Angle Scattering** section of the **Materials and Methods**:

The upper and lower q limits for fitting are shown (grey, vertical dashed lines in **Fig. 2b** and **Supplementary Fig. 3a** — except for the fitting of hFICD•dBiP-AMP in 60% D₂O buffer where the lower q limit (q_{\min}) is denoted by a purple, vertical dashed line). These fitting ranges resulted in $0.15 < q_{\min}R_g < 0.57$ and $0.39 < q_{\max}R_g < 1.3$ (with the exception of the fitting of dFICD•hBiP-AMP in 80% D₂O buffer data where $q_{\max}R_g = 1.4$).

The Guinier fits shown in **Fig. 2b** and Fig. 3a are extended past the low q limit (q_{\min}) down to $q = 0$ as both the slope and **y-axis** intercept information are important parameters used in the subsequent analyses. Technical limitations of the method of scattering data collection, e.g. the beam stop, prohibit data collection to $q = 0$. Specifically, as can be seen from the Guinier approximation derived equation presented in *Materials and Methods* (Equation 1) the y-intercept value is equivalent to $\ln(I(0))$. The resulting $I(0)$ values for each scattering curve are then used in both the M_w calculation and for calculation of the contrast match point.

Finally, the authors seem to denote the zero y-axis by dotted black symbols. This choice is confusing since it recalls the symbols of the flex-fits in Fig. 2a.

We acknowledge that this may have been confusing. The dotted line at $y = 0$ has been removed from the figure panels in question. We thank the reviewer for pointing this out.

Finally, it must read “Supplementary Fig. “2a” (and not “2b”) at the bottom of p. 23.

We thank the reviewer for spotting this error. All figure (panel) references have now been updated.

Minor points:

a) Did the authors determine the final deuteration degree of the “match-out deuterated” BiP and FICD proteins experimentally, e.g. by mass spectrometry? Or can they at least cite the theoretically expected values of the deuteration protocols (the information is probably available in references 49 or 50 but it would be preferable if it could be explicitly indicated in the supplement). In this context, it would also be useful if the authors could quickly comment on the distribution of deuteration throughout the protein structures: is it uniform/random for all amino acids or targeted at certain types of amino acids? Finally, was the determined (or expected) deuteration degree in agreement within the error margins of the experimentally determined match-points (Fig. 2c) and the values predicted by MULCh?

The final deuteration level of dBiP-AMP and dFICD was not determined by mass spectrometry.

The matchout deuteration level of a protein is defined as the level of non-exchangeable hydrogen deuteration required such that the scattering length density of the protein in question is very close to that of 100% D₂O. Under such circumstances the protein has zero contrast from the fully deuterated solvent and effectively becomes invisible. It is known that 85% D₂O in the growth medium in the presence of hydrogenated glycerol (match-out growth conditions) leads to deuterated proteins that match at around 100% D₂O. This corresponds to an overall deuteration level of about 75% in 100% D₂O buffer.

Matchout labelling is also called random-fractional deuteration. The deuteration level in non-exchangeable positions is, at match-out growth conditions, (i) somewhat dependent on the amino acid composition and (ii) experimentally difficult to obtain with high precision. At the resolution of solution scattering experiments and as a first approximation, the random-fractional deuteration of the protein can be considered to be rather uniform.

As described by Leiting *et al.*¹, 85% D₂O in the growth medium would lead to proteins that are about 65–70% deuterated in non-exchangeable hydrogen positions. A 65% deuterated protein in non-exchangeable positions corresponds to a protein with an overall deuteration level of about 75% in pure D₂O.

Based on the SANS data and the contrast match point values described in this paper, the level of deuteration of non-exchangeable hydrogens was estimated to be 66.5% deuteration of dBiP-AMP and 63.8% deuteration of dFICD. These values are very close to values reported by Leiting¹ and found for a model protein (MBP) expressed in *E. coli* under the same conditions. The deuteration level of MBP was determined by MS to be 64.1% of the non-exchangeable hydrogen atoms for a contrast match point of 99.5% D₂O².

We have expanded the methods section to include a comment to this end:

Comparison of the experimental CMPs with theoretical values calculated by MULCh⁶² (which takes into account buffer composition effects (at 20 °C) and protein sequence, whilst assuming a 1:1 complex and 95% labile H/D-exchange) suggested that there was 66.5% deuteration of dBiP-AMP and a 63.8% deuteration of dFICD. Note, these calculated values of non-exchangeable hydrogen deuteration are in-line with those expected from the 85% deuterated media and non-deuterated carbon source *E. coli* growth conditions, see above. For instance, under the same growth conditions, maltose binding protein was found to be 64% deuterated at non-exchangeable hydrogens by intact protein mass spectrometry⁵⁶.

b) Supplementary Fig. 2b reveals a certain tendency of the quality of fits as a function if either the one or the other protein is deuterated: in the case of the hFICD-dBiP complex, the best fits are obtained at high D₂O levels of the solvent, whereas for the oppositely labeled complex, the best fits are obtained at low D₂O levels. Can the authors comment on this tendency?

If one excludes the seemingly anomalous scattering of dFICD•hBiP-AMP in 100% D₂O (as conducted in the analysis of the 'reduced dataset'), the scattering of dFICD•hBiP-AMP from 0–90% D₂O appears relatively uniform. The same holds true for the flex-fit analysis without the FICD dimer interface constraints (this is now presented alongside the previous χ^2 heat map in **Supplementary Fig. 3b** (was **Supplementary Fig. 2b**)).

It is true that for the majority heterotetramer models the χ^2 values from the model fitting against the scattering data generated by hFICD•dBiP-AMP, in both 0 and 10% D₂O, appears relatively large relative to the rest of the scattering data fits. It is possible that this reflects an instability of deuterated BiP-AMP in low %D₂O buffer. However, the lack of any large positive deviation from linearity at low-*q* in the Guinier plot presented in **Supplementary Fig. 3a** appears inconsistent with a particular (aggregation inducing) instability of the protein complex under these buffer conditions. It is noteworthy, that the best flex-fit models (#5 from the restrained dimer interface analysis and #1 from the unrestrained dimer interface flex-fit analysis) no longer exhibit any such trend across all scattering curves of the reduced data set (**Supplementary Fig. 3b**). This is consistent with these flex-fit models possessing more reliably/uniformly good fits across the entire scattering data set (and hence the statistically significant reduction in χ^2 variance) and suggests that these models may well be more representative of the heterotetramer solution structure. See answer to point 4.

c) The statement/observation that leaving the dimer interface unrestrained yields better fits to the SANS curves than a restrained interface (p.8, bottom) seems obvious due to the greater degree of freedom, and needs some explanation.

Indeed, the fact that the χ^2 is reduced upon increasing the degrees of freedom of the fitting strategy is trivial. However, this was not the intended message of the sentence in question. This has now been reworded as follows:

Interestingly, the best-fit structure derived from leaving the high affinity FICD dimer interface unconstrained (mean χ^2 goodness-of-fit across the reduced data set 1.7 ± 0.4) maintains an intact dimer interface and is in fact more similar in overall conformation to the input structure than that obtained with a restrained dimer-interface (mean χ^2 2.4 ± 0.8), with an RMSD of 5.4 and 7.1 Å (across 1,892 C α pairs), respectively.

d) It is a little bit confusing to present an average chi2 value of 3.4 with a standard deviation of +/- 4 (p. 7, bottom). Chi2 are always positive. Can the authors comment?

Although all χ^2 values are positive the large χ^2 values that occur from fitting to some of the datasets (in particular dFICD•hBiP-AMP in 100% D₂O and hFICD•dBiP-AMP in both 0 and 10% D₂O) results in the large χ^2 variance of the input structure (see **Supplementary Fig. 3b**) and hence a χ^2 SD of ± 4 (of the input model against all scattering curves).

e) A couple of uncommon abbreviations (e.g. "CI", "SEM") should be introduced throughout the manuscript upon their first occurrence.

This has now been rectified.

f) Please provide some general references for SAXS and in particular for SANS when they are first mentioned (p. 7, top).

References 26–29 have now been introduced, at the beginning of the SANS results section, as suggested.

g) The black dots in Fig. 2a are hardly visible. Please either use a different symbol or increase dot size.

The dot size has been increased in both **Fig. 2** and **Supplementary Fig. 3** in order to aid clarity.

Reviewer #2 (Remarks to the Author):

The manuscript by Perera et al reported a crystal structure of a FICD-BiP complex stabilized by a number of mutations on both FICD and BiP. This structure was validated by SAXS/SANS and biochemical analysis on mutations. As the master regulator of protein homeostasis in ER, BiP is regulated through AMPylation by FICD. Understanding the molecular mechanism of the AMPylation of BiP by FICD is important for dissecting the regulation of BiP's function in stress response. This manuscript is from the laboratory of Dr. David Ron, who is one of the leaders in the chaperone field and has published extensively in the area of BiP regulation by AMPylation.

However, there are a number of major concerns to prevent it from publishing.

1. Although the structure determination, solution and biochemical analysis are solid, there is no novel mechanism proposed. The model proposed in Fig. 6 is essentially identical to the model published in their 2019 EMBO paper (Fig. 7) except that BiP was positioned. Even the claimed “unanticipated finding” on Glu234 flexibility was published in the EMBO paper. It seems that nothing mechanistically new was gained through this manuscript.

As noted by the reviewer, the mechanism of FICD-mediated deAMPylation had been the subject of speculation in the past. Our speculative inferences, as pointed out in the manuscript, were based on the observation that Glu234 and His363 were required for FICD-mediated deAMPylation of BiP and that unmodified BiP and AMP were the products of the deAMPylation reaction. Plausible as these speculations might have been, crucial details were missing; hard-won details that are provided here for the first time. As highlighted in the **Discussion** and in **Supplementary Fig. 8** a number of other catalytic mechanisms could have generated the observed deAMPylation products. The current study, therefore, provides experimental details that validate a mechanism of eukaryotic deAMPylation previously proposed and clarifies the oligomeric-state linked regulatory role of Glu234 in the catalysis of deAMPylation. These, in our view, constitute considerable new mechanistic insights. The word ‘unanticipated’ has been removed from the abstract.

2. To support their FICD-BiP crystal structure, SAXS/SANS was carried out. There are a number of serious issues with this work.

1) The authors kept on describing this work as “solution structure”. SAXS/SANS can not generate any solution structure. They are simply models/predictions. These models/predictions are of very low resolution (less than 20 Å). These solution approaches generate only 2D data and try to fit 3D structures. Their application is quite limited.

We have removed various references to ‘solution structures’ although this term is commonly used in the literature to describe structural models derived from small angle scattering (SAS) experiments (for example³⁻⁵). The small angle scattering biological data bank estimates that SAS experiments can provide “structural information on biological macromolecules in solution at a resolution of 1–2 nm” (<https://www.sasbdb.org/>). To quote from a review on the subject⁶: “Although small-angle solution scattering is often described as a low-resolution technique (as it does not provide information on atomic coordinates), it is more appropriate to describe it as a technique capable of providing high-precision information with respect to size and shape.”

We maintain that the SANS data presented here lends strong support to the conclusion that BiP deAMPylation is carried out by a FICD dimer, whose architecture is depicted in **Fig. 1c**.

In the current manuscript, we believe we have demonstrated considerable utility of contrast variation SANS both through the more traditional analysis of the low- q scattering region (which provides information about the complex internal arrangement and R_g information) and by a recently developed flexible-fitting approach (across the entire accessible scattering range). The utility of the latter to determine biologically relevant protein motions (i.e. deviations of a protein's average solution state from that accessible in crystallo) has also been recently demonstrated by others^{4,5}.

2) When fitting their SAXS/SANS data on full-length BiP, the authors used their truncated BiP structure with modeled SBD alpha. Structures of full-length BiP in complex with ATP are available. The authors should use these structures. Otherwise, the deduced model is less reliable since it's based on modeled structures.

We choose to model the complete lid of BiP-AMP (in the deAMPylation complex heterotetramer) by aligning the full-length BiP:ATP structure (PDB 5E84) in the region of SBD α -A (which is also present in the heterodimeric deAMPylation complex). The SBD α -B and α -helical bundle from PDB 5E84 was then combined with the AMPylated (and FICD bound) BiP to generate the modelled full-length deAMPylation complex BiP (**Fig. 1c**). As the only structured BiP region missing from the heterodimeric deAMPylation complex BiP is the distal part of the lid, and as localised conformational changes induced by AMPylation of BiP(SBD β) $\ell_{7,8}$ and engagement of BiP-AMP with FICD are only present in the deAMPylation complex BiP (and absent from PDB 5E84) we consider the model in **Fig. 1c** to provide the best estimate of the structure of a heterotetrameric deAMPylation complex (derivable from extant crystallographic data). Moreover, the $\ell_{3,4}$ truncation introduced into the BiP:ATP of PDB 5E84 causes further SBD β conformational changes in $\ell_{3,4}$ and $\ell_{5,6}$ (relative to other ATP-state Hsp70 structures e.g. PDB 4B9Q and 5O4P). These are not present in the heterodimeric deAMPylation complex BiP on which the modelled heterotetramer is based. See below.

Alignment (via the NBD) of BiP:ATP (PDB 5E84; blue NBD and green SBD) with SBD α -truncated BiP-AMP:Apo (PDB 5O4P; yellow) and DnaK:ATP (PDB 4B9Q; grey). The deviation in SBD β conformation visible between BiP:ATP and BiP-AMP:Apo is largely localised to flexible loops (in particular $\ell_{5,6}$), attributable to the corrupting effects of the $\ell_{3,4}$ truncation introduced into the former and AMPylation of $\ell_{7,8}$ in the latter.

As is clear from the above structural superposition each of the structures represent very similar ATP-state Hsp70 structures. For completeness, a full alignment with PDB 5E84 structure is presented in **Supplementary Movie 1** and alignment with DnaK:ATP (bound to a J domain protein, PDB 5NRO) is now presented in **Supplementary Fig. 6a**.

3) For these studies, the authors first need to show conformational homogeneity of the protein sample. Otherwise, it's almost impossible to deduce any meaningful structural information.

As noted in the answer to Reviewer #1 point (1) and minor point (b) the SANS data itself (especially at %D₂O concentrations close to the contrast match point and the linearity of scattering at low- q in the Guinier plots) is highly indicative of the samples being highly homogenous and monodisperse with a stoichiometry and arrangement very close to that predicted by the model presented in **Fig. 1c**. In

addition, we present below representative gel filtration chromatograms from the purification of the oppositely labelled heterotetramer samples in 0% D₂O buffer (dFICD•hBiP-AMP and hFICD•dBiP-AMP, respectively, see **Materials and Methods**):

And their subsequent buffer exchange into 100% D₂O solvent (at the ILL):

In both cases the left-most peak (which was relatively uniform and corresponded in elution time to a heterotetrameric complex) was pooled and concentrated before being subjected to neutron scattering.

3. For the reported crystal structures, I/σ at the highest resolution shell is 1, i.e. $I=\sigma$. There is no signal at all. The 100% completeness is meaningless.

We respectfully disagree with the reviewer on this point noting the following arguments in support the resolution cut-off of the state 2 deAMPylation complex presented here:

1. Although the signal magnitude is approximately equal to the standard deviation of the signal it is clear that the diffraction in the high-resolution shell still contains substantial information content, as indicated by the half-dataset correlation ($CC_{1/2}$ 0.536). If there were no information content in the given high-resolution shell the $CC_{1/2}$ would equal 0.
2. Indeed the $CC_{1/2}$ has been robustly demonstrated to be a much better indicator of the correct resolution cut-off of a dataset (than other metrics including I/σ), see⁷.
3. As exemplified in⁷, we have carried out a paired refinement of the dataset, demonstrating in an unbiased fashion that inclusion of the high-resolution data improves the resulting model: Analysis against the lower resolution diffraction data (2.0 Å high-resolution cut-off, in which mean I/σ in the high-resolution shell of 2.04-2.00 Å is now 2.1) gives a poorer R_{work}/R_{free} with a model refined at 2.0 Å (0.194/0.222) than with the model refined at the original low resolution ($CC_{1/2}$ based) cut-off of 1.87 Å (0.204/0.206).
4. The analysis of X-ray crystal datasets in which I/σ in the high-resolution shell < 1 is now relatively commonplace throughout the field. To cite just one example, a paper published in

Acta Crystallographica from the lab of Wayne Hendrickson⁸ contains a dataset with a mean I/σ in the high resolution shell of 0.7 ($CC_{1/2}$ 0.319).

4. The authors kept on saying that the ATP-bound conformation of BiP is essential for FICD binding based on their complex structure. This seems one of the key points of the manuscript. The authors should test this claim using BiP mutations.

We have previously demonstrated that FICD specifically AMPylates⁹ and binds¹⁰ the ATP-state of BiP and not the domain-undocked state. We have now made references to these findings more apparent in the manuscript, where applicable.

In addition we have now data which directly demonstrates that the TPR domain of FICD specifically binds BiP:ATP and not BiP:Apo (new **Fig. 4b–c**). We thank the reviewer for stimulating this addition to the manuscript.

Moreover, Hsp40 co-chaperones also specifically recognize and bind the ATP-bound conformation of Hsp70s. Do the observed BiP-FICD interfaces overlap with the Hsp70-Hsp40 interfaces?

Indeed, the TPR domain of FICD does engage BiP via a similar tripartite Hsp70 interface to that bound by the J domain of J domain proteins (JDs). In fact the salt bridge formed by the conserved HPD motif of JDs and BiP Arg197 is mimicked in the engagement of FICD(TPR1) with BiP(NBD). This structural similarity is now highlighted in the new **Supplementary Fig. 6a**.

5. Hsp70s are highly conserved, especially the ATP-bound conformation as shown by a number of structural analyses. Are the BiP residues involved in interacting with FICD conserved in Hsp70s?

We have now addressed the degree of evolutionary conservation of the deAMPylation complex interface (across Hsp70 and Fic protein homologues) in the new **Supplementary Fig. 1c**. This structural depiction of evolutionary conservation indicates that the interacting residues on the BiP surface are not particularly conserved across Hsp70s (with the exception of Arg197, see above, and Pro444). The evolutionary conservation is much higher on the FICD surface of the complex. However, as noted in the figure legend it has recently been demonstrated that the FICD(TPR) interacting BiP residues are conserved amongst metazoan BiP homologues¹¹.

If they are, does FICD modify cytosolic Hsp70 and Hsc70s since FICD has been found in the cytosol?

We have not addressed this question experimentally, however, it has recently been demonstrated that FICD can modify human Hsp70 and Hsc70 *in vitro*. The relevance of these findings remains unclear as there is no evidence that endogenous FICD is appreciably localised outside of the ER. Studies, to date, identifying cytosolic FICD targets have either been conducted *in vitro* or *ex vivo* settings or have been dependent on the massive overexpression of a hyper-AMPylating FICD variant. Therefore, the biological relevance of FICD's ability to AMPylate metazoan Hsp70 and Hsc70, *in vitro*, remains to be determined.

In addition, the authors should do a detailed structural comparison of the reported FICD-BiP complex with published Fic-enzymes in complex with their substrates (such as the IbpA-Cdc42 complex). It seems that the TPR-like motifs of IbpA also interacts with Cdc42. Are there any similarities?

In response to the reviewer's suggestion we have included a comparison with the post-AMPylation Fic-substrate complex of IbpA•Cdc42-AMP (new **Supplementary Fig. 2**). This complex is dissimilar to the deAMPylation complex of FICD•BiP-AMP. The two (unrelated) substrate-specific targeting modules of IbpA and FICD are annotated (arm and TPR domains, respectively) and engage their distinct substrates in different fashions. As noted in the text both Fic proteins engage the target residue bearing loop in a sequence-independent fashion via the Fic domain flap. However, the hydrophobic clamping of the target residue by the flap (noted in IbpA•Cdc42-AMP and in other pseudo-substrate engaged Fic proteins^{12,13}) is absent in the deAMPylation complex.

6. The authors should test the interface between the FICD catalytic domain and the SBD beta of BiP where T518 is located. The key question is how BiP-T518 is recognized by FICD as a substrate. Can their structure provide any insights?

The BiP(SBD β) region bearing Thr518 ($\ell_{7,8}$) is engaged in an intermolecular β -sheet with FICD's Fic domain flap (**Fig 1b(ii)** and **Supplementary Fig. 2a**), see above. This is characteristic of Fic protein substrate or pseudo-substrate engagement^{12,13}. The resulting backbone-backbone hydrogen bonds are not readily amenable to investigation by mutagenesis. There are a small number of other sidechain mediated contacts between the BiP(SBD β) $\ell_{7,8}$ and $\ell_{5,6}$ and FICD's Fic domain (**Supplementary Fig. 1d**) a number of which upon disruption by mutation have been observed to affect BiP AMPylation (see¹¹ and new **Supplementary Fig. 11e**).

7. Throughout the manuscript, the authors have been comparing the new FICD-BiP complex structure with only structures solved by their own group, ignored the structural work on BiP and other Hsp70s published by other groups. To understand mechanism, their BiP structures may not be the best for comparison since these structures carry more truncations and mutations. For Fig. 1, besides using their published SBD alpha truncation structure of BiP for comparison, they should compare their new FICD-BiP complex structure with the full-length BiP structures.

See answer to (2), **Supplementary Movie 1** and the new **Supplementary Fig. 6a**

Overall, the authors referenced their published work extensively. The authors should also reference works from other labs.

Where PDB files were previously referenced, we have now included specific references to the original manuscript (e.g. PDB 5E84¹⁴).

Of the 70 papers referenced in this manuscript a total of 9 originate from our lab. Not counting papers only referenced in the **Materials and Methods** section there are a total of 39 references of which 7 originate from our lab. We believe this quantitative analysis speaks against an undue bias in referencing.

Page 9: the third line, should reference Hsp70-ATP and Hsp70-ADP crystal structures, not only NMR work.

We thank the reviewer for allowing us to clarify this sentence, it now reads:

This flexibility is inaccessible to crystallographic analysis of BiP (complexes) but is consistent with previous observations of Hsp70 conformational dynamics in the Hsp70 ATP-state^{5,31}.

8. TRPox: should test the reduced mutant protein to exclude the possibility that the observed effect is due to the cysteine mutations themselves. Does the reduced form behave like WT? If not, the observed effect is caused by the Cys mutations, not the disulfide bond.

We thank the reviewer for suggesting this line of experimentation. In order to address this point, we have characterised the melting temperature of the cysteine reduced form of this (monomeric FICD) protein (this data has now been added to **Supplementary Fig. 4a–b**) and compared the ability of monomeric and dimeric reduced and oxidised variants to bind to BiP-AMP:ATP and BiP:ATP (new **Supplementary Fig 4f** and **Supplementary Fig 6b**, respectively). Both the thermal stability and BiP binding characteristics of the FICD variants were significantly altered by intramolecular disulphide bond formation. As expected FICD^{L258D-H363A}(TPRred) was considerably less stable than its oxidised counterpart (on account of the weakening of the TPR domain to capping helix and TPR domain to Fic domain contacts induced by the cysteine mutagenesis). Importantly, unlike FICD^{L258D-H363A}(TPRox), FICD^{L258D-H363A}(TPRred) did not reversibly associate with either BiP-AMP or BiP:ATP (new **Supplementary Fig 4f** and **Supplementary Fig 6b**, respectively), consistent with its thermal instability ($T_m < 40$ °C, **Supplementary Fig. 4a–b**). However, despite the fact that FICD^{L258D-H363A}(TPRox) appears to interact more avidly with both BiP-AMP and BiP:ATP than FICD^{L258D-H363A} (and FICD^{L258D-H363A}(TPRred)) it still possesses diminished deAMPylation and AMPylation catalytic activities — implicating a role of TPR domain flexibility in the k_{cat} of both enzymatic modalities.

The last three paragraphs of **Results: The deAMPylation complex crystal structure is representative of the internal arrangement of dimeric FICD engaged with AMPylated BiP in solution** and the fourth paragraph of **FICD's TPR domain is responsible for recognition of unmodified ATP-state BiP** have now been re-written to include reference to the experiments conducted with TPRred FICDs.

9. Fig. 3bii, there is no significant difference between the WT and the three mutations (E105R, H131A, and TRPox), especially H131A. But, the authors described in the manuscript that all the mutants have significant difference. This is misleading.

In order to more clearly demonstrate the differences referred to in the text, the second dissociation phase (in which the dissociation buffer is supplemented with 2 mM ATP) of the representative BLI trace in **Fig. 3b(ii)** has been fit to a bi-exponential decay. The kinetic parameters of the biphasic dissociation of the FICD variants from BiP-AMP are shown in the new **Supplementary Fig. 4e**. The quantification demonstrates that mutation of FICD(TPR) stimulates the dissociation from BiP-AMP (in particular, through increasing the slower dissociation rate constant (k_{slow}) and the percentage of the biphasic dissociation attributed to the fast dissociation (%fast)).

10. For the BioLayer Interferometry, the authors should do the BiP apo and BiP-ADP controls to exclude the possibility that FICD binds BiP as a substrate.

All BiPs used in the current study are both inhibited with respect to their ATPase activity (Thr229Ala) and substrate binding ability (Val461Phe). Using the same immobilised BiP construct we previously demonstrated that FICD did not bind appreciably to BiP:Apo but did bind to BiP:ATP¹⁰.

Moreover, should check ATP hydrolysis by BiP at 30 degree during the course of the experiments. Normally Hsp70s hydrolyze ATP completely within 30 min at 30 degree.

The reviewer expresses a valid concern pertaining to the rate of BiP-mediated ATP hydrolysis. We have previously established that N-terminally immobilised BiP^{T229A-V461F}:Apo, which responds to ATP by attaining a domain-docked configuration, maintains that domain-docked state for thousands of seconds, after exchange into a buffer lacking ATP¹⁰. AMPylation of BiP has been previously documented to further decrease its intrinsic ATPase and activity and heavily bias BiP towards its domain-docked ATP-like state^{9,15,16}.

Furthermore, the rate of stochastic domain undocking of immobilised BiP ligands is unlikely to vary between samples introduced into different analytes. Therefore, as the purpose of the BLI experiments, presented in **Fig. 3–4**, is to correlate structural information with the ability of FICD variants to bind BiP±AMP (when introduced into the assay as analytes), the comparison of (parallel) FICD variant binding events is unlikely to be influenced by rate of BiP:ATP undocking (following exposure to an ATP-depleted buffer at $t = 0$).

11. Fig. 3C: why was E105R not tested?

Whilst we agree with the reviewer that Glu105Arg would have been yet another informative mutation, for reasons of experimental expediency it was not tested. That said, we believe that the panel of mutants that were tested is adequate to establish that the contacts between FICD and BiP observed in the crystal are relevant to the interaction of the two proteins, *in vitro* and *in vivo*.

Reviewer #3 (Remarks to the Author):

The eukaryotic FICD enzyme targets the Hsp70 chaperone BiP from the ER. FICD is a bifunctional enzyme with antagonistic activities: AMPylation of BiP inactivates the chaperone, whereas deAMPylation relieves the block. The deAMPylation activity of FICD was discovered by the Ron group a few years ago (Preissler et al., 2017), followed by an investigation into the structural determinants (oligomeric state) of FICD regulation (Perera et al., 2019).

Now, Perera et al report the crystal structure of the Michaelis-Menten complex of FICD with AMPylated BiP. Turnover of the BiP-AMP substrate was thereby prevented by using a FICD mutant that had the histidine of the active site replaced by alanine. The structure represents an important advancement in our understanding of (1) FICD/target recognition, which is dependent on the conformation of the BiP target, and (2) the mechanism of FICD catalyzed deAMPylation. Latter result may well be of relevance also for the understanding of bacterial Fic enzymes (see the quoted Veyron et al., 2019 paper).

More specifically, the central role of FICD Glu234 in the deAMPylation reaction (in addition to its well-known inhibitory role in AMPylation) is rationalized in that it directly aligns the hydrolytic water. The statement in the abstract, however, that monomerization-induced increase in flexibility of this glutamate is an “unanticipated finding” of this study seems not to be correct, since this was the major conclusion of the Perera et al., 2019 paper on the differential regulation of FICD.

Since the crystal structure was determined from a monomerization mutant, solution studies were carried out on the dimeric wild-type enzyme confirming that the same FICD/BiP interactions occur as in the crystal. Furthermore mutant data are presented validating the observed FICD/BiP interactions and their dependence on the conformational state (domain-docked or not) of BiP.

The presentation of the paper should be stream-lined and improved considerably. Wouldn't it be better to (1) present the overall complex structure, (2) show the solution and mutant studies that are relevant to the overall complex structure, and (3) zooming into the details of the active site and discuss deAMPylation mechanism?

We thank Professor Schirmer for this suggestion.

We have now substantially reorganised the figures and results section to provide what we feel is now a more streamlined and more logical progression of the narrative. We now present the manuscript in the following order:

1. A more global analysis of the crystal structure of heterodimeric deAMPylation complex
2. Validation of the structure in context of heterotetramer by SANS
3. Structure based mutagenesis: effects on BiP-AMP binding and deAMPylation
4. Demonstration that TPR domain is necessary/sufficient for ATP-state (unmodified) BiP binding and AMPylation
5. Focus on deAMPylation complex active site and the inferences that can be made on the mechanism of eukaryotic deAMPylation
6. Comparison of the state 1 and state 2 deAMPylation complexes, which clarifies the role of FICD's monomerisation-induced increase in Glu234 flexibility in the enfeebled ability of the monomeric FICD to catalyse deAMPylation of BiP.

Below I'm adding specific comments/suggestions.

Figures:

0. A figure showing a schematics of the employed constructs (as in Preissler et al. eLife 2017, Fig. 2a) would be very helpful.

Schematics of both BiP and FICD are now provided in the new **Supplementary Fig. 1b**.

1. Amazingly all the main findings are crammed into Fig. 1. I'd suggest to
1.1. Move panel a to Supplement.

Done.

1.2. Show the overview (1.bi) without superposition. Superpositions merely demonstrating no difference are better placed in the Suppl. (if needed at all).

We feel that the superposition of the ATP-state BiP structure in **Fig. 1a** (was **Fig. 1b(i)**) provides valuable information. Namely, illustrating that the deAMPylation complex BiP is also in a domain-docked ATP-like state. Furthermore, given the light grey, translucent colouring of the overlaid structure, we also feel it minimally obscures the rest of the presented figure.

1.3. Show the two contact areas (1.bii) in roughly the same orientation as in the overview.

Having experimented with various orientations in which to represent the two interaction surfaces, the presented view provided the most clarity (enabling all contact points of interest to be viewed in a single view). However, **Supplementary Movie 2** does show a 3D view of both interaction surfaces in considerable detail.

1.4. For the Fic - SBDbeta contact: are there other interactions apart from those with T518-AMP?

See answer to reviewer #2, point (6).

1.5. Show flap with same color in panel bi as in panel bii.

Done.

1.6. Panel c is way too complex. I understand that it is merely there to demonstrate that BiP can bind also to dimeric FICD. This is shown later in Fig. 2 anyway. So, omit this panel.

In order to reduce the complexity of this panel we have now removed the coulombic potential surface representation. We have chosen to retain the panel (in its new representation) as it provides (i) a model for what we believe a heterotetramer composed of two full-length BiP-AMP molecules would look like bound to a dimeric FICD (based on extant crystal structures and as such represents the input model used for the proceeding SANS analysis) and (ii) an indication of the intra-TPR domain deviation observed in the deAMPylation complex (relative to the isolated FICD dimer structure).

2.1. 1.d and 1.e deserve their own figure. Show 1.d without density, for clarity. Don't forget the H-bonds between water * and the main-chain amides of the N-cap of the underlying helix. Enlarge the figure, such that the FICD flap is seen (with H-bonds of augmented β -sheet).

These panels are now enlarged and present in a new **Fig. 6**. In addition a view of the active site is shown in **Fig. 6a** without the polder OMIT map and including additional hydrogen bond information (including highlighting the potential hydrogen bonds of the catalytic water to the anion-binding nest of the Fic motif). More of the FICD flap and augmented intermolecular β -sheet is also now visible in the figure. However, for further details regarding the intermolecular β -sheet hydrogen bonding we

refer the reviewer to the new **Supplementary Fig. 2, Supplementary Fig. 1d, Fig. 4a(ii)** and **Supplementary Movie 2**.

2.2. I'd strongly suggest to add a figure of the competent FICD/AMPPNP complex (6i7l) in same orientation (side-by-side comparison). Should be helpful for discussing FICD's bifunctionality.

In the new **Supplementary Fig. 2** we include an alignment with FICD:MgADP (PDB 4U0U) (which contains the Pa and Mg in the canonical/AMPylation competent position). In addition, **Supplementary Fig. 7** (including the new **Supplementary Fig. 7b**) is also heavily reliant on alignment with the AMPylation competent FICD^{L258D}:MgATP structure (PDB 6I7K).

3.1. Fig. 3a: What is meant by 2 mM ATP dissociation? Addition of ATP to start AMPylation reaction? Explain in the legend.

As highlighted in the relevant figures, text and **Materials and Methods**, all binding assays are carried out with catalytically dead (His363Ala bearing) FICD variants. We previously observed that nucleotide was able stimulate dissociation of FICD from a preAMPylation complex of FICD•BiP:ATP (likely via some form of allosteric mechanism)¹⁰. The increased dissociation rate of FICD from BiP-AMP is most parsimoniously explained through direct competition between the ATP present in the buffer and AMP covalently linked to BiP for the active site of FICD (accelerating the multiphasic dissociation of FICD from BiP-AMP).

Regarding the former allosteric mechanism, inspired by the modelled AMPylation complex active site (new **Supplementary Fig. 7b**), one could speculate as to the nature of the allosteric pre-AMPylation complex destabilisation induced by nucleotide¹⁰ (**Fig. 4d**). The narrow ATP γ -phosphate to FICD Glu234 tolerances, and incomplete γ -phosphate electron density within the isolated FICD^{L258D}:ATP crystal structure¹⁰ (PDB 6I7K) suggests that MgATP binding may itself induce greater displacement of FICD's α_{inh} and Glu234 (via steric and electronic repulsion) than that captured crystallographically. Addition of MgATP to the active site of monomeric FICD may, in this way, lower the propensity of the complex to form the intermolecular FICD(Glu234) to BiP contacts that are intimated by the energy minimised model presented in **Supplementary Fig. 7b**.

3.2. Comparing Figs. 3 and 4 I'm rather confused. Fig. 3 deals with deAMPylation of BiP-AMP only, as suggested by the title? Fig. 4 deals with AMPylation? Would it not be better to organize into separate figures corresponding to the methods employed?

In the new layout of the manuscript **Fig. 3** still addresses the role of FICD mutants (made based on predictions from the deAMPylation complex crystal structure) on the ability of said FICD variants to bind and deAMPylate BiP-AMP. **Fig. 4** now exclusively addresses the ability of the FICD to specifically bind the domain-docked ATP-state of unmodified BiP. In **Fig. 5** we present the effects of FICD(TPR) mutation on the ability of FICD to AMPylate BiP both in vitro and in cells. The results section and respective figure legends have been updated to match the revised order. We believe that the current presentation conveys the scientific narrative of the manuscript in a logical fashion.

4.1. Fig. 6: show Ndelta proton on active His. Show protons of catalytic water. The double arrow between low and high would be better placed in the middle of the catalytic cycle.

Modification have been made to the figure (**Fig. 8**) as suggested.

5. Using depth-cueing would help for the complex figures.

Various new structural panels with depth cueing have now been added to the manuscript and where appropriate pre-existing structural panels have been modified to include depth cueing.

Figure legends:

Always refer to related Suppl. Figures in the legends. E.g. Fig.1 > Fig. S1.

Extra references to supplementary figure panels have now been included within the main figure legends, as suggested.

1061 and in other places: ...attack into the alpha-phosphate >> ... onto ...

We prefer the term 'into' in this context, and feel it better describes the process than the word 'onto'.

1058: ...is essential for AMPylated BiP binding and deAMPylation >> ...is essential for BiP-AMP:ATP binding and deAMPylation 1070: phrase analogously as in 1058 (unless figures get re-organized)

Again, we do not feel there was any issue with the original phrasing. Either version is accurate.

1092: The title is misleading. Glu234 flexibility (compared to what?) is not investigated in this chapter. So, don't confuse results and interpretation.

The figure legend title (now **Fig. 7**) has now been revised.

1104: Be more clear. The authors mean that the initial velocities are the same for the two concentrations and conclude that the enzyme is operating at saturating.

In the legend of **Fig. 7b** we now draw the reader's attention to **Supplementary Fig. 9d–e**, where the derivation and quantification of the very similar initial enzyme velocities (at both substrate concentrations) is explicitly illustrated.

Tables:

5.1 Would be very helpful to have, for comparison, the results (k_{cat} , K_M) of all kinetic measurements in a table. This would also help to make the text more readable.

The k_{cat}/K_M , k_{cat} and K_M values of dimeric and monomeric FICD (previously noted within the text of the last three paragraphs of **Results: Increased Glu234 flexibility enfeebles monomeric FICD deAMPylation activity**) have now been collated into the new **Supplementary Table 2**, and the text adjusted accordingly.

Text:

40 ff: expand, introduce the BiP domains and define the domain-docked, linker-bound conformation

We now include the new **Supplementary Fig. 1b**, which includes a schematic of the BiP protein (with annotated domain architecture).

67: Unclear. You mean substrate concentration and whereby amount of dimeric species?

We have now clarified this line. We were referring to the availability of the substrate for AMPylation (that is to say the concentration of free BiP:ATP in the ER).

Paragraph starting 67: Review the current understanding how AMPylation renders BiP inactive. After all, the modification affects a very peripheral site (T518).

Within the scope of the brief introduction, we felt we were not able to do this justice. We have now included references to literature which deals with this topic as a more primary focus.

89: Define more clearly the constructs (best with a scheme).

Noted and revised (see **Supplemental Fig. 1b**).

97: Sounds strange: "...despite having hydrolysed its bound MgATP".

Noted and edited.

paragraph starting line 94: use present tense throughout.

We have chosen to stick with the use of the past tense, in keeping with the rest of the surrounding results section.

101: You may want to rephrase to s.th. like "Beta-strand augmentation of the Fic flap by the modifyable BiP segment "

We prefer the description of the Fic flap engagement as previously stated: an intermolecular β -sheet between BiP's Thr518 bearing loop ($\ell_{7,8}$) and the Fic domain flap. We feel this is an accurate description. The term augmented (meaning to make greater in size and/or amount) would suggest that there was a pre-existing β -sheet in the region of the Fic flap in the absence of substrate (which is not the case).

104: (implicated in a bacterial Fic protein AMPylation-substrate binding 16,19) >> (as has been observed for bacterial Fic proteins 16,19)

This section of text has now been changed to reflect the inferences made from the new **Supplementary Fig. 2**. It now reads:

The Fic domain flap has previously been implicated in a similar sequence-independent recognition of AMPylation-substrate or pseudo-substrate binding^{15,18} (**Supplementary Fig. 2**).

116: "...linker between ... and the ER membrane" Be more clear. How is BiP anchored to the membrane?

Noted and revised.

127: "oxyanion hole" Term used for proteases. Here the situation is rather an "anionic nest" at the N-term of a helix (see Goepfert et al., 2013 and reference therein)

Noted and revised.

132: "and likely participates in catalysis " >> "most likely constitutes the hydrolytic water molecule "

Noted and revised.

244: “TPR oxidation “ comes out of the blue. In the wild-type, is there a disulfide? This has no been introduced.

Noted and revised.

267 (and elsewhere): Adopt a consistent and simple notation. E.g. BiP:ATP, BiP-AMP:ATP, etc. So, use BiP-AMP:ATP, instead of “unmodified ATP-state BiP”

Noted and revised.

322: “Increased flexibility”: this is not shown here (increased w/r to what ? No comparative measurements shown). It is rather an interpretation. So the title should be rather something like : “ monomerization mutant shows decreased deAMPylation activity”

The title of the figure (now **Fig. 7**) has been revised.

408: It appears improbable that Glu234 can act as a base for deprotonating the hydrolytic water. It forms ion interactions with two arginines, so the energy cost of protonation would be very high (pK shift towards low value)! Please, revise. However, the highly positive environment of the catalytic water (bonded to main-chain amides (oxyanion hole. By the way: better call in anionic nest, see Goepfert reference) and the Mg⁺⁺ in the vicinity may shift the water pK (Lewis base concept).

We used the term ‘catalytic base’ to distinguish the potential role of Glu234 from that of a general base (for precisely the reason outlined in Professor Schirmer’s comment regarding the pK_a of Glu234). We have now made this distinction more explicit:

Glu234, may act as a catalytic (but not a general) base through a mechanism involving late proton transfer analogous to the role played by the catalytic aspartates of some protein kinases^{35,36}.

In the preceding sentence we also draw attention to the schematised reaction pathway (now **Supplementary Fig. 10**) where the roles of Glu234, the anion-binding nest and Mg²⁺ are speculated on further. As stated therein, it seems very likely that the coordination of the α -phosphate group (of BiP-AMP) by Mg²⁺ and its localisation within FICD’s electron withdrawing anion-binding nest (**Fig. 6a** and **Supplementary Fig. 8**) contributes to: (i) stabilising the position of P α (so that it can be efficiently attacked by the catalytic water), (ii) increasing P α ’s electrophilicity and (iii) stabilising the negative charge development during the nucleophilic substitution reaction (see **Supplementary Fig. 10**).

We find it unlikely that the pK_a of the catalytic water is dropped sufficiently by its particular localisation (in an electropositive environment) to induce spontaneous deprotonation (either before or during the formation of the pentavalent transition state), for the following reasons:

1. The putative catalytic water molecule is tightly engaged by the negative Glu234 in the state 1 deAMPylation complex.
2. As can be seen in the new **Fig. 6a**, left panel, the putative catalytic water only (potentially) forms rather weak hydrogen bonds with the protein backbone NH groups of the Fic motif’s anion-binding nest and Arg371 (presumably as a result of the former having originally evolved to bind/stabilise the much larger β -phosphate anion of ATP). Indeed, these hydrogen bonds to the anion-binding nest can only be identified by relaxing the distance and angle hydrogen bond constraints (by 0.4 Å and 20°, respectively, from the geometric criteria listed in¹⁷, as implemented in UCSF Chimera).

3. As the catalytic water forms two high confidence hydrogen bonds — to Glu234 and to a primary hydration shell water of Mg^{2+} (**Fig. 6a**) — the additional putative hydrogen bonds to the anion-binding nest cannot be accommodated within the tetrahedral geometry of hydrogen bonding required of a single (maximally coordinated) water molecule.

Conversely, we find the situation (as specified in the *Discussion* and in **Supplementary Fig. 10**) that Glu234 may be able to (briefly) accept a proton, after the formation of a (protonated) pentavalent transition state, much more likely. Such a role, as a late-stage proton trap, would be analogous to the role played by the catalytic aspartate of some protein kinases (which also possesses a low pK_a)^{18,19}.

However, it is possible that Glu234 never becomes (even transiently) protonated during the reaction mechanism (and therefore does not act as a catalytic base). Glu234's essential role in the deAMPylation reaction^{20,21} could simply be reserved to: (i) properly aligning the catalytic water for in-line nucleophilic attack into the backside of AMPylated BiP's $P\alpha-O\gamma(\text{Thr518})$ phosphodiester bond, (ii) making the catalytic water oxygen atom more nucleophilic (by virtue of Glu234's negative charge) and (iii) facilitating the positioning of the Mg^{2+} first-coordination sphere into a location compatible with the alignment of the catalytic water molecule .

Clarification of the protonation states of the various catalytic residues and transition states, during the course of the deAMPylation reaction mechanism, would likely require quantum mechanical and/or molecular mechanical computational techniques (as conducted for establishing the role of the catalytic aspartate of PKA)^{18,19}. But this is outside of the scope of the current paper.

411: What, in fact, is the role of the Mg^{++} ?

See above.

435: ... binds more tightly to unmodified BiP than BiP-AMP > ... binds more tightly to unmodified BiP than to BiP-AMP By the way: how can this be explained by the structure?

We have now included the new **Supplementary Fig. 7b** which speaks to this point, and we have included speculation on this point (alongside reference to this panel in the *Discussion*).

447: The last sentence of the discussion is cryptic. Can't be understood without checking the cited papers probably.

Noted and revised.

Point-by-Point References

1. Leiting, B., Marsilio, F. & O'Connell, J. F. Predictable deuteration of recombinant proteins expressed in *Escherichia coli*. *Anal. Biochem.* **265**, 351–355 (1998).
2. Dunne, O. *et al.* Matchout deuterium labelling of proteins for small-angle neutron scattering studies using prokaryotic and eukaryotic expression systems and high cell-density cultures. *Eur. Biophys. J.* **46**, 425–432 (2017).
3. Neylon, C. Small angle neutron and X-ray scattering in structural biology: Recent examples from the literature. *European Biophysics Journal* **37**, 531–541 (2008).
4. Vermot, A. *et al.* Interdomain Flexibility within NADPH Oxidase Suggested by SANS Using LMNG Stealth Carrier. *Biophys. J.* **119**, 605–618 (2020).
5. Grudin, S., Laine, E. & Hoffmann, A. Predicting Protein Functional Motions: an Old Recipe with a New Twist. *Biophys. J.* **118**, 2513–2525 (2020).
6. Jacques, D. A. & Trehwella, J. Small-angle scattering for structural biology - Expanding the frontier while avoiding the pitfalls. *Protein Science* **19**, 642–657 (2010).
7. Diederichs, K. & Karplus, P. A. Better models by discarding data? *Acta Crystallogr. Sect. D Biol. Crystallogr.* **69**, 1215–1222 (2013).
8. Wang, W. & Hendrickson, W. A. Intermediates in allosteric equilibria of DnaK–ATP interactions with substrate peptides. *Acta Crystallogr. Sect. D Struct. Biol.* **77**, 606–617 (2021).
9. Preissler, S. *et al.* AMPylation matches BiP activity to client protein load in the endoplasmic reticulum. *Elife* **4**, e12621 (2015).
10. Perera, L. A. *et al.* An oligomeric state-dependent switch in the ER enzyme FICD regulates AMPylation and deAMPylation of BiP. *EMBO J.* **38**, e102177 (2019).
11. Fauser, J. *et al.* Specificity of AMPylation of the human chaperone BiP is mediated by TPR motifs of FICD. *Nat. Commun.* **12**, 2426 (2021).
12. Xiao, J., Worby, C. A., Mattoo, S., Sankaran, B. & Dixon, J. E. Structural basis of Fic-mediated adenylation. *Nat. Struct. Mol. Biol.* **17**, 1004–10 (2010).
13. Goepfert, A., Stanger, F. V., Dehio, C. & Schirmer, T. Conserved Inhibitory Mechanism and Competent ATP Binding Mode for Adenylyltransferases with Fic Fold. *PLoS One* **8**, e64901 (2013).
14. Yang, J., Nune, M., Zong, Y., Zhou, L. & Liu, Q. Close and Allosteric Opening of the Polypeptide-Binding Site in a Human Hsp70 Chaperone BiP. *Structure* **23**, 2191–2203 (2015).
15. Preissler, S. *et al.* AMPylation targets the rate-limiting step of BiP's ATPase cycle for its functional inactivation. *Elife* **6**, e29428 (2017).
16. Wieteska, L., Shahidi, S. & Zhuravleva, A. Allosteric fine-tuning of the conformational equilibrium poises the chaperone BiP for post-translational regulation. *Elife* **6**, e29430 (2017).
17. Mills, J. E. J. & Dean, P. M. Three-dimensional hydrogen-bond geometry and probability

- information from a crystal survey. *J. Comput. Aided. Mol. Des.* **10**, 607–22 (1996).
18. Valiev, M., Kawai, R., Adams, J. A. & Weare, J. H. The role of the putative catalytic base in the phosphoryl transfer reaction in a protein kinase: First-principles calculations. *J. Am. Chem. Soc.* **125**, 9926–9927 (2003).
 19. Cheng, Y., Zhang, Y. & McCammon, J. A. How does the cAMP-dependent protein kinase catalyze the phosphorylation reaction: An ab Initio QM/MM study. *J. Am. Chem. Soc.* **127**, 1553–1562 (2005).
 20. Preissler, S., Rato, C., Perera, L., Saudek, V. & Ron, D. FICD acts bifunctionally to AMPylate and de-AMPylate the endoplasmic reticulum chaperone BiP. *Nat. Struct. Mol. Biol.* **24**, 23–29 (2017).
 21. Veyron, S. *et al.* A Ca²⁺-regulated deAMPylation switch in human and bacterial FIC proteins. *Nat. Commun.* **10**, 1142 (2019).

REVIEWERS' COMMENTS

Reviewer #1 (Remarks to the Author):

The authors have replied to all my queries in a very detailed and satisfactory way.

Reviewer #2 (Remarks to the Author):

I appreciate the authors have carefully addressed my concerns; however, several concerns remain:

1. Removing “unanticipated” from the abstract clarified the message. However, it seems the presented beautiful structure and solution and biochemical analyses mainly validated the model proposed in their 2019 EMBO paper as stated in their Rebuttal “The current study, therefore, provides experimental details that validate a mechanism of eukaryotic deAMPylation previously proposed...”, i.e., there is no novel mechanism proposed.

2. The authors provided gel filtration chromatograms. It’s better for the authors to include these chromatograms in the Supplementary information.

8. About TRPox, it seems the TRPred has a similar defect as the TRPox. Then, the conclusion on the TRPox is not valid.

10. About the Thr229Ala mutation, it only slows down the ATPase activity by several folds. Thus, the ATPase activity is not inhibited as the authors assumed. The authors should test their Thr229Ala mutation to confirm.

Without validating the ATP hydrolysis at 30 degree during their assays, it is hard to interpret the data although they have published results using BioLayer Interferometry before. Doing the ATPase assay should be straightforward.

11. Lacking the E105R test seems to compromise the integrity of the manuscript and make readers suspect the authors may hide some information.

Point-by-point response to reviewers' comments (received 17-Jun-21) on the revised version NCOMMS-21-06435A (submitted 18-May-21)

Reviewer #1 (Remarks to the Author):

The authors have replied to all my queries in a very detailed and satisfactory way.

Reviewer #2 (Remarks to the Author):

I appreciate the authors have carefully addressed my concerns; however, several concerns remain:

1. Removing “unanticipated” from the abstract clarified the message. However, it seems the presented beautiful structure and solution and biochemical analyses mainly validated the model proposed in their 2019 EMBO paper as stated in their Rebuttal “The current study, therefore, provides experimental details that validate a mechanism of eukaryotic deAMPylation previously proposed...”, i.e., there is no novel mechanism proposed.

We believe there is value not only in proposing plausible hypotheses for how things work — as we have done in the 2019 EMBOJ paper referred to by the reviewer — but also in testing these ideas experimentally, as we have done here.

2. The authors provided gel filtration chromatograms. It's better for the authors to include these chromatograms in the Supplementary information.

This is already a very detail laden paper and including this data would add bulk with little added value.

8. About TRPox, it seems the TRPred has a similar defect as the TRPox. Then, the conclusion on the TRPox is not valid.

Our conclusion, that fixation of the TPRs affects the engagement of substrate is based on differences observed between the oxidized and reduced form of the D160C; T183C double mutants. We believe that the basal defect brought about by this mutation does not undermine that conclusion (please see our response to point 8 of this reviewer's comments on the original version of the paper).

10. About the Thr229Ala mutation, it only slows down the ATPase activity by several folds. Thus, the ATPase activity is not inhibited as the authors assumed. The authors should test their Thr229Ala mutation to confirm.

We are aware that BiP^{T229A} is only impaired in its ATPase activity. In addition to measurements reported in the literature¹⁴ this is also evident from the observation that the BiP^{T229A-V461F}-AMP of the state 2 deAMPylation complex crystal form had clearly hydrolysed the bound ATP. However, as we explain in detail below, the conclusions we draw from the BLI experiment are valid despite leakiness of the Thr229Ala mutation.

Without validating the ATP hydrolysis at 30 degree during their assays, it is hard to interpret the data although they have published results using BioLayer Interferometry before. Doing the ATPase assay should be straightforward.

In the BLI experiment, we immobilised (in parallel) identical Avi-tagged BiP^{T229A-V461F} samples on identical BLI probes and exposed them in parallel to identical solutions of MgATP. This is followed by an identical brief (50 second) immersion of the probes into identical solutions

lacking ATP. At the conclusion of these preparatory steps — which occur before the commencement of the experimental traces shown in **Fig. 3–4** — the samples are identical (see **Supplementary Fig. 4d**). Moreover, they are identical regardless of how much ATP had been hydrolysed and regardless of how much or little ATP hydrolysis has led to domain undocking of the immobilised BiP. Thus, the differences observed in the BLI traces upon these IDENTICAL BiP ligand preparations being confronted by diverse analytes (i.e., FICD mutants in presence or absence of ATP) reflect differences in the BiP-FICD interaction wrought by mutations in FICD, fixation of their TPRs etc. It is this feature on which our conclusions from these experiments rest. Knowledge of the rate of ATP hydrolysis will not affect these conclusions.

Given our assertion that it is the ATP-bound domain docked BiP that engages FICD, one might legitimately ask: How plausible is it to imagine that Avi-tagged BiP^{T229A-V461F} remains in this state, given reasonable assumptions of rates of ATP hydrolysis, timelines of the experiment and the temperatures involved?

We began by measuring the rate of ATP hydrolysis by the Avi-tagged mutant BiP^{T229A-V461F} at 30 °C: k_{cat} $\sim 7.2 \times 10^{-3} \text{ min}^{-1}$].

Shown here is a multi-turnover ATPase assay (based on a NADH oxidation-coupled ATP-regenerating system as described in²², with a total ATP concentration of 5 mM). Each molecule of ATP hydrolysed and subsequently regenerated, results in the oxidation of a molecule of NADH (which in turn results in a reduction in absorbance at 340 nm, A_{340}). Based on the known ϵ_{340} of NADH, subtracting the spontaneous rate of NADH degradation at 30 °C, a BiP-mediated ATP hydrolysis turnover number can be calculated. See above.

If we now make the unfavourable assumption that the rate of ATP-hydrolysis of the N-terminally immobilised Avi-tagged BiP^{T229A-V461F} at 30 °C (in the BLI assays presented here and previously¹⁰) to be even greater and up to four-fold more than the rate of BiP^{T229A} ATP-hydrolysis measured at 20 °C in reference¹⁴ [$k_{cat} \leq 1.0 \times 10^{-2} \text{ min}^{-1}$], at $t = 0$ (i.e. the start of the association step which immediately follows the 50 second baseline step in a buffer lacking ATP), less than 1% of the BiP molecules are likely to have hydrolysed their bound ATP (see **Supplementary Fig. 4d**). Even after the 50 second baseline step and the entire cycle of FICD binding and dissociation in buffers lacking ATP (a 1,350 second duration), the extent of ATP hydrolysis of the (initially) ATP-bound, immobilised BiP is expected to be less than 20%. Moreover, binding of FICD to BiP^{T229A} does not detectably modulate its intrinsic ATP hydrolysis rate: In Fig. 2B from¹⁰ the presence of 2 μM FICD was observed to not detectably increase the amount of ATP hydrolysed into ADP by 5 μM BiP^{T229A}, over the course of 2 h incubation at 30 °C. Thus, it seems reasonable to conclude that $\geq 80\%$ of the immobilised Avi-tagged BiP^{T229A-V461F} in the sample remains ATP-bound and domain docked. This is entirely consistent with its ability to interact with FICD in the mode revealed by the crystals and with the sensitivity of the interaction to mutations in FICD that are predicted to compromise the interaction, as revealed by our experiments.

We have now included these arguments in a new **Supplementary Note 1** and referenced

this note within the figure legend of **Supplementary Fig. 4d**.

11. Lacking the E105R test seems to compromise the integrity of the manuscript and make readers suspect the authors may hide some information.

We thank the reviewer for their concern for our reputation.

Readers and reviewers are understandably suspicious when a paper lacks an obvious experiment that could be readily performed with little effort with tools and setups described in other experiments in the same paper. However, that is **not** the case here:

The Glu105Arg mutants studied in this work were tested for their ability to bind either unmodified BiP:ATP or BiP-AMP:ATP, they were thus constructed and purified in enzymatically dead (His363Ala bearing) FICD backgrounds. These same protein preparations are of no use in measuring the effect of Glu105Arg on in vitro AMPylation or in vitro deAMPylation assays, as this requires catalytically competent versions of the same proteins. Similarly, the mammalian expression construct of FICD^{E105R- L258D} (shown to be defective in **Fig. 5b**) is of no help.

Thus, the experiments suggested by the reviewer would require the construction of bacterial expression plasmids for Glu105Arg mutants within an enzymatically competent FICD background, protein expression, purification, biophysical validation of the protein's integrity etc. We believe our paper, in its current form, provides adequate support for the conclusion that FICD and BiP interact by the mode revealed in the crystal structures and that the catalytic mechanisms thus inferred are correct (and that the effort involved in providing the additional information on the behaviour of E105R in these in vitro enzymatic assays exceeds the knowledge value of the experiment).

To maximise transparency, we have now explicitly noted in the legend of **Fig. 3c** that Glu105Arg mutants were not tested in the context of in vitro deAMPylation or AMPylation assays simply on the grounds of experimental expedience.

Point-by-Point References

1. Leiting, B., Marsilio, F. & O'Connell, J. F. Predictable deuteration of recombinant proteins expressed in *Escherichia coli*. *Anal. Biochem.* **265**, 351–355 (1998).
2. Dunne, O. *et al.* Matchout deuterium labelling of proteins for small-angle neutron scattering studies using prokaryotic and eukaryotic expression systems and high cell-density cultures. *Eur. Biophys. J.* **46**, 425–432 (2017).
3. Neylon, C. Small angle neutron and X-ray scattering in structural biology: Recent examples from the literature. *European Biophysics Journal* **37**, 531–541 (2008).
4. Vermot, A. *et al.* Interdomain Flexibility within NADPH Oxidase Suggested by SANS Using LMNG Stealth Carrier. *Biophys. J.* **119**, 605–618 (2020).
5. Grudinin, S., Laine, E. & Hoffmann, A. Predicting Protein Functional Motions: an Old Recipe with a New Twist. *Biophys. J.* **118**, 2513–2525 (2020).
6. Jacques, D. A. & Trehwella, J. Small-angle scattering for structural biology - Expanding the frontier while avoiding the pitfalls. *Protein Science* **19**, 642–657 (2010).
7. Diederichs, K. & Karplus, P. A. Better models by discarding data? *Acta Crystallogr. Sect. D Biol. Crystallogr.* **69**, 1215–1222 (2013).
8. Wang, W. & Hendrickson, W. A. Intermediates in allosteric equilibria of DnaK–ATP interactions with substrate peptides. *Acta Crystallogr. Sect. D Struct. Biol.* **77**, 606–617 (2021).
9. Preissler, S. *et al.* AMPylation matches BiP activity to client protein load in the endoplasmic reticulum. *Elife* **4**, e12621 (2015).
10. Perera, L. A. *et al.* An oligomeric state-dependent switch in the ER enzyme FICD regulates AMPylation and deAMPylation of BiP. *EMBO J.* **38**, e102177 (2019).
11. Fauser, J. *et al.* Specificity of AMPylation of the human chaperone BiP is mediated by TPR motifs of FICD. *Nat. Commun.* **12**, 2426 (2021).
12. Xiao, J., Worby, C. A., Mattoo, S., Sankaran, B. & Dixon, J. E. Structural basis of Fic-mediated adenylation. *Nat. Struct. Mol. Biol.* **17**, 1004–10 (2010).
13. Goepfert, A., Stanger, F. V., Dehio, C. & Schirmer, T. Conserved Inhibitory Mechanism and Competent ATP Binding Mode for Adenylyltransferases with Fic Fold. *PLoS One* **8**, e64901 (2013).
14. Yang, J., Nune, M., Zong, Y., Zhou, L. & Liu, Q. Close and Allosteric Opening of the Polypeptide-Binding Site in a Human Hsp70 Chaperone BiP. *Structure* **23**, 2191–2203 (2015).
15. Preissler, S. *et al.* AMPylation targets the rate-limiting step of BiP's ATPase cycle for its functional inactivation. *Elife* **6**, e29428 (2017).
16. Wieteska, L., Shahidi, S. & Zhuravleva, A. Allosteric fine-tuning of the conformational equilibrium poises the chaperone BiP for post-translational regulation. *Elife* **6**, e29430 (2017).
17. Mills, J. E. J. & Dean, P. M. Three-dimensional hydrogen-bond geometry and probability information from a crystal survey. *J. Comput. Aided. Mol. Des.* **10**, 607–22 (1996).
18. Valiev, M., Kawai, R., Adams, J. A. & Weare, J. H. The role of the putative catalytic base in the phosphoryl transfer reaction in a protein kinase: First-principles calculations. *J. Am. Chem. Soc.* **125**, 9926–9927 (2003).

19. Cheng, Y., Zhang, Y. & McCammon, J. A. How does the cAMP-dependent protein kinase catalyze the phosphorylation reaction: An ab Initio QM/MM study. *J. Am. Chem. Soc.* **127**, 1553–1562 (2005).
20. Preissler, S., Rato, C., Perera, L., Saudek, V. & Ron, D. FICD acts bifunctionally to AMPylate and de-AMPylate the endoplasmic reticulum chaperone BiP. *Nat. Struct. Mol. Biol.* **24**, 23–29 (2017).
21. Veyron, S. *et al.* A Ca²⁺-regulated deAMPylation switch in human and bacterial FIC proteins. *Nat. Commun.* **10**, 1142 (2019).
22. Preissler, S. *et al.* Calcium depletion challenges endoplasmic reticulum proteostasis by destabilising BiP-substrate complexes. *Elife* **9**, 2020.11.03.366484 (2020).